# A kinematic formalism for tracking ice-ocean mass exchange on the Earth's surface and estimating sea-level change

Surendra Adhikari[1], Erik R. Ivins[1], Eric Larour[1], Lambert Caron[1], and Helene Seroussi[1]

[1]Jet Propulsion Laboratory, California Institute of Technology, Pasadena, CA 91109, USA.

**Correspondence:** Surendra Adhikari (surendra.adhikari@jpl.nasa.gov)

**Abstract.** Polar ice sheets are important components of the Earth System. As the geometries of land, ocean, and ice sheets evolve, they must be consistently captured within the lexicon of geodesy. Understanding the interplay between the processes such as ice-sheet dynamics, solid-Earth deformation, and sea-level adjustment requires both geodetically consistent and mass conserving descriptions of evolving land and ocean domains, grounded ice sheets and floating ice shelves, and their respective interfaces. Here we present mathematical descriptions of a generic level set that can be used to track both the grounding lines and coastlines, in light of ice-ocean mass exchange and complex feedbacks from the solid Earth and sea level. We next present a unified method to accurately compute the sea-level contribution of evolving ice sheets based on the change in ice thickness, bedrock elevation and mean sea level caused by any geophysical processes. Our formalism can be applied to arbitrary geometries and at all time scales. While it can be used for applications with modeling, observations and the combination of two, it is best suited for Earth System models, comprising ice sheets, solid Earth and sea level, that seek to conserve mass.

## 1 Introduction

Recently there has been intense interest in defining the physics involved in determining multidecadal change in the location and the migration rate of the grounding line, a boundary separating a grounded ice sheet from its floating extension, usually a floating ice shelf (e.g., *Nowicki and Wingham*, 2008; *Schoof*, 2012; *Sergienko and Wingham*, 2019). Indeed, how well a numerical model of marine ice sheets predicts the sea-level contribution largely depends on its ability to capture the subtle migration of grounding lines. The non-equilibrium thermodynamics, fluid dynamics, plastic failure criteria, and conditions governing nonlinear stability of ice sheets are, quite generally, up for lively debate. In order to better tackle the difficult nonlinear physics and to better address the associated numerical challenges (e.g., *Schoof*, 2007; *Durand et al.*, 2009; *Sayag and Grae Worster*, 2013; *Favier et al.*, 2016; *Seroussi and Morlighem*, 2018) as well as to define proper observational criteria for locating the grounding lines and their migrations (e.g., *Hogg et al.*, 2018; *Millillo et al.*, 2019), it is important to agree on some of the baseline variables and boundary conditions. Direct interactions with the ocean (e.g., *Seroussi et al.*, 2017; *Nakayama et al.*, 2018) and the solid Earth (e.g., *Gomez et al.*, 2010; *Larour et al.*, 2019) are now seen as critical elements that must

be incorporated into projections, or retrospective paleoclimate simulations, of the rate of grounding line retreat in a warming climate (e.g., *Jones et al.*, 2015; *Whitehouse et al.*, 2017). Given the computational complexity of this problem, however, it is essential to properly define the simple geometrical parameters, primarily moving boundaries at the ice/bedrock/ocean interfaces, for there to be rationally organized intercomparison among various research teams and their results.

A general description of mechanical analysis of ice-sheet evolution at the ice/bedrock/ocean interfaces has been given for a set of simplified geometries (for example in Chapter 3 of *Hutter*, 1983), owing to the lack of constraining data or computational resources. A similar geometric approach is also familiar in the development of glacial isostatic adjustment (GIA) theory for ice sheets, sea level and bedrock evolution following the Last Glacial Maximum with migrating grounding lines and coastlines (e.g., *Milne*, 1998; *Lambeck et al.*, 2003; *Mitrovica and Milne*, 2003). Modern satellite techniques have allowed us

to gain knowledge of both the present locations and migration rates of the grounding line (e.g., *Rignot et al.*, 2011; *Milillo et al.*, 2019). However, both observations and numerical simulations of subtle change in grounding line positions are complicated by the presence of kilometer-scale geometric features, such as ice rises and rumples, rugged fjord geometries, and uneven bedrock topography. These features complicate the required geometrical simplifications used in the previous studies of ice/bedrock/ocean interface changes, especially when the system of ice sheets, solid Earth and sea level is fully interactive

(e.g., *Lingle and Clark*, 1985; *Gomez et al.*, 2013; *de Boer et al.*, 2014; *Konrad et al.*, 2015; *Larour et al.*, 2019). Here we consider a simple level-set method, which has been previously applied for tracking grounding lines (e.g., *Seroussi et al.*, 2014) and calving-front positions (e.g., *Bondzio et al.*, 2017), and generalize it to facilitate a precise tracking of both the grounding lines and coastlines of arbitrary geometries in a seamless manner. The method is very generic and can be used for applications based on modeling, observations, or combination of models and observations.

Evolving bedrock and sea level impact the ice-sheet dynamics via the modulation of bedrock slope, grounding-line positions, and gravitational driving stress. For the marine portions of the ice sheet having retrograde bedrock slopes, this effect promotes the stability, as has been demonstrated by both the observation-based (e.g., *Barletta et al.*, 2018; *Kingslake et al.*, 2018) and the model-based studies (e.g., *Lingle and Clark*, 1985; *Gomez et al.*, 2010, 2015; *Adhikari et al.*, 2014; *Konrad et al.*, 2015; *Larour et al.*, 2019). The inclusion of evolving bedrock and sea level in a dynamical ice-sheet model, however, requires a modification

to the common method of estimating sea-level contribution. The method, based on the concept of ice-height above floatation (e.g., *Bindschadler et al.*, 2013), yields inaccurate results for the marine portions of the ice sheet. *Goelzer et al.* (2020) recently provide appropriate corrections for the effects of bedrock elevation change and externally-forced sea level. Our goal here is to formulate a unified method to calculate the exact fraction of ice thickness that contributes to the sea-level change over a given period by considering evolving bedrock and sea level driven by any geophysical processes. In conjunction with observational

data, the method can be applied to a variety of models such as stand-alone ice-sheet models and those that account for isostatic bedrock adjustment (e.g., *Le Meur and Huybrechts*, 1996; *Pattyn*, 2017) or a self-consistent GRD (gravitational, rotational, deformational) response of the solid Earth (e.g., *Gomez et al.*, 2013; *Larour et al.*, 2019). In the latter set of models, the presented formalism ensures mass conservation in the Earth System by exchanging mass between the land and the ocean, accounting for the induced GRD response of solid Earth, and adjusting the ocean area through migration of grounding lines

and coastlines, simultaneously.

In the following, we begin by presenting a generalized description of land, ocean and ice domains and their respective interfaces (Section 2). We consider a global ocean, composed of an interconnected system of oceanic basins, and distributed system of ice domains, comprising glaciers, ice sheets and ice shelves, that can be straightforwardly employed in any Earth System model in order to track the global mass transport and assess the evolution of a dynamic system of ice sheets, solid

Earth and sea level. In Section 3, we briefly review the common method of estimating sea-level contribution of ice sheets and present a new method, wherein we isolate mass and volume contributions to the ocean, which is critical to accurately drive the GRD response of the solid Earth. In Section 4, we assess our formalism in a broader context of sea-level change and mass conservation in the Earth System. Finally, in Section 5, we summarize the key conclusions.

## 2  Land, ocean and ice domains and their interfaces

To begin our discussion, we consider a spherical planet whose surface is divided into complementary domains of land and ocean. The ocean may be thought of as an interconnected system of oceanic basins – just like Earth's ocean that also includes fjords and marginal seas such as Mediterranean – that are able to freely exchange and redistribute mass between them. This assumption simplifies what would otherwise be an arduous task for mass attribution and conservation in the Earth System. Distributed ice domains including glaciers, ice sheets and ice shelves exist on the land or the ocean (Figure 1). We generally

consider ice domains as part of the land, except where they float on the oceanic water as ice shelves. In order to present mathematical descriptions of these domains and their interfaces at time $t$, we denote 2-D spatial coordinates on the planetary surface by $\omega$. Depending upon the spatial scale (e.g., the ocean versus glaciers), we interchangeably use $\omega$ to represent geographic coordinates $(\theta, \phi)$ or Cartesian $(x, y)$, assuming that an appropriate coordinate transformation is applied. The entire formalism presented in this study can be derived from three field variables: the solid Earth surface (i.e., land surface or sea floor) or simply

bedrock $B(\omega, t)$, mean sea level (MSL) $S(\omega, t)$, and ice thickness $H(\omega, t)$. The first two fields must be defined relative to the same reference ellipsoid (e.g., *Altamimi et al.*, 2016).

Our definition of MSL complies with that given by *Gregory et al.* (2019): Time-mean of sea surface over a sufficiently long period so that the effects of waves, tides, or meteorologically-driven high-frequency fluctuations are eliminated. The period of time-mean may be on the order of 20 years or longer, a timescale over which interactions between sea level and ice sheet

may become important (e.g., *Hillenbrand et al.*, 2017; *Larour et al.*, 2019). One key difference, however, is that the change in MSL in the present context does not account for the steric component that is due to the change in the ocean density. Here, in the strict sense of the word, the change in global-mean of MSL is given by the so-called barystatic sea-level change, which is the global-mean sea-level (GMSL) change due to the exchange of water between the land and the ocean, and the evolving spatial pattern of MSL is dictated by the GRD response of the solid Earth to land-ocean (water or sediment) mass exchange

and the tectonic activities. This definition of evolving MSL is familiar in GIA modeling wherein there is a requirement to solve for a gravitationally self-consistent solution of evolving bedrock and (non-steric) MSL driven by the ice-ocean mass exchange following the Last Glacial Maximum (*Farrell and Clark*, 1976; *Milne and Mitrovica*, 1998). The MSL as defined above represents an equipotential surface whose spatial pattern matches the geoid (*Tamisiea*, 2011).

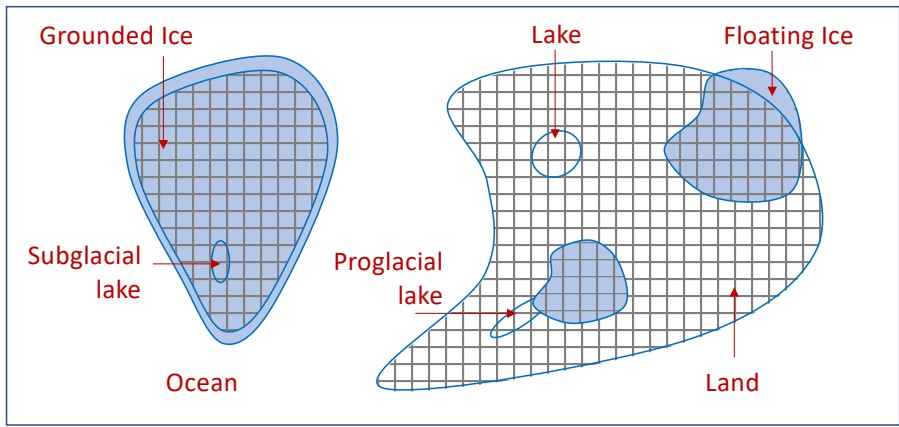

**Figure 1. Conceptual depiction of land, ocean and ice domains in the Earth System.** Gridded areas represent land and the rest ocean. Lakes are considered as part of the land. Ice can have multiple domains, shown here with blue sheds. The land-ocean boundary is generally defined as the coastline, which is called grounding line when it is part of the ice domain. Because our focus is on grounding line migration in marine portions of an ice-sheet/shelf system, we assume that all of ice on land (gridded portions of blue sheds) is grounded. Consequently, floatation of ice on subglacial and proglacial environments are not considered in this study.

## 2.1 Coastlines and grounding lines as a seamless interface

We develop our formalism based on the principle of hydrostatic equilibrium for a system of ice and ocean. Since $H(\omega,t)$ may be considered as a globally-defined field, with $H(\omega,t) = 0$ outside the ice domains, this concept can be generalized to deduce a criterion for delineating boundaries between the land and the ocean, and the floating and the grounded ice. We define:

$$F(\omega,t) = H(\omega,t) - \frac{\rho_o}{\rho_i}\Big[S(\omega,t) - B(\omega,t)\Big], \tag{1}$$

such that $F(\omega,t) = 0$ satisfies the hydrostatic equilibrium between ice and the oceanic water in the marine sectors where $B(\omega,t) < S(\omega,t)$. Here $\rho_i$ and $\rho_o$ are the average densities of ice and ocean water, respectively. Our goal here is to use equation (1) as a basis for defining the land-ocean boundaries consistently. The equation, at first glance, suggests that the ocean (land) takes negative (positive) values of $F(\omega,t)$ and their interfaces have zero values. However, a few aspects should be further clarified. To simplify a mathematical description of the land-ocean boundaries, we ensure the absence of marine ice cliffs that have larger thickness than the floatation height (i.e., negative of the second term on the right-side of the equation) by assuming that $F(\omega,t) \leq 0$ at the "ice front". The ice front that satisfies the equality (inequality) here represents the calving face of a tidewater glacier (an ice shelf). Along the same lines, we assume that the terrestrial ice cliffs are not present where $B(\omega,t) = S(\omega,t)$. These assumptions are generally valid, as the flow is diffusive on the timescale (decades or longer) we are interested in.

We now define a level set of the function $F(\omega,t)$ such that:

$$T(F) = \{(\omega,t) \mid F(\omega,t) = 0\}. \tag{2}$$

This zero-level set may consist of several simple curves, $T_i(F)$, that divide the planetary surface into several non-overlapping regions, $\Omega_i(F)$. Let $\Omega_i^-(F)$ denote the regions in which the function $F(\omega,t)$ takes negative values, and are therefore the candidates of the ocean domain. Since we consider the ocean to be an interconnected water volume, termed the global ocean, as in traditional physical oceanography and sea level studies, only the largest amongst $\Omega_i^-(F)$ forms the ocean domain. Smaller $\Omega_i^-(F)$, if there are any, and their boundaries $T_i(F)$ are considered to be part of the land, meaning they are unable to freely exchange mass with the global ocean by GRD processes. One obvious example of the region that does not belong to the ocean in spite of having $F(\omega,t) < 0$ is a continental trough with bathymetry below MSL. Unless this trough is physically connected to the global ocean via oceanic water, we consider this to be part of the land rather than the ocean. Let $\Omega_S^-(F)$ be the union of all these small non-oceanic regions and $T_S(F)$ be the union of corresponding boundaries. We modify $F(\omega,t)$ to define a new function

$$
\begin{aligned}
\mathcal{F}(\omega,t) &= |F(\omega,t)| + \epsilon \quad \text{if } \omega \in \Omega_S^-(F) \text{ and } \omega \in T_S(F) \\
&= F(\omega,t) \qquad\quad \text{otherwise,}
\end{aligned}
\tag{3}
$$

so that its zero-level set

$$
\mathcal{T}(\mathcal{F}) = \{(\omega,t) \mid \mathcal{F}(\omega,t) = 0\}
\tag{4}
$$

represents the land-ocean boundaries. Here $\epsilon$ is a positive number to ensure $\mathcal{F}(\omega,t) > 0$ at $\omega \in T_S(F)$.

The land-ocean boundaries are generally known as coastlines. Given the definition of the level-set function (equation 3), coastlines are free of ice where $B(\omega,t) = S(\omega,t)$. No coastline exists with $B(\omega,t) > S(\omega,t)$. Only in the marine sectors where $B(\omega,t) < S(\omega,t)$, does a coastline have finite ice thickness and then is replaced by the term grounding line.

## 2.2 Definitions of land, ocean, and ice domains

Given the definition of coastlines and grounding lines, we may define the ocean domain as follows:

$$
\begin{aligned}
\mathcal{O}(\omega,t) &= 1 \quad \text{if } \mathcal{F}(\omega,t) < 0; \\
&= 0 \quad \text{otherwise, except when } \omega \in \mathcal{T}(\mathcal{F}).
\end{aligned}
\tag{5}
$$

The land domain is simply given by $\mathcal{L}(\omega,t) = 1 - \mathcal{O}(\omega,t)$. Surface areas of these complementary domains together make up the total area of the planetary surface, a necessary condition for mass conservation in the Earth System. Note that neither $\mathcal{O}(\omega,t)$ nor $\mathcal{L}(\omega,t)$ is defined at the coastlines or grounding lines. These interfaces rather form their own level set, $\mathcal{T}(\mathcal{F})$, as defined in Section 2.1. For practical purposes, however, one may carry all these masks and level sets as a single field. For example, the Ice-sheet and Sea-level System Model (ISSM; https://issm.jpl.nasa.gov/) uses the field: `md.mask.ocean_levelset`, which takes $-1$ in the ocean, $1$ in land, and $0$ at the coastlines and grounding lines.

We define $\mathcal{I}(\omega,t)$ to be a globally distributed system of ice domains, such that

$$
\begin{aligned}
\mathcal{I}(\omega,t) &= 1 \quad \text{if } H(\omega,t) > 0; \\
&= 0 \quad \text{otherwise.}
\end{aligned}
\tag{6}
$$

For many applications, it may be useful to decompose $\mathcal{I}(\omega,t)$ into a number of sub-domains: $\mathcal{I}(\omega,t) = \{\mathcal{I}_1, \mathcal{I}_2, ..., \mathcal{I}_i, ...\}$, where $\mathcal{I}_i(\omega,t)$ represents the $i$-th ice domain. Individual ice sheets and glaciers can be thought of individual ice domains. As defined in Section 2.1, the grounding line within a given ice domain is given by the level-set $\mathcal{T}(\mathcal{F})$. Using equations (5) and (6), we may define the grounded ice mask simply as $\mathcal{G}(\omega,t) = \mathcal{I}(\omega,t)\mathcal{L}(\omega,t)$ and the floating ice mask as $\mathcal{I}(\omega,t)\mathcal{O}(\omega,t)$.

Equation (6) is a simple, and perhaps the most generic, definition for ice domains, which can accommodate any geometric features such as kilometer-scale pinning points or rugged fjords that can modulate marine ice-sheet instability on retrograde slopes (e.g., *Matsuoka et al.*, 2015; *Whitehouse et al.*, 2017). The employed definition of floating ice mask, however, limits us from capturing the floating ice on subglacial and proglacial lakes that are not part of the global ocean (Figure 1). We believe that the localized processes of ice-lake interactions are of secondary importance, at least, for the purpose of capturing large-scale interplay between the continental ice sheets, solid Earth and sea level in the current Earth System models.

Our definition of coastlines and grounding lines, and hence that of the land and ocean and the grounded and floating ice, facilitates direct evaluation of the interaction between a dynamic system of ice sheets, solid Earth and sea level, as well as the estimation and interpretation of ice-driven global and regional sea-level change by conserving mass in the Earth system. Although a distributed system of ice domains is an integral part of the Earth System, in the following we consider, for brevity, a single domain as an ice sheet, while other ice domains are collectively referred to as far-field ice.

## 3 Sea-level contribution from an ice sheet

The estimation of the sea-level contribution from an evolving ice sheet, featuring marine-based grounded and floating ice, is not trivial, particularly in light of evolving bedrock and MSL. Here we review the common method and its limitations, and present a new method that is applied to arbitrary ice geometries, all kinds of bedrock and MSL forcings, and at all time scales.

### 3.1 Change in ice-height above floatation

We use the bedrock and MSL to define a floatation height for ice

$$H_0(\omega,t) = \frac{\rho_o}{\rho_i} \max\left[\left\{S(\omega,t) - B(\omega,t)\right\}, 0\right], \tag{7}$$

such that the ice thickness in excess of $H_0(\omega,t)$ represents the so-called height above floatation (HAF). We may interpret HAF as the fraction of ice thickness that can potentially contribute to sea level by changing the mass of the oceanic water, and is therefore only defined in the grounded ice domain. Mathematically,

$$H_F(\omega,t) = \mathcal{G}(\omega,t)\left[H(\omega,t) - H_0(\omega,t)\right]. \tag{8}$$

The physical interpretation of HAF is illustrated in Figure 2a. It is clear from the above equations and the figure that $H_F(\omega,t) = H(\omega,t)$ for the grounded ice sheet that rests on the bedrock whose elevation is at or above MSL. For grounded portions of the marine ice sheet, $H_F(\omega,t) < H(\omega,t)$. In fact, $H_F(\omega,t)$ can take negative values (see Sector D in Figure 2a). Such a region, when physically connected to the ocean by oceanic water, can take up water and contribute to sea-level fall.

The evolving ice-sheet geometry is usually described in terms of ice thickness and ice-sheet margins. Indeed, prognostic simulations of ice-sheet models track the transport of mass in terms of equivalent ice-thickness distribution. The transport of mass within the ice domain and ice-ocean mass exchange induce a GRD response of the solid Earth, which further redistributes the mass in the Earth System. This modulates the bedrock topography as well as MSL. These evolving fields may also have components that are forced by external processes such as contemporaneous melting of far-field ice, or GIA, or tectonics. We may describe the evolving ice-sheet geometry in terms of ice thickness, bedrock elevation, and MSL (see Figure 2b-e). We denote $\Delta H(\omega, \Delta t)$, $\Delta B(\omega, \Delta t)$, and $\Delta S(\omega, \Delta t)$ to be the change in respective fields over the time interval $\Delta t$. For the new ice-sheet geometry at time $t + \Delta t$, equation (8) gives $H_F(w, t + \Delta t) = \mathcal{G}(\omega, t + \Delta t)\Big[H(\omega, t) + \Delta H(\omega, \Delta t) - H_0(\omega, t + \Delta t)\Big]$, where $H_0(\omega, t + \Delta t) = \rho_o/\rho_i \, \max\Big[\big\{S(\omega, t) + \Delta S(\omega, \Delta t) - B(\omega, t) - \Delta B(\omega, \Delta t)\big\}, 0\Big]$ is given by equation (7). Similarly, we may rewrite equation (1) for $F(\omega, t + \Delta t)$ and define the new ocean domain $\mathcal{O}(\omega, t + \Delta t)$, land domain $\mathcal{L}(\omega, t + \Delta t)$, and grounded ice domain $\mathcal{G}(\omega, t + \Delta t)$ as described in Section 2.

In what follows, we assume that the net change in grounded ice mass results in the equivalent change in ocean mass, ensuring mass conservation in the Earth System. On the one hand not all of $\Delta H(\omega, \Delta t)$ contributes to change in mass of oceanic water, but on the other hand, in response to the externally forced bedrock and MSL change, the ice sheet may still contribute to change in ocean mass even when $\Delta H(\omega, \Delta t) = 0$ as we count floating ice in the ocean mass (see Appendix A). The stand-alone ice-sheet models evaluate the change in HAF in order to calculate the sea-level contribution of an ice sheet, termed the HAF method for brevity (e.g., *Bindschadler et al.*, 2013; *Nowicki et al.*, 2016):

$$\Delta H_F(\omega, \Delta t) = H_F(\omega, t + \Delta t) - H_F(\omega, t). \tag{9}$$

These models generally (but incorrectly) calculate the equivalent oceanic water volume (rather than the freshwater volume) by spatially integrating $-[\rho_i/\rho_o]\Delta H_F(\omega, \Delta t)$ and divide it by the ocean surface area to estimate the GMSL change. Apart from this water density related error, the HAF method in absence of evolving bedrock and MSL yields the correct estimates of the sea-level contribution (see Appendix A). In fact, the effects of $\Delta B(\omega, t)$ and $\Delta S(\omega, t)$ may be negligible over the timescale of a few decades or shorter. The stand-alone ice-sheet models typically inherit this assumption, even though simulation timescales can be on the order of centuries. Over such relatively longer timescales, this simplistic approach, yields some error, especially in the marine portions of an ice sheet (*Larour et al.*, 2019; *Goelzer et al.*, 2020).

### 3.2 A new field for estimating sea-level contribution

In order to overcome the limitations of the HAF method, we define a unified field, $\Delta H_S(\omega, \Delta t)$, that contributes to sea-level change by modulating both the mass and volume of oceanic water over the period $\Delta t$. This field captures the effects of evolving bedrock and MSL induced by any geophysical processes and is applied to arbitrary ice geometries and at all timescales. We find it convenient to partition $\Delta H_S(\omega, \Delta t)$ upfront into two components:

$$\Delta H_S(\omega, \Delta t) = \Delta H_M(\omega, \Delta t) + \Delta H_V(\omega, \Delta t), \tag{10}$$

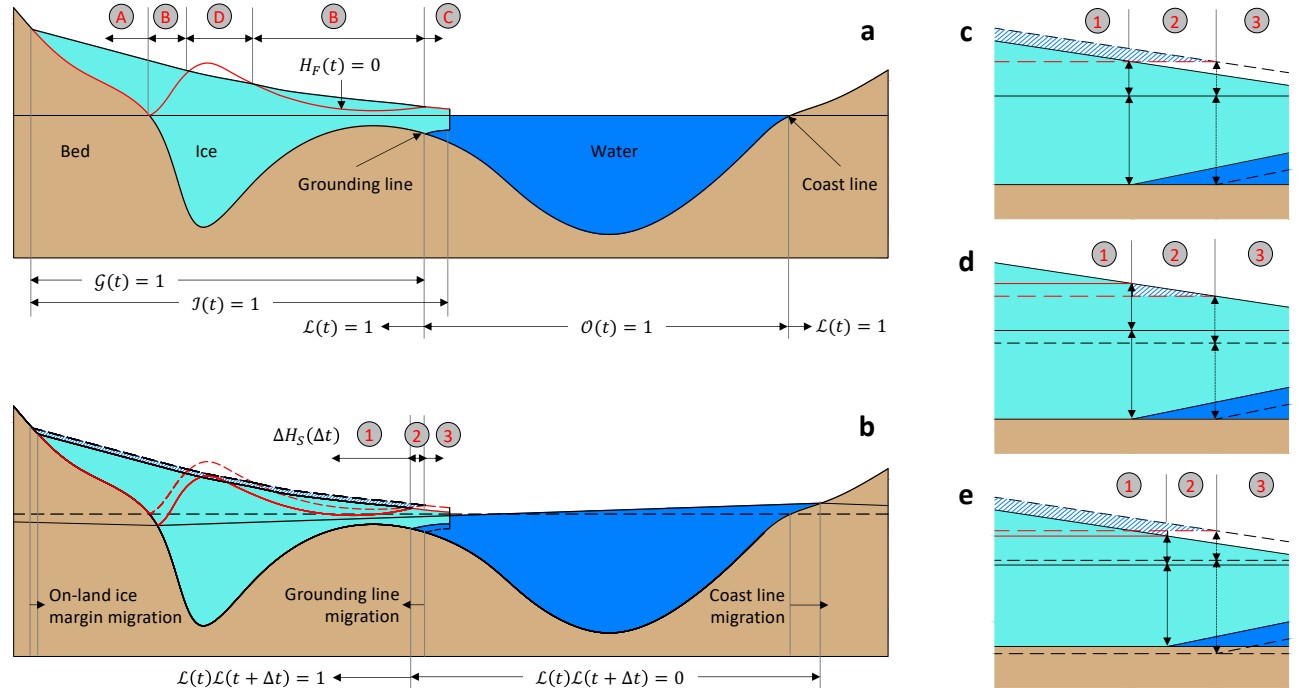

**Figure 2. Conceptual depiction of an evolving ice-sheet geometry. (a)** Domains of ocean $\mathcal{O}$, land $\mathcal{L}$, ice $\mathcal{I}$, and its grounded portion $\mathcal{G}$ at time $t$. The floatation height, having $H_F = 0$, is represented by the red line. Ice-height above floatation satisfies the condition $H_F = H$ in Sector A, $0 < H_F < H$ in Sector B, $H_F = 0$ in Sector C, and $H_F < 0$ in Sector D. **(b)** Ice-sheet geometry at time $t + \Delta t$ after changes in ice thickness and MSL. (For simplicity, bedrock change is not considered.) Old geometry and field variables are shown with dashed lines. Ice thickness that contributes to sea-level change, $\Delta H_S$, is given by $\Delta H$ in Regime 1 and by $\Delta H_F + (1 - \rho_w/\rho_o)(\Delta H - \Delta H_F)$ in Regimes 2 and 3. The hatched area contributes to the ocean mass change (equation 11). Since $\Delta H_F = 0$ in Regime 3, it contributes to sea level by modulating the ocean volume (not mass) alone (equation 12). We zoom in around the grounding line to assess different scenarios: **(c)** when ice thickness changes but the bedrock and MSL do not, typically assumed in stand-alone ice-sheet models; **(d)** when externally-forced MSL changes but ice thickness does not; and **(e)** when ice thickness, bedrock and MSL all evolve simultaneously. Sketches are not to scale.

such that the first component $\Delta H_M(\omega, \Delta t)$ modulates both the mass and volume of the oceanic water, while the second component $\Delta H_V(\omega, \Delta t)$ can only modulate the ocean volume. The following relationships hold for generalized ice geometries, and bedrock and MSL forcings:

$$\Delta H_M(\omega, \Delta t) = \Delta H(\omega, \Delta t)\, \mathcal{L}(\omega, t)\, \mathcal{L}(\omega, t + \Delta t) + \Delta H_F(\omega, \Delta t) \left[ 1 - \mathcal{L}(\omega, t)\, \mathcal{L}(\omega, t + \Delta t) \right], \tag{11}$$

$$\Delta H_V(\omega, \Delta t) = \left( 1 - \frac{\rho_w}{\rho_o} \right) \left[ \Delta H(\omega, \Delta t) - \Delta H_F(\omega, \Delta t) \right] \left[ 1 - \mathcal{L}(\omega, t)\, \mathcal{L}(\omega, t + \Delta t) \right], \tag{12}$$

where $\rho_w$ is the fresh water density. For grounded ice sheets, the mass component makes up about 97% of $\Delta H_S(\omega, \Delta t)$, which loads the solid Earth and induces its GRD response and sea-level adjustment, which will be further discussed in Section 4.

While a detailed interpretation of individual terms appearing in equations (11-12) is given in the Appendix by considering all possible scenarios of evolving ice thickness, bedrock elevation and MSL, Figure 2 illustrates a few representative scenarios. In reference to this figure and equations (11-12), we outline three distinct regimes:

- **Regime 1: Where ice remains grounded at both times $t$ and $t + \Delta t$.**

  All of $\Delta H(\omega, \Delta t)$ in this regime contributes to sea-level change by modulating both the mass and volume of the ocean (first term on the right-side of equation 11), irrespective to the elevation of bedrock upon which the ice is grounded. It turns out $\Delta H(\omega, \Delta t) \neq \Delta H_F(\omega, \Delta t)$ only in the marine portions of the regime and only when evolving bedrock and MSL are considered (see Appendix A1). *Goelzer et al.* (2020) present a method to backtrack $\Delta H(\omega, \Delta t)$ from $\Delta H_F(\omega, \Delta t)$ in such situations, assuming that $\Delta B(\omega, \Delta t)$ and $\Delta S(\omega, \Delta t)$ are known.

  This regime also includes land areas covered by the evolving ice-sheet margins. When ice margin advances over the period $\Delta t$, newly glaciated areas must satisfy $H(\omega, t) = 0$ and $\Delta H(\omega, \Delta t) > 0$. When it retreats, $\Delta H(\omega, \Delta t) = -H(\omega, t)$ must hold in the recently deglaciated areas. In both cases, all of $\Delta H(\omega, \Delta t)$ contributes to the sea-level change.

  Externally-forced $\Delta B(\omega, \Delta t)$ or $\Delta S(\omega, \Delta t)$ does not affect the estimate of $\Delta H_S(\omega, \Delta t)$ in this regime although it may alter bedrock slope or gravitational driving stress and possibly modulate the ice-flow dynamics. While the effects of far-field ice melting and associated ocean loading may be negligible due to their relatively long-wavelength imprints, $\Delta B(\omega, \Delta t)$ due to large earthquakes beneath the ice sheet may have some impact on ice dynamics.

- **Regime 2: Where ice transitions from grounded to floating, or the reverse, over the period $\Delta t$.**

  The sea-level contribution from this regime mainly depends on the change in HAF (second term on the right-side of equation 11), which modulates both the mass and volume of oceanic water. In the absence of externally-forced bedrock and MSL, it follows that $|\Delta H_F(\omega, \Delta t)| < |\Delta H(\omega, \Delta t)|$ (see Appendix A2). The change in ice thickness in excess of the change in HAF (right-side term in equation 12) nominally modulates the volume of the oceanic water. This is due to the difference in volume between the freshwater that would be produced when ice melts and the oceanic water that would be displaced when it floats.

  This is the only regime where, in response to the externally-forced $\Delta B(\omega, \Delta t)$ or $\Delta S(\omega, \Delta t)$, an ice sheet may modulate both the mass and volume of the ocean even when $\Delta H(\omega, \Delta t) = 0$. Specific examples are given in Appendix A2.

- **Regime 3: Where ice remains floating at both times $t$ and $t + \Delta t$.**

  Since $\Delta H_F(\omega, \Delta t) = 0$ holds true in this regime, the change in ice thickness does not modulate the ocean mass itself but it releases or takes up freshwater that has slightly larger volume than the oceanic water upon which it floats. This minor difference in water volume (right-side term in equation 12) contributes to the sea-level change (see Appendix A3). More importantly, the change in ice thickness in this regime can affect the interior-ice sheet dynamics via modulation of buttressing force (e.g., *Gudmundsson et al.*, 2019) and may amplify the future sea-level change.

Given the new field for estimating sea-level contribution (equation 10), we may readily calculate the GMSL change by spatially integrating $-[\rho_i/\rho_w]\Delta H_S(\omega, \Delta t)$, which yields the total freshwater volume being added to the ocean over the period

$\Delta t$, and dividing it by the ocean surface area at time $t + \Delta t$. Assume that an ice sheet collapses instantaneously and that all of the melt water makes it to the ocean. Resulting GMSL change represents the "potential sea level" of the ice sheet at time $t$, and it can be readily derived from equation (10) by setting $\Delta H(\omega, \Delta t) = -H(\omega, t)$ and $\mathcal{G}(\omega, t + \Delta t) = 0$ in the limit of $\Delta t \to 0$. Note that $\Delta H(\omega, \Delta t)$ and $\mathcal{G}(\omega, t + \Delta t)$ are implicit via equations (9, 11-12).

### 3.3 Quantitative comparison of the two methods

Here we present a case study to demonstrate the level of improvements possible by employing the new method (equation 10) over the HAF method (equation 9). For a quantitative comparison, we rely on the recent work of *Larour et al.* (2019), who provide consistent solutions of evolving $H(\omega, t)$, $B(\omega, t)$ and $S(\omega, t)$ for the Antarctic Ice Sheet over the next 500 years. They simulate a high-resolution dynamical ice-flow model (*Larour et al.*, 2012) that is fully coupled with a global solid-Earth deformation and sea-level adjustment model (*Adhikari et al.*, 2016) under the present-day surface climatology and a realistic sub-ice shelf melting scenario (*Seroussi et al.*, 2017). They also account for the effects of far-field ice-mass change on the evolution of bedrock and MSL in Antarctica. To this end, they consider mass balance of the Greenland Ice Sheet and global glaciers, extrapolated into the next 500 years based on the space gravimetry-based measurements. The ongoing change in bedrock and MSL due to the viscous response of the solid Earth to the global deglaciation since the Last Glacial Maximum is also accounted for through an off-line coupling of a GIA model (*Caron et al.*, 2018).

In Figure 3, we compare the two methods both in terms of their spatial and temporal patterns. We show $\Delta H_S(\omega, \Delta t)$ and $\Delta H_F(\omega, \Delta t)$ computed at AD 2350 relative to AD 2000 for Pine Island and Thwaites glaciers (Figures 3a-c). To facilitate the interpretation, we separate the model domain into three regimes as in Figure 2b: Regime 1 (Regime 3) consists of regions that are grounded (floating) at both times; and Regime 2 consists of regions that transition from grounded to floating over the course of simulation. Only $\Delta H(\omega, \Delta t)$ contributes to $\Delta H_S(\omega, \Delta t)$ in Regime 1 (see equation 11). In the marine portions of this regime, $\Delta H(\omega, \Delta t)$ and hence $\Delta H_S(\omega, \Delta t)$ differs from $\Delta H_F(\omega, \Delta t)$ due to the effects of evolving bedrock and MSL in the latter field. The difference between $\Delta H_S(\omega, \Delta t)$ and $\Delta H_F(\omega, \Delta t)$ in Regime 2 and Regime 3 are due to $\Delta H_V(\omega, \Delta t)$ (equation 12), which accounts for the volumetric contribution of ice-thickness change in excess of the change in HAF (see Appendix A2). We also show the time series of the total Antarctic ice-volume change that is attributable to the GMSL change (Figures 3d-e). We find that the new method predicts systematically larger sea-level contribution, compared to the HAF method, throughout the model simulation. The difference in the first 100 years is more than 15% and in the last 100 years is about 8-10%. We isolate the mass (equation 11) and the volume component (equation 12) of the new method to show that the former component alone, which drives the GRD response of the solid Earth, consistently predicts larger sea-level contribution than the HAF method by about 5%. Note that the HAF method usually converts the ice-volume change (Figure 3d) into the equivalent oceanic water (rather than the freshwater) volume change (e.g., *Bindschadler et al.*, 2013) and hence systematically underpredicts the amplitude of GMSL change by additional 2-3%, which is not accounted for in the above analysis.

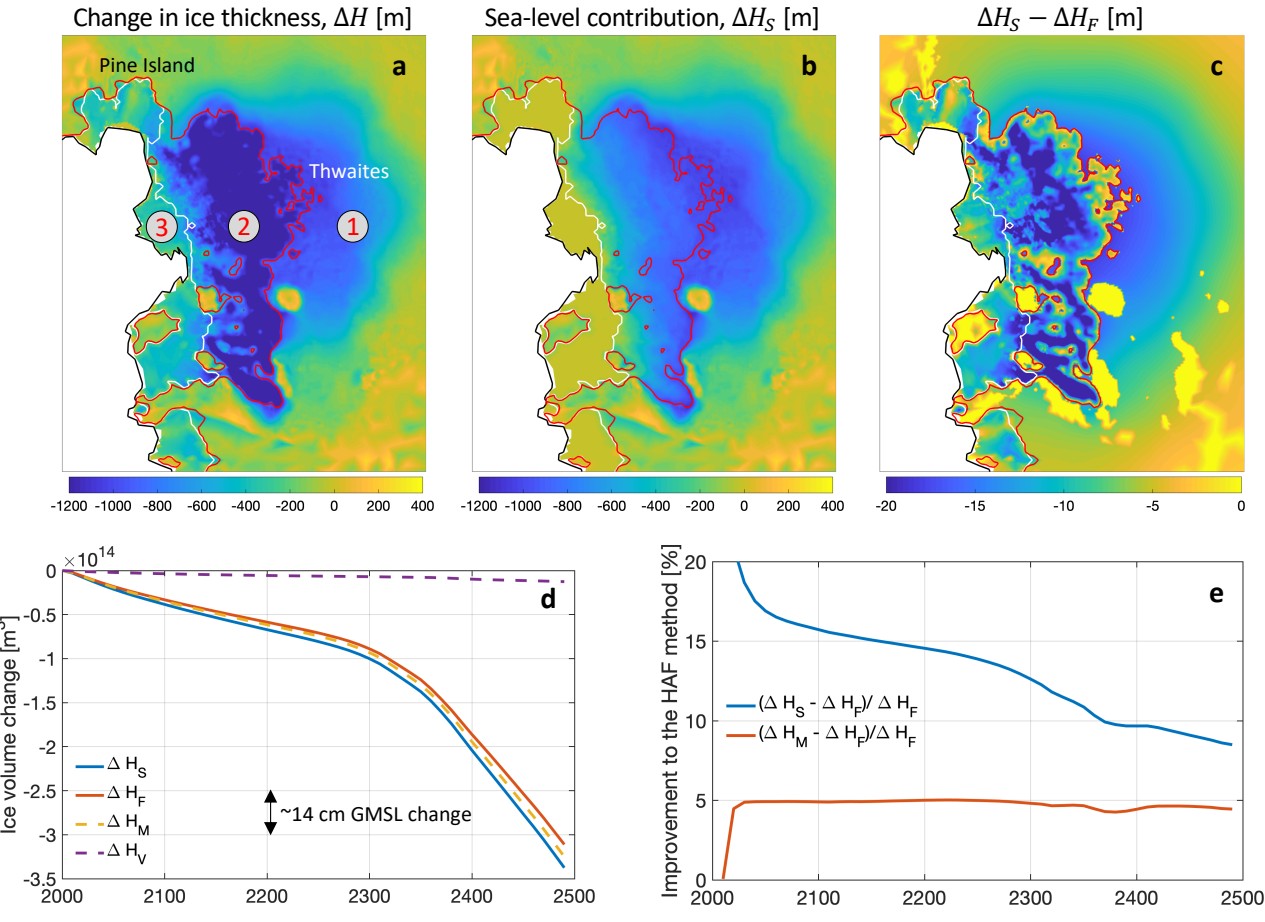

**Figure 3. Example of ice-thickness change and its contribution to the sea level. (a)** Modeled change in ice thickness at AD 2350 for portions of Pine Island and Thwaites glaciers adjacent to the Amundsen Sea (*Larour et al.*, 2019). The black line denotes the ice-ocean interface at AD 2000 and the white (red) line denotes the land-ocean interface i.e., grounding lines, at AD 2000 (AD 2350). These interfaces are used to separate three regimes of the ice sheet as defined in Section 3 (see also Figure 2b). **(b)** Estimation of ice thickness that contributes to the sea level over the next 350 years based on the new method proposed in this study (equation 10). **(c)** Comparison of our method with respect to the HAF method (equation 9). Note that only in portions of Regime 1 where the bedrock elevation is higher than the MSL, the two methods agree (yellow patches). **(d)** The total volume change of the Antarctic Ice Sheet that is attributable to the sea-level change. While $\Delta H_M$ and $\Delta H_F$ modulate both the mass and volume of the ocean, $\Delta H_V$ only modulates the ocean volume. **(d)** Difference between the new method and the HAF method. The latter method underpredicts the sea-level contribution of the ice sheet throughout the model simulation. The mass component of the new method alone consistently predicts 5% more sea-level contribution than the HAF method.

## 4   Sea-level change and mass conservation in the Earth System

Delineation of evolving coastlines (and grounding lines) and estimation of $\Delta H_S(\omega, \Delta t)$ require the knowledge of $\Delta H(\omega, \Delta t)$, $\Delta B(\omega, \Delta t)$, and $\Delta S(\omega, \Delta t)$ along with accurate information of the solid Earth surface (e.g., bedrock topography in ice domains, and land-surface topography and ocean bathymetry in the vicinity of coastlines). Parts of $\Delta B(\omega, \Delta t)$, and $\Delta S(\omega, \Delta t)$ are induced by $\Delta H_S(\omega, \Delta t)$ and associated ocean-mass change themselves. These fields are therefore intertwined with each other, and only by using a mass conserving Earth System model that can capture ice-sheet dynamics, solid-Earth deformation and sea-level adjustment may we find self-consistent solutions. We find it convenient to treat the change in bedrock and MSL collectively in terms of the change in relative sea level (RSL), which by definition is the MSL relative to the sea floor or bedrock (*Gregory et al.*, 2019). Mathematically, $\Delta R(\omega, \Delta t) = \Delta S(\omega, \Delta t) - \Delta B(\omega, \Delta t)$. Since $\Delta R(\omega, \Delta t)$ may be induced by processes other than $\Delta H_S(\omega, \Delta t)$, we must consider them as they impact the estimate of evolving $\mathcal{T}(\mathcal{F})$, and hence the ocean surface area, and $\Delta H_S(\omega, \Delta t)$ itself. In fact, we should also consider the change in steric MSL, which is not accounted for in $\Delta S(\omega, \Delta t)$ (see Section 2.1). The inclusion of this component, however, must be accompanied by the spatially- and temporally-varying ocean density (see, for example, equation 1), which modulates the coastlines, and hence the ocean surface area, but does not affect the buoyant force on the ice, grounding-line migration, and the estimate of $\Delta H_S(\omega, \Delta t)$.

To further diagnose $\Delta R(\omega, \Delta t)$, especially in light of space-based observations and existing GRD models, we present a synopsis of contributing processes as follows:

$$\Delta R(\omega, \Delta t) = \Delta R_C^I(\omega, \Delta t) + \Delta R_C^L(\omega, \Delta t) + \Delta R_P^I(\omega, \Delta t) + \Delta R_P^L(\omega, \Delta t) + \Delta R_O(\omega, \Delta t). \tag{13}$$

The first four terms on the right-side of the equation represent the processes that exchange water between the land and the ocean and contribute to sea-level change by inducing GRD response of the solid Earth (termed, for brevity, the barystatic components). We use the superscript $I$ to refer to the *ice sheet* under consideration and superscript $L$ to other parts of the *land*, including far-field ice and hydrological basins. When these sources of freshwater contribute to sea-level change over a *contemporaneous* period $[t, t + \Delta t]$, corresponding changes in RSL are denoted with the subscript $C$. The land-ocean water exchange may have occurred in the *past* i.e., over the period $(-\infty, t]$, and the induced viscous response of the solid Earth may still contribute to the RSL change over the period $[t, t + \Delta t]$. These components are denoted with the subscript $P$. The last term appearing in the equation captures other non-barystatic processes that may or may not induce GRD response of the solid Earth, but at least modulate the ocean bathymetry or coastal geometry. These processes include earthquakes, landslides, sediment transport and coastal subsidence, amongst others. Assuming that the contemporaneous period $\Delta t$ is on the order of 10 years, we may interpret $\Delta R(\omega, \Delta t)$ as the non-steric component of ongoing RSL change monitored by the satellite gravimetry and altimetry (*WCRP Global Sea Level Budget Group*, 2018). We may interpret $\Delta R_C^I(\omega, \Delta t) + \Delta R_C^L(\omega, \Delta t)$ and $\Delta R_P^I(\omega, \Delta t) + \Delta R_P^L(\omega, \Delta t)$ as ongoing RSL change driven by contemporary global surface mass redistribution (e.g., *Adhikari et al.*, 2019) and by global GIA processes (e.g., *Peltier et al.*, 2015; *Caron et al.*, 2018), respectively.

As defined in equation (10), only part of $\Delta H_S(\omega, \Delta t)$ potentially contributes to the ocean mass change, loads the underlying solid Earth, and induces its GRD response and contributes to sea-level adjustment. Because the GRD effect is applied to the entire column of the ocean water, only $[\rho_i/\rho_o]\Delta H_M(\omega, \Delta t) \approx 0.892 \times \Delta H_M(\omega, \Delta t)$ induces the barystatic component of the

RSL change, which in reference to equation (13) is equivalent to $\Delta R_C^I(\omega, \Delta t)$. The freshwater equivalent of other parts of $\Delta H_S(\omega, \Delta t)$, i.e. the sum of $[\rho_i/\rho_w - \rho_i/\rho_o]\Delta H_M(\omega, \Delta t) \approx 0.025 \times \Delta H_M(\omega, \Delta t)$ and $[\rho_i/\rho_w]\Delta H_V(\omega, \Delta t)$, contributes to the RSL change by modulating oceanic water density, and hence it may be considered as part of the steric MSL. For grounded ice sheets, this component of RSL change is about 97% smaller that the barystatic component. The remaining terms of equation

(13) are what we collectively refer to as the "externally-forced" RSL change. In other words, these are the RSL components not directly induced or contributed by the ice sheet under consideration over the period $\Delta t$.

To solve for the spatial pattern of the steric MSL due to $\Delta H_S(\omega, \Delta t)$, one must consider a dynamic ocean circulation model. Such computations are generally not warranted in the longer-term (decadal or longer timescale) sea-level studies, owing to their smaller amplitudes compared to those of the barystatic component. The spatial pattern of the barystatic RSL due to

$\Delta H_S(\omega, \Delta t)$ can be obtained by solving the so-called "sea-level equation" on a self-gravitating, viscoelastically compressible, rotating Earth (*Farrell and Clark*, 1976; *Milne and Mitrovica*, 1998). To this end, we must consider a mass conserving field that describes the net change in mass per unit area on the solid Earth surface:

$$\Delta M(\omega, \Delta t) = \rho_i \, \Delta H_M(\omega, \Delta t) + \rho_o \, \Delta R_C^I(\omega, \Delta t) \, \mathcal{O}(\omega, t + \Delta t), \tag{14}$$

such that its global integral is zero. Here $\Delta H_M(\omega, \Delta t)$ is given by equation (11) and $\Delta R_C^I(\omega, \Delta t)$ is precisely the same as the

first term on the right-side of equation (13). Because the RSL is defined globally, including in land, we must invoke the ocean mask in the equation. Solving the sea-level equation, in essence, means that we load the solid Earth by the mass-conserving surface load (equation 14) and let its GRD response dictate the self-consistent patterns of RSL, MSL and bedrock changes as well as the new positions of coastlines and grounding lines. The ice-sheet models that account for local or regional isostatic adjustment of bedrock (e.g., *Le Meur and Huybrechts*, 1996; *Bueler et al.*, 2007; *Pattyn*, 2017) do not consider $\Delta R_C^I(\omega, \Delta t)$

as part of the surface load. As a result, these models violate mass conservation in the Earth System and capture incomplete signals of $\Delta B(\omega, \Delta t)$ and $\Delta S(\omega, \Delta t)$ in the estimation of $\Delta H_S(\omega, \Delta t)$.

## 5 Conclusions

We have divided the Earth's surface into complementary domains of land and ocean, which are separated by coastlines. While there may be multiple land domains, we maintain a single global ocean of interconnected oceanic water as in the majority of

studies in physical oceanography and sea level. Distributed bodies of ice intersect the land and the ocean to form glaciers, ice sheets and ice shelves. Grounding lines are defined as the coastlines that belong to the ice domains. The set of generic, and quite simple, mathematical descriptions presented here can handle the complex geometries of both the coastlines and grounding lines, complementary to those of far-field land, ocean and ice domains and their respective overall evolutionary history.

Based on this formalism of evolving coastlines and grounding lines, we present a unified method to calculate the exact

fraction of ice thickness that contributes to the sea-level change over a given period. The method is a function of evolving ice thickness, bedrock elevation and mean sea level driven by any geophysical processes. Along with its obvious application to estimate the global-mean sea-level change, it is absolutely critical to track the global mass transport, and assess the response of

a dynamic system of ice sheets, solid Earth and sea level, while accounting for kilometer-scale features in ice/bedrock/ocean geometries. Our method requires bookkeeping the global land and ocean domains. This is crucial for considering distributed ice and land domains with complex geometries in Earth System models. For the treatment of an individual ice sheet, however, it is sufficient to track a continental or regional land domain, provided that there is an understanding that ocean water may recede

5 from, or reinundate, continental land (e.g., *Johnston*, 1993; *Milne*, 1998). In fact, it may often be possible to use grounded ice masks in place of the land domains. In the most simplified case when the bedrock and mean sea level do not evolve, our method reduces to the common method that is based on the concept of ice-height above floatation. For an example model simulation considered in this study (*Larour et al.*, 2019), we find that the new method systematically yields 10-15% more sea-level contribution from the Antarctic Ice Sheet. We recommend that the ice-sheet modeling community consider the proposed

10 method as a metric to quantify the sea-level contribution of evolving ice sheets. This is especially appropriate for model analysis that is informed by ice and ocean mass monitoring from space assets, such as ocean and ice altimetry, radar interferometry and space gravimetry (e.g., *Bentley and Wahr*, 1998).

## Notation

| | |
|---|---|
| $B$ | Solid Earth surface (i.e., land surface or sea floor), or simply bedrock, elevation |
| $\Delta B$ | Change in bedrock elevation over the period $\Delta t$ |
| $F$ | A function such that $F = 0$ satisfies the hydrostatic equilibrium between ice and the oceanic water |
| $\mathcal{F}$ | A function such that $\mathcal{F} = 0$ represents the grounding lines or coastlines |
| $\mathcal{G}$ | Mask of the grounded portions of an ice sheet |
| $H$ | Ice thickness |
| $\Delta H$ | Change in ice thickness over the period $\Delta t$ |
| $H_0$ | floatation height for ice |
| $H_F$ | Ice height above floatation (HAF) defined for grounnded ice |
| $\Delta H_F$ | Change in HAF over the period $\Delta t$ |
| $\Delta H_M$ | A component of $\Delta H_S$ that modulates both the mass and volume of the ocean over the period $\Delta t$ |
| $\Delta H_S$ | A new field for estimating the sea-level contribution of ice sheets over the period $\Delta t$ |
| $\Delta H_V$ | A component of $\Delta H_S$ that can only modulate the ocean volume over the period $\Delta t$ |
| $\mathcal{I}$ | A globally distributed system of ice domains |
| $\mathcal{L}$ | A globally distributed system of land domains |
| $\Delta M$ | Change in mass per unit area on the solid Earth surface over the period $\Delta t$ |
| $\mathcal{O}$ | The global ocean domain |
| $\Delta R$ | Non-steric component of the relative sea level (RSL) change over the period $\Delta t$ |
| $\Delta R_C^I$ | A component of $\Delta R$ due to the contemporary ice-sheet mass change |
| $\Delta R_C^L$ | A component of $\Delta R$ due to the contemporary change in far-field ice and land water storage |

| $\Delta R_P^I$ | A component of $\Delta R$ due to the past ice-sheet mass change |
|---|---|
| $\Delta R_P^L$ | A component of $\Delta R$ due to the past change in far-field ice and land water storage |
| $\Delta R_O$ | A component of $\Delta R$ due to other non-steric processes |
| $\rho_i$ | Density of ice |
| $\rho_o$ | Mean density of the oceanic water |
| $\rho_w$ | Density of freshwater |
| $S$ | Mean sea level (MSL), excluding its steric component |
| $\Delta S$ | Change in $S$ over the period $\Delta t$ |
| $T$ | The zero-level set of $F$ that satisfies the hydrostatic equilibrium between ice and the oceanic water |
| $\mathcal{T}$ | The zero-level set of $\mathcal{F}$ that represents the coastlines and grounding lines |
| $t$ | time |
| $\Delta t$ | time period |
| $\omega$ | 2-D spatial coordinates, geographic $(\theta, \phi)$ or Cartesian $(x, y)$, on the planetary surface |

*Code and data availability.* The data and code used to produce Figure 3 are available online at https://doi.org/10.7910/DVN/9LUJTD (*Adhikari et al.*, 2020).

## Appendix A: Interpretation of $\Delta H_F(\omega, \Delta t)$ and $\Delta H_S(\omega, \Delta t)$

Here we provide an in-depth comparison between the HAF method (equation 9) and ours (equation 10) in light of evolving ice
5   thickness, bedrock elevation and MSL. The latter two fields may be treated collectively in terms of the relative sea level (RSL) which, by definition, is the MSL relative to the bedrock or sea floor. In the following, we consider all plausible scenarios by combining the change in ice thickness, $\Delta H(\omega, \Delta t)$, and relative sea level, $\Delta R(\omega, \Delta t)$, over the period $\Delta t$.

### A1   Where ice remains grounded at both times $t$ and $t + \Delta t$

In our method, all of $\Delta H(\omega, \Delta t)$ irrespective to $\Delta R(\omega, \Delta t)$ contributes to sea-level change by modulating both the mass and
10   volume of the oceanic water (see the first term on the right-side of equation 11). The same is true for the HAF method as long as ice remains grounded on the bedrock whose elevation is at or above MSL at both times $t$ and $t + \Delta t$, in which case $\Delta H_F(\omega, \Delta t) = \Delta H(\omega, \Delta t)$. There is nonetheless a minor density-related difference between the two methods: our method evaluates the freshwater equivalent height $[\rho_i/\rho_w]\Delta H(\omega, \Delta t)$ whereas the HAF method generally evaluates the oceanic-water equivalent height $[\rho_i/\rho_o]\Delta H(\omega, \Delta t)$. As a result, the HAF method systematically underestimates the amplitude of the global-
15   mean sea-level (GMSL) change by about 2 to 3%.

If the ice is grounded on the marine bedrock (whose elevation is below MSL) at least at time $t$ or $t + \Delta t$, the HAF method generally yields incorrect solution in addition to the density-related error noted above. In this case, $\Delta H_F(\omega, \Delta t) \neq \Delta H(\omega, \Delta t)$

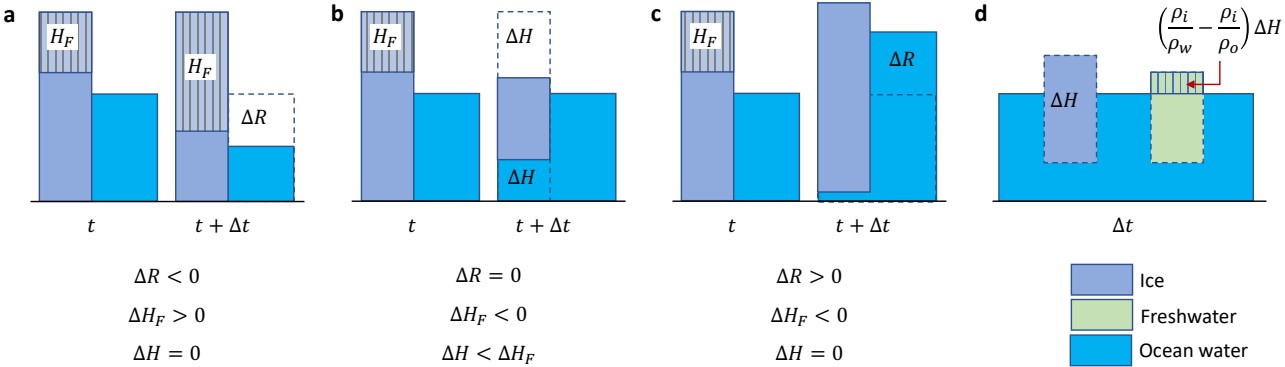

**Figure A1. Scenarios of ice-thickness and RSL change and sea-level contributions. (a)** Since the column of ice remains grounded at both times and its thickness does not change over the period, the ice column does not contribute to GMSL change. The HAF method incorrectly predicts GMSL drop because of a positive value of $\Delta H_F$ in response to the imposed drop in RSL. **(b)** For the grounded ice to float in the case of fixed RSL, it must thin sufficiently, such that $\Delta H < \Delta H_F < 0$ holds true, contributing to the GMSL rise. **(c)** Significant rise in RSL may cause the grounded ice to float even if its thickness does not change. As a result, the column of ice contributes to the GMSL rise. **(d)** Melting of the floating ice produces freshwater that occupies slightly larger volume than that of the ocean water that was replaced by the ice. This excess volume, the hatched portion of the freshwater column, causes the GMSL to rise. Sketches are not to scale.

generally holds true, and depending upon the relative amplitudes and signs of $\Delta H(\omega, \Delta t)$ and $\Delta R(\omega, \Delta t)$, the HAF method may over- or under-predict the sea-level contribution compared to our method. Two special cases are worth mentioning:

– Case A1.1: $\Delta R(\omega, \Delta t) = 0$. When the effect of evolving RSL is not considered, we find that $\Delta H_F(\omega, \Delta t) = \Delta H(\omega, \Delta t)$ and, consequently, the two methods are equivalent.

5    – Case A1.2: $\Delta H(\omega, \Delta t) = 0$. When the thickness of grounded ice does not change, the HAF method may incorrectly predict non-zero sea-level contribution in cases when $\Delta R(\omega, \Delta t) \neq 0$ (see Figure A1a). In this case, the HAF method systematically overestimates (underestimates) GMSL change when $\Delta R(\omega, \Delta t)$ is greater (less) than zero.

## A2    Where ice transitions from grounded to floating, or the reverse, over the period $\Delta t$

Here the working principle of both methods is same and as follows. We derive the potential sea level (PSL) contributions of
10   ice thicknesses at time $t$ and $t + \Delta t$. We then compute their difference to derive the actual sea level (ASL) contribution over the period $\Delta t$. We may define PSL and ASL in terms of freshwater equivalent height as follows (see Figure A1):

$$\text{PSL}(t) = \frac{\rho_i}{\rho_w} H_F(t) + \left( \frac{\rho_i}{\rho_w} - \frac{\rho_i}{\rho_o} \right) \left[ H(t) - H_F(t) \right], \tag{A1}$$

$$\text{PSL}(t + \Delta t) = \frac{\rho_i}{\rho_w} H_F(t + \Delta t) + \left( \frac{\rho_i}{\rho_w} - \frac{\rho_i}{\rho_o} \right) \left[ H(t) + \Delta H(\Delta t) - H_F(t + \Delta t) \right], \tag{A2}$$

$$\text{ASL}(\Delta t) = \frac{\rho_i}{\rho_w} \Delta H_F(\Delta t) + \left( \frac{\rho_i}{\rho_w} - \frac{\rho_i}{\rho_o} \right) \left[ \Delta H(\Delta t) - \Delta H_F(\Delta t) \right]. \tag{A3}$$

Ice equivalent height of the terms appearing on the right-side of equation (A3) are reported in the main text: the second term on the right-side of equation (11) and the term on the right-side of equation (12), respectively. Since the HAF method deals with the oceanic water density, rather than the freshwater density, both PSL and ASL in this method can be deduced from the above equations by replacing $\rho_i/\rho_w$ by $\rho_i/\rho_o$. The second terms in these equations vanish, and the ASL is given in terms of oceanic water equivalent height by $[\rho_i/\rho_o]\Delta H_F(\Delta t)$ whose ice equivalent height is reported in the main text (equation 9).

When ice transitions from grounded to floating, $H_F(t+\Delta t) = 0$ and the first term appearing in equation (A3) always takes a negative value. Three distinct scenarios of evolving ice thickness and RSL may be of interest:

– Case A2.1: $\Delta R(\omega, \Delta t) \leq 0$ and $\Delta H(\omega, \Delta t) < 0$. In this case, the only condition for the grounded ice to float is through its sufficient thinning such that $\Delta H(\omega, \Delta t) < \Delta H_F(\omega, \Delta t) < 0$ (see Figure A1b). Both terms appearing in equation (A3) take negative values, causing the GMSL to rise.

– Case A2.2: $\Delta R(\omega, \Delta t) > 0$ and $\Delta H(\omega, \Delta t) = 0$. Here the condition $\Delta H_F(\omega, \Delta t) < \Delta H(\omega, \Delta t) = 0$ holds true (see Figure A1c). Since the first term appearing in equation (A3) is about 97% larger in magnitude than the second term (which takes a positive value), it causes the GMSL to rise. In other words, the externally forced RSL rise causes the ice to contribute to GMSL rise even though its thickness does not change.

– Case A2.3: $\Delta R(\omega, \Delta t) > 0$ and $\Delta H(\omega, \Delta t) \neq 0$. Only when $\Delta H(\omega, \Delta t) > 0$ and its amplitude is significantly larger (by about a factor of 35) than that of $\Delta H_F(\omega, \Delta t)$, the second term appearing in equation (A3) that takes a positive value dominates and causes the GMSL to fall. Otherwise, the GMSL rises even when ice thickens over the period $\Delta t$.

When ice transitions from floating to grounded, $H_F(t) = 0$ and the first term appearing in equation (A3) always takes a positive value. Three distinct scenarios of evolving ice thickness and RSL may be of interest:

– Case A2.4: $\Delta R(\omega, \Delta t) \geq 0$ and $\Delta H(\omega, \Delta t) > 0$. In this case, the only condition for the floating ice to be grounded is through its sufficient thickening such that $0 < \Delta H_F(\omega, \Delta t) < \Delta H(\omega, \Delta t)$. Both terms appearing in equation (A3) take positive values, causing the GMSL to fall.

– Case A2.5: $\Delta R(\omega, \Delta t) < 0$ and $\Delta H(\omega, \Delta t) = 0$. Here the condition that $\Delta H(\omega, \Delta t) = 0 < \Delta H_F(\omega, \Delta t)$ holds true. Since the first term appearing in equation (A3) is about 97% larger in magnitude than the second term (which takes a negative value), it causes the GMSL to fall. In other words, the externally forced RSL drop causes the ice to further contribute to GMSL drop even though its thickness does not change.

– Case A2.6: $\Delta R(\omega, \Delta t) < 0$ and $\Delta H(\omega, \Delta t) \neq 0$. Only when $\Delta H(\omega, \Delta t) < 0$ and its amplitude is significantly larger (by about a factor of 35) than that of $\Delta H_F(\omega, \Delta t)$, the second term appearing in equation (A3) that takes a negative value dominates and causes the GMSL to rise. Otherwise, the GMSL falls even when ice thins over the period $\Delta t$.

## A3 Where ice remains floating at both times $t$ and $t + \Delta t$

We may evaluate PSL and ASL contributions from this region based on equations (A1)-(A3). Since $H_F = 0$ at both times $t$ and $t + \Delta t$, the evaluation of the sea-level contribution does not depend on the evolving RSL. In this scenario, the ASL is

given in terms of freshwater equivalent height by $[\rho_i/\rho_w - \rho_i/\rho_o]\Delta H(\omega, \Delta t)$ whose ice equivalent height can be deduced from equation (12) by setting $\Delta H_F(\omega, \Delta t) = 0$ (see Figure A1d). When the ice thins (thickens), it causes the GMSL to rise (fall) by modulating the volume (not mass) of the oceanic water. The ASL contribution in the HAF method can be deduced by replacing $\rho_i/\rho_w$ by $\rho_i/\rho_o$ and is effectively zero and, therefore, does not appear explicitly in equation (9). This suggests that the HAF method systematically underestimates the amplitude of GMSL change.

*Author contributions.* SA conceived and conducted the research. All authors contributed to the theory. ERI helped write the first draft of the manuscript. EL helped produce Figure 3. All authors contributed to the writing and editing of the manuscript.

*Competing interests.* The authors declare that they have no competing interests.

*Acknowledgements.* This research was carried out at the Jet Propulsion Laboratory (JPL), California Institute of Technology, under a contract with National Aeronautics and Space Administration (NASA), and was funded through the JPL Research, Technology & Development programs (grant no. 01STCR-R.17.235.118; 2017-2019 and 01STCR-R.19.021.241; 2019), and through the NASA Sea-Level Change Team (grant no. 16-SLCT16-0015; 2018-2020) and the NASA Earth Surface and Interior program (grant no. 19-ESI19-0020; 2020-2022). We acknowledge three anonymous reviewers for their thorough and constructive reviews that improved the manuscript.

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
