# Peer review of "A kinematic formalism for tracking ice-ocean mass exchange on the Earth's surface and estimating sea-level change"

_The Cryosphere, 2020_

## Referee Comment (RC1) · Anonymous Referee #1 · 9 Mar 2020

\* Summary

The paper presents a formalism to geometrically interpret changes in ice sheets and underlying bedrock and their combined effect on the ocean and sea level. The approach defines two distinct domains (land and ocean) that can both intersect areas of ice cover. It then traces ice and bedrock changes and transitions between different domains to determine the sea-level contribution of the ice sheet.

\* General comments

The paper is well written, clearly structured and deals with the important question of how to calculate the sea-level contribution of a marine ice sheet, among others. I believe it would make an interesting contribution to The Cryosphere given that the

points raised below are addressed adequately.

One of the main conclusions of this paper reads very similar to the one in Goelzer et al. (2020), both papers proposing an alternative to conventionally calculating the sea-level contribution of marine ice sheets based on volume above flotation. it seems important to clarify what the similarities and differences are. The results presented by Goelzer et al. (2020) imply that ice and bedrock changes have to be considered together at least in any place where ice could ground over the course of the experiment. This is the direct consequence of the claim that the sea-level contribution calculated from one point in time to another should be independent of what happens in between. Since the example that is put forward (see their Fig 2a and related text) matches with regime 3 here, there seems to be a direct disagreement between the two approaches: bedrock changes are taken into account in their case, but not here. This may point to a flaw in the approach that should be clarified and discussed. If bedrock changes are not considered as part of the ice sheet change in regime 3, what other component is taking it into account (if any) and how does the domain separation between those components work? If both approaches are not compatible, why and under what circumstances do they differ? What would needs to be changed to make the two approaches compatible? Are the two approaches addressing a different modelling framework, which explains the differences?

While the first few words in the abstract seems to say that this paper is about representing ice sheets in models, the mention of geodesy later may suggest that observations of ice sheets are equally addressed. If the aim is indeed modelling, I think this should be made clear and a clearer distinction be made from observations. If both should be addressed at the same time, I suggest to make sure that the presented formulations make sense in both realms.

While I appreciate the formal description of the case, I miss better guidance of the reader through what is a difficult problem to understand and visualise. Recurrent re-definitions of variables (see example S(w,t) below) should be avoided, individual terms

[Figure]

in the equations should be better explained and examples should be given where possible. This particularly applies to cases where the formalism uses familiar concepts and applies them to something else (e.g. floatation condition for ice applied to define the ice free coastline).

For understanding and reproducing the results it would be useful to provide access to the data and tools used to produce the results and plots in Figure 3. Please consider making the geometry and scripts available.

*Specific comments

P1.l1 Not all ESMs include ice sheet components. Reformulate.

P1.l2 The connection between ESMs and geodesy is not clear to me. E.g. observation don't exist for ESM paleo simulation where the formalism should also hold.

P1.l5 "grounded and floating masks" suggests a modelling perspective, but "as viewed from space" relates to an observational dataset. What is the perspective of this paper?

P1.l5 "Here we present ...". The subject in this sentence is not clear. Reformulate.

P1.L13-15 This is clearly true for simulations of the Antarctic ice sheet, but not really for the Greenland ice sheet, which is dominated by surface mass balance changes. Similar cases may also exist during other climates with ice sheets that were mainly land based. Suggest to reformulate.

P2.l2 "Defining geometry". Do you mean "Defining the bedrock"? How could the geometry be defined upfront for an intercomparison when the models are supposed to produce an evolving geometry?

P2.l5 What do you mean with "basic configuration setup" and "Similar setups"? Could you describe this in more detail?

P2.l12 Why are "floating ice shelves" and "retrograde bedrock slopes" complex features? They occur in the very simple flowline model setups you may be referring to

above. Clarify.

P2.l13 What is the "traditional theory for ice-bedrock-ocean interface changes"? Clarification needed.

p2.l23 "setup" -> maybe "interpretation"?

p2.l24 What is the difference between "glaciers" and "ice sheets" in your description?Clarify if the two terms are interchangeable or distinct. If the latter, what sets them apart in your formalism?

p2.l24 I found the upfront separation between land and ocean confusing for your context, because it is not intuitive where that separation is to be made for a marine-based ice sheets. The definition what is to be considered land and ocean comes too late. I suggest to make that clearer much earlier.

P2.l25 Same for S(w,t), defined here first as the sea-surface elevation. How should we think of S for an ice shelf? Why not start with defining S as the geoid as you do later.

P2.l26 I admit, I had to look up what the ITRF is. For other readers not familiar with it, you may want to add a sentence or two to say what the ITRF provides. In practice, if I use Bedmachine data, is it registered on the ITRF or are you suggesting this is something the user would have to take care of herself?

P2.l31 So far S(w,t) is defined as sea-level. As such, any case B > S is not well defined. The interpretation of S as the geoid must come earlier for this to make sense.

P2.l32 You say here that S(w,t) includes high-frequency noise and variability, but on the next page you want S(w.t) to refer only to the quasi-static component of the sea-surface. Why not introduce S directly as the quasi-static component of the geoid, rather than going through three redefinitions along the way (sea surface -> quasi-static sea-surface -> sea-level -> geoid).

P3.l1 Remove "changing" before interactions?

p3.l7 What is the "interior of marine ice sheets"? Clarify

P3.l9 Say and explain what F(w,t) means. Traditionally it determines if the ice is floating. But you seem to extend it to locations with H = 0? Maybe it would be worth it to mention that.

p3.l13 Please define what "open ocean" means and what "contact with the open ocean" means. This definition comes too late in the manuscript. Does w have to be part of O? Maybe we need the definition of O already before this part not on page 4?

p3.l14 A more obvious definition of a generalised coastline for me, that also exists in presence of a marine-based ice sheet would be the point where the bedrock and the geoid intersect (1|2 in Fig2). That doesn't help for your formalism, but it goes to show that it is not immediately obvious to think the coastline at he grounding line. Better guidance needed.

P3.l8-15 I found this paragraph difficult to follow. You start with reference to Fig. 2a, where our focus is on the left hand side and with F(w,t) which suggests it is about ice. But then you describe R(w,t) and the coastline, which are difficult to visualise in a place with an ice shelf. It may help to guide the reader by being explicit about the two 'generalisations' that take place here: floatation criterion (for ice) –> definition of the coastline (everywhere). grounding line (for ice) –> coast line (everywhere).

P4.l1 I don't see why there could not be a grid point in a model with B=S and H>0. A glacier terminating on land or on a sill exactly at sea-level? Please clarify.

P4.l6 The fact that neither O nor L are defined at the grounding line seems problematic. How can your formalism be mass conserving when grounding line grid points in an ice sheet model are not part of these masks? How do you track the grid cells that fulfil this condition, do they form a separate category? Why would it not matter to consider them?

P4.l9 Remove "deep" and "well". I suppose the condition could also be true for shallow

troughs with bathymetry moderately below sea-level.

P4.l10 While I understand the use of this connectivity concept in your formalism, I find it problematic in practice. It means that small changes in ice or bedrock can lead to very large changes in O and L. In an unfavourable configuration, the short term grounding and ungrounding of a critical point could e.g. switch an entire system of connected fjords on and off.

p4.l18 With the above, combining the grounded and floating ice masks leaves a hole at the grounding line. Is this desired?

P5.l13 The geoid typically changes first, then the bedrock. Maybe re-order in the sentence.

p5.l22-23 I am confused about this sentence. Isn't "quantifying the fraction of ice mass change that contributes to sea level" exactly what you are doing by defining $dH_{S}$ below? Reformulate?

p5.23 "the assumption" appears three sentence back, maybe refer to it more specifically.

p5.23 Remove "all the time".

p5.26 "As we show below". This has been shown before by others (see references). Reformulate to avoid confusion.

p5.l30 Can you please explain what the three terms mean physically. E.g. the first term accounts for thickness changes of ice that is and remains grounded ...

p5.l30 Could you explain why H0 appears as an *absolute* contribution in the third term compared to considering *changes* in $H_{S}$ and $H_{F}$ in term one and two?

p6.l6 Not clear what "holding in the areas of on-land ice margin migration" means. Reformulate.
p7.l15 I suppose you mean that the nonzero $\Delta H_{F}$ is compensated by other terms in Eq. 7. Which ones? This is important to understand.

p7.l22-23 This sentence may be confusing because Goelzer et al. (2020) consider not only transitions between grounded and floating ice, but all three regimes. Reformulate.

p7.l26 Important to note that in regime 3 bedrock changes can contribute to sea-level change, even if you are not considering them as part of the ice sheet change. Also important to realise that ice floating at t and t+$\Delta$t may have been grounded at some point in between. I think this makes your solution dependent on the time stepping. See Fig 2a in Goelzer et al. (2020) for an example of such a case.

p8.l7 The change in $\Delta H_{S}$ and $\Delta RCI(\Delta t)$ is not only due to ice mass changes, but also due to bedrock changes under the ice. It may be good to mention that here.

p8.l8 For clarity, could you mention why potential changes in ocean area from O(w,t) to O(t+$\Delta$t) do not matter for $\Delta RCI(\Delta t)$ in Eq 8? I suppose the underlying assumption is that we should be interested in sea level at time t+$\Delta$t?

p8.l12 G(w,t) is not explicit in Eq. 7 only implicitly by evoking Eq. 6. This could be mentioned.

p8.l14. Maybe remind us what R(w,t) is here, as it is only introduced inline and back at p3.

p8.l15 To make this equation more digestible, maybe start with combined symbols. E.g. by combining all the barystatic components like you do in the text.

p9.l1-24 My experience with the paragraph including Eq. 9 is that a lot of new concepts are suddenly thrown in here without much preparation. Especially the idea to separate the effect of the past from the contemporary would profit from some more introduction. Maybe a new section with a few introductory sentences could be started p8.l6 to prepare the ground for this discussion.

p9.l3 Since $\Delta RV$ ($\omega$,$\Delta t$) may be the component that takes into account bedrock changes in regime 3 and is complementary to the bedrock changes happening under the ice, I would be interested to see how it is calculated and how the masking works in that case.

p9.l28 By your definition, the grounding lines are neither part of the grounded ice nor part of the floating ice domain. What is this category called and how are you accounting for it?

p10.l4-5 How does your approach compare to/ differ from that proposed by Goelzer et al. (2020).

Figures

Figure 1. It seems confusing to introduce lakes, subglacial lakes and proglacial lakes and than not consider them at all. It would make the figure much clearer to remove them.

Figure 2. The geoid (and S) is also defined over land. Please add in both panels.

Figure 3. Please include a panel with $\Delta H\_\{F\}$. This is important as it is discussed as the conventional method and appears in the difference in panel c. Please also add contour lines to delineate the regions 1-3. Mention that regime 4 does not exist in this region if that is true. Otherwise, delineate regime 4. If different from results in Goelzer et al (2020) it would also be interesting to add a comparison as figure here.

References Goelzer, H., Coulon, V., Pattyn, F., de Boer, B., and van de Wal, R.: Brief communication: On calculating the sea-level contribution in marine ice-sheet models , The Cryosphere, 14, 833–840, https://doi.org/10.5194/tc-14-833-2020, 2020.

---

## Referee Comment (RC2) · Anonymous Referee #2 · 14 Mar 2020

Summary

This paper presents a framework for describing the geometry of an evolving ice sheet margin in Earth system models, in which the geometry of the bedrock and mean sea level also can be dynamic. The authors define relevant terms and give a mathematical description of the way different quantities are related. In particular, they quantify the portion of ice thickness change that contributes to changes in ocean mass and global mean sea level. As Earth system modelers work to integrate dynamic ice sheet models with solid-Earth and sea-level models, this paper will be a useful reference.

In general, the paper is well written, and the figures are very helpful. Sometimes, however, technical terms related to sea level are used without precision, or are introduced without giving enough background information. Some of the equations contain terms

that are not clearly defined or explained physically. In the comments below, I suggest where the exposition could be improved, especially for readers approaching these concepts for the first time.

Also, the text refers to "traditional" or "customary" approaches of estimating sea-level contributions from ice sheets based on the change in height above flotation, in contrast to the approach described here. It would be helpful to see some specific examples of customary approaches from published papers, with estimates of the magnitude and sign of the associated errors. This would enable readers to better assess the value of the proposed formalism.

Major comments

p. 1, Title: The title includes the words "mass conserving", but in the text I did not find an operational definition of what this means in an ESM with evolving ice sheets. Please provide such a definition, and perhaps an example of how mass conservation would be violated.

Also, the title implies that there will be a detailed analysis of ice-sheet interactions with solid Earth and sea level, but the actual scope seems narrower: to accurately compute the contribution of dynamic ice sheets to barystatic sea level rise (i.e., the sea-level component associated with a redistribution of mass between land-based ice and the ocean) in ESMs. I suggest a revised title that better reflects the scope.

p. 1, Abstract: The abstract is very general and does not provide a clear sense of the scope of the paper. If a central goal is to describe how to accurately compute the ice sheet contribution to barystatic sea level rise, then this goal should be clearly stated.

p. 1, l. 1: Although I am fully in favor of including dynamic ice sheets in ESMs, I would not go so far as to suggest that "any Earth System model" already includes them. For some ESMs, ice sheets are still on the back burner.

p. 2, l. 2: "Defining geometry". I am not sure what this means–something like defining

geometric concepts that are relevant to models with dynamic ice sheets?

p. 2, l. 4: "future debate and reconciliation." Ideally, the concepts are set forth clearly in a way that leads to greater mutual understanding and less debate.

p. 2, l. 5: "basic configuration setup". I am not sure what this means. It seems an odd way to describe what seem to be theoretical or analytical frameworks.

p. 2, l. 8: I am not clear why these previous analyses are referred to as "traditional configurations." The word "traditional" suggests a contrast with something novel and untraditional to be introduced here. However, I don't see this work as heading off in a different direction from the cited papers, but rather as clarifying concepts that are particularly relevant for ESMs. As above, "configuration" doesn't seem to be the right word.

p. 2, l. 13: Similarly, what is meant by "traditional theory for ice-bedrock-ocean interface changes"?

p. 2, l. 25: "sea surface elevation". This is an ambiguous term; it can refer either to the mean (on some appropriate time scale) or to a quantity that varies on short time and spatial scales. There is some explanation below, but it is better to be as clear as possible when introducing the quantity S(omega,t). I think that what is meant here is what Gregory et al. (2019) call "mean sea level", a term they recommend in place of the deprecated term "mean sea surface." If S is actually meant to represent the geoid, which is not quite the same as mean sea level, then this should be stated clearly.

In general, Gregory et al. (2019) is a comprehensive, carefully written reference. I suggest that the authors adopt similar terminology, paraphrasing and referring to that paper as appropriate.

p. 2, l. 26: What is the International Terrestrial Reference Frame, and how is it defined? Is it similar to what Gregory et al. (2019) call the reference ellipsoid?

p. 2, l. 32: It is stated first that S is highly variable in space and time, and then it is stated

that S is quasi-static and does not, in fact, include short-term dynamic processes. Please use S only to refer to quasi-static mean sea level, and use a different term when discussing short-term dynamics.

p. 3, l. 5: "Sea level" is another ambiguous term, as discussed by Gregory et al. I suggest "mean sea level."

p. 3, l. 5: "represents an equipotential surface whose spatial pattern mimics the geoid." This is confusing. First, how is the geoid defined? Gregory et al. define it as the geopotential surface chosen so that the volume between the geoid and the sea floor is equal to the time-mean volume of sea water (including the liquid-water equivalent of floating ice) in the ocean. Second, what is meant by "mimics" the geoid? Does "mimics" mean "is equivalent to", or "is similar to"? If the latter, in what way does S(omega,t) differ? If S is mean sea level, then it is not an equipotential surface; for instance, mean sea level has a higher geopotential on one side of the Gulf Stream or ACC than the other. (Though it could be convenient to define S as an equipotential surface in areas not covered by ocean.)

p. 3, l. 11: Since ocean and ice have variable density, it would be clearer to refer to rho_o and rho_i as reference densities.

p. 3, l. 12: "sea surface relative to the seafloor". I suggest "local mean sea level relative to the seafloor".

p. 4, Eq. (3): Why are the ocean and land functions undefined at coastlines and grounding lines? Is it problematic not to include them in one domain or the other?

p. 4, l. 7: Please say precisely what is meant by "connected to the open ocean". I would guess that the connected regions include marginal seas (e.g., the Mediterranean) but not inland lakes (e.g., the Great Lakes). Also, one needs an operational definition of the open ocean before defining a connection to the open ocean.

p. 4, l. 17: "is connected to" is more precise than "is in direct contact with"

p. 4, l. 23: Are there published examples of frameworks (i.e., "traditional theory") that cannot handle pinning points? It seems natural to define the ice domain as in Eq. (4), and I'm not aware of frameworks with a different or less natural definition.

p. 4, l. 24: Which "employed assumption" is being referenced here? Maybe the assumption that only ice that is part of the ocean domain is included within the floating ice mask?

p. 4, l. 27: "the first generation of Earth system models". I'm not sure we are still in the first generation, since ESMs have been around for about a decade. Maybe "current Earth system models".

p. 5, Eq. (6): Why is the grounded mask needed in this expression, if it is true that $H = H_0$ for floating ice shelves?

p. 5, l. 8: What is the referent in "it can be negative"?

p. 5, l. 21: What is meant by "directly affects"? Is this just the (fairly trivial) statement that some, but not necessarily all, of the net change in ice mass results in a change in ocean mass?

p. 5, ll. 21-22: "Quantifying the fraction of ice mass change that contributes to sea level. . ." I thought that this was the main point of the section. In what sense is it analytically unapproachable and beyond the scope of the study?

p. 5, l. 23: "Despite the assumption. . ." Which assumption? If the reference is to the assumption that "the net change in ice sheet mass directly affects the ocean mass", I'm not sure there is a contradiction, but I'm not clear on the precise meaning of the assumption.

p. 5, l. 27: "...yields some error." After reading this statement, I was expecting to see quantitative error estimates later in the text. There is an illustration in Fig. 3, but is it possible to state the typical order of magnitude of the error? For instance, is it closer to 1% or 10%?

p. 5, l. 28: Please say precisely what is meant by "contributes to sea level". For example, if an ice sheet loses mass, the geoid will change because of gravitational and rotational effects, but these changes aren't part of DeltaH_S. What is meant, I think, is the part of the ice loss that adds to the mass of the ocean, i.e. the barystatic sea-level component. If so, then barystatic SLR and related terms should be defined here or earlier.

p. 5, l. 30: Since this is a central equation in the paper, I would like to see a clearer description of the physical meaning of each term, and when appropriate a derivation. I convinced myself that the first two terms on the RHS are correct, but I was not able to derive the third term or understand the physical motivation. In the text, the closest thing to an explanation is on p. 7, l. 20: "The last term in the equation accounts for the fact that fresh water density evolves during the accretion and ablation of ice, whereas the average ocean water density in the vicinity of the grounding line acts to determine the ablation height." This is confusing, in part because freshwater density rho_w is a physical constant that does not evolve. Please provide a clearer explanation and, if possible, a supporting figure.

p. 6, Figure 2: The figure and caption are helpful, especially panels a and b with the four different regimes.

p. 7, ll. 3ff: The text refers to three distinct "regimes", whereas Fig. 2 refers to four regimes that are defined differently from the regimes in the text. Please use the term "regimes" consistently. In the text, paragraph 1 corresponds to the first term on the RHS of Eq. (7), and paragraph 2 to the second and third terms. But as stated above, the explanation of the third term is not clear. Perhaps revise so that paragraph 2 addresses the second term and paragraph 3 the third term. Then the current paragraph 3 would become a short paragraph 4.

p. 7, l. 19: Could you give an example of when the magnitude of the change in H_F would be equal to the magnitude of the change in H, and when it would be less?

p. 7, ll. 29ff: I am confused about the difference between events in regimes 1 and 2. My understanding is that regime 1 consists of regions that are grounded at both the start and end of the simulation, whereas regime 2 consists of regions that transition from grounded to floating during the simulation. If so, this should be stated clearly. • For regime 1, it is stated that DeltaH_S is different from DeltaH_F because of "evolving bedrock and sea level". Are bedrock and sea level not evolving in regime 2? Or is the point rather that in region 1, the ice remains grounded throughout the simulation, and therefore the entire DeltaH contributes to DeltaH_S, whereas DeltaH differs from deltaH_F because of bedrock changes? • For regime 2, it is stated that the discrepancy is due to the "missing fraction of newly grounded or newly floating ice." I am not sure what this means. I understand why DeltaH_S differs from DeltaH in this region, but not why DeltaH_F differs from DeltaH_S.

p. 7, l. 30: Please say more precisely what is meant by the "customary approach of using DeltaH_F." Can you cite specific examples in the literature in which the ice-sheet contribution to SLR was derived from DeltaH_F, yielding a significant error? In the literature (beyond this specific example from Larour et al. (2019)), do the errors have a systematic sign? Are these errors prevalent in ice sheet models that include isostatic adjustment (i.e., where DeltaH_S could have been computed accurately, but DeltaH_F was reported instead)? Or is the problem that most ice sheet models ignore isostatic adjustment, so that they are missing a key term needed to compute deltaH_S?

Also, can you state the magnitude of the systematic error? That is, what is the magnitude of the integrated error in Fig. 3 panel c, relative to the integrated value of deltaH_F?

p. 8, Fig. 3: In addition to the 2D fields, it would be useful to show a graph consisting of time series of the area-integrated values of H_S and H_F. This graph could show not only the total values, but also the values computed separately for regions 1, 2, and 3. Also, please cite Larour et al. (2019) in the caption.

[Figure]

p. 8, l. 6: Here, barystatic sea level change is finally defined. I suggest introducing and defining this concept earlier in the paper. Also "ocean mass-related" is a bit vague; I suggest phrasing similar to that of N19 in Gregory et al. (2019): e.g. "the part of global-mean sea-level rise which is due to the addition to the ocean of water mass that formerly resided within the land area as land water storage or land ice." Then DeltaRˆI, introduced below, would be the land-ice contribution, and DeltaRˆL would be the contribution from other land terms.

p. 8, Eq. (8): The term on the LHS includes a subscript, a superscript, and an overbar, without immediately saying what these things mean. I suggest a more gradual and systematic introduction to the notation. Also, could you explain why the denominator contains rho_w instead of rho_o? At first, I assumed that the denominator represents the mass of the ocean, but I think the reason for rho_w is that we are converting a mass of fresh ice into an equivalent ocean volume, ignoring halosteric effects. Again, a more detailed physical explanation would be helpful.

p. 8, Eq. (9): This equation introduces several more terms without preamble, and the reader has to study the following paragraph carefully to translate each term. Please rewrite in a way that is gentler for the reader.

p. 9, ll. 7-9. I am not clear on the meaning of the third and fourth ("past") terms, and how these terms change the ocean mass. I understand that past ice-sheet changes affect sea level through ongoing glacial isostatic adjustment, but isn't GIA included in term 6, the vertical-land-motion term?

A more general comment: It could be easier for the reader if the text were organized from general to specific, instead of specific to general. That is, first define the various kinds of sea-level rise, introduce notation, and state the various source terms. Then state that this paper is focused on R_IˆC, as computed in equation (8) based on DeltaH_S. Finally, show how to compute DeltaH_S. This would be a fairly major rewrite, and I don't want to be too prescriptive, but it is challenging for readers to introduce basic

concepts just a page or two before the conclusions.

p. 9, l. 16: When deltaM is described as a mass-conserving field, is this equivalent to saying that its global integral is zero? Also, is it strictly true that rho_i*deltaH_S is equal to the change in ice mass at each location? Here, I'm wondering about the third term in Eq. (7); is that term associated with a change in the local mass per unit area?

p. 9, Conclusions: As mentioned above, it would be helpful to quantify the benefits of the new methods, e.g. by estimating the errors associated with the older methods.

Minor corrections

p. 1, l. 6: "and include the ice shelves and adjacent ocean mass". The phrasing is awkward. Please use a parallel grammatical construction.

p.1, l. 7: "is" -> "can be"?

p. 1, l. 15: "grounding line" -> "grounding lines"

p. 1, l. 17: Delete "involved"

p. 2, l. 2: "first order" -> "first-order"

p. 2, l. 28: Delete "for"

p. 3, l. 5: "refer" -> "refer to"

p. 4, l. 30:"ice sheet driven" -> "ice-sheet-driven"

p. 4, l. 32: "far field" -> "far-field"

p. 5, l. 2: Add "the" after "estimate"

p. 5, l. 20: "farfield" -> "far-field"

p. 5, l. 26: Add a comma after "below"

p. 6, l. 7: "predict" -> "predicts"

p. 9, l. 15: Insert "the" before "ice sheet"

p. 10, l. 6: "analyses" -> "analysis"

---

## Referee Comment (RC3) · Anonymous Referee #3 · 8 Apr 2020

The paper by Adhikari et al. presents a formalism to calculate the contribution of dynamic ice sheets to mean sea level by considering a changing sea-level, bedrock and land-ice-ocean mask. Compared to standard methods the contribution to mean sea level is computed from ice thickness and not from the height above flotation. The formalism is valid on all timescales, but the full benefit is on longer timescales. As there are currently huge efforts to include dynamic ice sheets in Earth system models the presented paper is a good reference.

Generally, I found the paper well written and structured with illustrative and clear figures. The paper is worth publishing in The Cryosphere but as the focus is rather technical it would fit much better to GMD. The current form of the manuscript lacks a bit in the presentation and clarity. Therefore, I have a few suggestions that should be

addressed in a revised version.

1) I found the title a bit misleading to the content of the paper. I am missing the definition of "mass conservation" in the text. I also would like to see (e.g. with an example), if the formalism is mass-conserving (or better mass-conserving than traditional or other methods). Also, the only example in the manuscript was about calculating SLE relevant thickness changes rather than mass-conservation.

2) The Introduction should refer to Goelzer et al. (2020). You do very quick comparisons to Goelzer et al. (2020) on page7,line6 and 23, but I think the Introduction should clearly say what you are doing differently and why. This could be a motivation to release your new formalism. An appropriate discussion to Goelzer et al. (2020) is also missing. Additionally, from the Introduction it was not really clear to me what is actually wrong with the traditional methods, the order of error on SLE they could introduce and what you are now aiming to improve.

3) Not sure if this could be really addressed, but it would be interesting to estimate the errors (i.e. traditional versus your formalism) of current projections of SLE from the two big ice sheets e.g. within the ISMIP6 framework (Antarctica: Seroussi et al., 2020; Greenland: Goelzer et al., 2020b). Or from current remote sensing products like IMBIE (Shepherd et al., 2019). My point here is, that I would get a better feeling for the error on e.g. different timescales, regional settings and how it differs for Greenland and Antarctica. The example based on the Larour et al. (2019) simulation is very helpful (see also my comment to P7,l29ff) but very specific - and, as I understood – not in line with current projections efforts. I do not strictly insist that you show an error for ISMIP6 or IMBIE, but as also commented below, I would like to have a better error estimate and its impact on current research (compare Fig. 3 in Goelzer et al. (2020)).

Minor comments:

P1,l8: I have not found in the text, which computational strategies you have simplified. What do you mean with computational strategies?

[Figure]

P2,l11: Why are "floating ice shelves, ice rises and rumples, and retrograde bedrock slopes" complex features?

P2,l13: What is" traditional theory for ice-bedrock-ocean interface changes"?

P2,l18: " ... that can be straightforwardly employed in any Earth System model ...". I think it would be worth to mention (somewhere), if this new formalism could be adopted to other disciplines (e.q. remote sensing, standalone ice sheet modelling). In the current form, it sounds the formalism in only valid/applicable in ESMs.

P4,l23: I cannot see from Eq. 4 that your new setup diverges from traditional approaches. Eq. 4 is a very common equation to define an ice-mask. On page5,line 25 you give another example of how the traditional setup differs from your setup. Maybe outlining the differences could be gathered together.

P5,l8: I don't understand this sentence. What is negative?

P5,l9: "... hence contribute to sea level inversely." Maybe say sea-level drop/fall to avoid confusion.

P5,l10: I am not a native speaker, but are both "evolving" really needed?

P5,l25: Can you add a reference to a Figure after "show"?

P7,l4: "... and the elevations"?

P7,l29ff: I would first describe the Larour et al. (2019) setup and then present the results. Can you give an integrated value (e.g. SLE) for both approaches to get a better feeling for the error? The simulations were run over 500 years. Why do you choose to present the results after 350 years?

P9,l6ff: The following paragraphs and Equations appeared very suddenly and without introducing their purpose. According to the title, I would expect Eq. 10 (the mass-conserving field M) is the main point in your paper. But this is not illustrated and somehow contradicts with your statement in the conclusion (p9,l31-32); here you say

[Figure]

DeltaH_s is the main point. This is perhaps personal matter, but I found it a bit brutal to stop the results of the paper with these equations. An illustrative example on "implication of this new geometrical setup for sea level and solid Earth loading studies" (your comment on p2,l20) would make more sense to me.

Fig.3: What is the grey line? And in the caption: Is "conventional"==" traditional"? I guess yes. Please use the same wording in the whole text. Eq. 3 and 4: consider rewriting with "latex-cases"

References: Goelzer, H., Coulon, V., Pattyn, F., de Boer, B., and van de Wal, R.: Brief communication: On calculating the sea-level contribution in marine ice-sheet models,The Cryosphere, 14, 833–840, https://doi.org/10.5194/tc-14-833-2020, 2020.

Goelzer, H., Nowicki, S., Payne, A., Larour, E., Seroussi, H., Lipscomb, W. H., Gregory, J., Abe-Ouchi, A., Shepherd, A., Simon, E., Agosta, C., Alexander, P., Aschwanden, A., Barthel, A., Calov, R., Chambers, C., Choi, Y., Cuzzone, J., Dumas, C., Edwards, T., Felikson, D., Fettweis, X., Golledge, N. R., Greve, R., Humbert, A., Huybrechts, P., Le clec'h, S., Lee, V., Leguy, G., Little, C., Lowry, D. P., Morlighem, M., Nias, I., Quiquet, A., Rückamp, M., Schlegel, N.-J., Slater, D., Smith, R., Straneo, F., Tarasov, L., van de Wal, R., and van den Broeke, M.: The future sea-level contribution of the Greenland ice sheet: a multi-model ensemble study of ISMIP6, The Cryosphere Discuss., https://doi.org/10.5194/tc-2019-319, in review, 2020.

Seroussi, H., Nowicki, S., Payne, A. J., Goelzer, H., Lipscomb, W. H., Abe Ouchi, A., Agosta, C., Albrecht, T., Asay-Davis, X., Barthel, A., Calov, R., Cullather, R., Dumas, C., Gladstone, R., Golledge, N., Gregory, J. M., Greve, R., Hatterman, T., Hoffman, M. J., Humbert, A., Huybrechts, P., Jourdain, N. C., Kleiner, T., Larour, E., Leguy, G. R., Lowry, D. P., Little, C. M., Morlighem, M., Pattyn, F., Pelle, T., Price, S. F., Quiquet, A., Reese, R., Schlegel, N.-J., Shepherd, A., Simon, E., Smith, R. S., Straneo, F., Sun, S., Trusel, L. D., Van Breedam, J., van de Wal, R. S. W., Winkelmann, R., Zhao, C., Zhang, T., and Zwinger, T.: ISMIP6 Antarctica: a multi-model ensemble

of the Antarctic ice sheet evolution over the 21st century, The Cryosphere Discuss., https://doi.org/10.5194/tc-2019-324, in review, 2020.

Shepherd, A., Ivins, E., Rignot, E., Smith, B., van den Broeke, M., Velicogna, I., Whitehouse, P., Briggs, K., Joughin, I., Krinner, G., Nowicki,S., Payne, T., Scambos, T., Schlegel, N., Geruo, A., Agosta, C., Ahlstrøm, A., Babonis, G., Barletta, V. R., Bjørk, A. A., Blazquez, A.,Bonin, J., Colgan, W., Csatho, B., Cullather, R., Engdahl, M. E., Felikson, D., Fettweis, X., Forsberg, R., Hogg, A. E., Gallee, H., Gardner,A., Gilbert, L., Gourmelen, N., Groh, A., Gunter, B., Hanna, E., Harig, C., Helm, V., Horvath, A., Horwath, M., Khan, S., Kjeldsen, K. K.,Konrad, H., Langen, P. L., Lecavalier, B., Loomis, B., Luthcke, S., McMillan, M., Melini, D., Mernild, S., Mohajerani, Y., Moore, P.,Mottram, R., Mouginot, J., Moyano, G., Muir, A., Nagler, T., Nield, G., Nilsson, J., Noël, B., Otosaka, I., Pattle, M. E., Peltier, W. R., Pie,N., Rietbroek, R., Rott, H., Sørensen, L. S., Sasgen, I., Save, H., Scheuchl, B., Schrama, E., Schröder, L., Seo, K.-W., Simonsen, S. B.,Slater, T., Spada, G., Sutterley, T., Talpe, M., Tarasov, L., Jan van de Berg, W., van der Wal, W., van Wessem, M., Vishwakarma, B. D.,645Wiese, D., Wilton, D., Wagner, T., Wouters, B., Wuite, J., and Team, T. I.: Mass balance of the Greenland Ice Sheet from 1992 to 2018,Nature, https://doi.org/10.1038/s41586-019-1855-2, 2019

---

## Author Comment (AC1) · 17 May 2020

**Referee #1**

**Summary**

The paper presents a formalism to geometrically interpret changes in ice sheets and underlying bedrock and their combined effect on the ocean and sea level. The approach defines two distinct domains (land and ocean) that can both intersect areas of ice cover. It then traces ice and bedrock changes and transitions between different domains to determine the sea-level contribution of the ice sheet.

**General comments**

The paper is well written, clearly structured and deals with the important question of how to calculate the sea-level contribution of a marine ice sheet, among others. I believe it would make an interesting contribution to The Cryosphere given that the points raised below are addressed adequately.

We thank the reviewer for a positive and constructive review. Please find our response to individual comments below. (For some questions, we kindly refer to our responses to similar questions by other referees.) Proposed changes to the revised manuscript are in red font.

One of the main conclusions of this paper reads very similar to the one in Goelzer et al. (2020), both papers proposing an alternative to conventionally calculating the sea-level contribution of marine ice sheets based on volume above flotation. It seems important to clarify what the similarities and differences are. The results presented by Goelzer et al. (2020) imply that ice and bedrock changes have to be considered together at least in any place where ice could ground over the course of the experiment. This is the direct consequence of the claim that the sea-level contribution calculated from one point in time to another should be independent of what happens in between. Since the example that is put forward (see their Fig 2a and related text) matches with regime 3 here, there seems to be a direct disagreement between the two approaches: bedrock changes are taken into account in their case, but not here. This may point to a flaw in the approach that should be clarified and discussed. If bedrock changes are not considered as part of the ice sheet change in regime 3, what other component is taking it into account (if any) and how does the domain separation between those components work? If both approaches are not compatible, why and under what circumstances do they differ? What would needs to be changed to make the two approaches compatible? Are the two approaches addressing a different modelling framework, which explains the differences?

We break this question into two parts. (1) What are the similarities and differences between our work and the recent work by Goelzer et al. (2020)? (2) Why are Figure 2a of the Goelzer paper and our regime 3 (as defined in our Figure 2b) inconsistent?

The goal of the Goelzer paper was to provide a correction to a widespread approach of estimating sea-level contribution from ice sheets. The approach, based on the concept of volume-above-flotation, predicts an incorrect sea-level contribution when the ice sheet model accounts for evolving bedrock and sea level, especially in the marine portions of the ice sheet. In particular, Goelzer et al. (2020) provide a correction for the effects of bedrock

elevation change and externally forced (spatially uniform) sea level in stand-alone ice sheet models. Our goal here is to formulate a *new* field – rather than a correction term – that can be utilized not only to quantify ice sheets' contribution to sea level, but also to track evolving grounding lines and coastlines in a seamless manner in light of feedbacks from solid Earth deformation and sea level adjustment. Our formulation can accommodate changes in bedrock and sea level caused by any geophysical processes, and is applied to a whole suite of modeling architectures: from stand-alone ice sheet models (e.g., many of ISMIP6 participating models), to models that account for isostatic adjustment of bedrock (e.g., Le Meur and Huybrechts, 1996; Pattyn et al., 2017) or a self-consistent GRD (gravitational, rotational, deformational) response of the solid Earth (e.g., Gomez et al., 2013; Larour et al., 2019). In the latter set of models, our formulation ensures mass conservation in the Earth System by exchanging mass between land and ocean, accounting for induced GRD response of solid Earth, and adjusting the ocean area through grounding lines and coastlines migration simultaneously. **We will include the following in the Introduction of the revised manuscript:**

> ...Inclusion of evolving bedrock and sea level in a dynamical ice sheet model, however, requires a modification to the existing method of estimating sea level contribution. The method, based on the concept of volume-above-flotation, yields inaccurate results for the marine portions of the ice sheet. In their recent work, Goelzer et al. (2020) provide appropriate corrections for the effect of bedrock elevation change and externally forced (spatially uniform) sea level. Our goal here is to formulate a new field, rather than a correction term, that can accommodate changes in bedrock and sea level caused by any geophysical processes, and is applied to a whole suite of modeling architectures: from stand-alone ice sheet models to those that account for a self-consistent GRD response of the solid Earth (e.g., Gomez et al., 2013; Larour et al., 2019). In the latter set of models, our formulation ensures mass conservation in the Earth System by exchanging mass between land and ocean, accounting for induced GRD response of the solid Earth, and adjusting the ocean area through migration of grounding lines and coastlines.

Regarding the second question, it appears that we were not clear enough in the manuscript in terms of explaining the physical significance of three regimes. To facilitate discussions, we compile relevant figures below (Figure 1). Based on the evolving ice thickness $\Delta H$, bedrock elevation $\Delta B$, and mean sea level $\Delta S$, we classify 3 regimes of the ice sheet. In regimes 1 and 3, grounded ice remains grounded and floating ice remains floating, respectively, over the considered time period $\Delta t$. Regime 2 only captures the portion of the ice sheet that experiences transition from grounded to floating state, or the reverse, over the course of $\Delta t$. Now let us interpret Figure 2a of Goelzer et al. (2020) from the lens of the three regimes we've classified. Over the period $\Delta t = t_4 - t_1$ (see top panel in the figure below), as correctly pointed out by the reviewer the floating ice remains floating (equivalently, our regime 3) and hence the column of ice does not contribute to barystatic sea-level change during this period. Even though $\Delta B$ and $\Delta S$ do not appear to play any role in this regime (as they do not appear explicitly in equation 7), the fact is that they

determine the new grounding line position and hence the domain of regime 3. Reviewer's remark that we do not consider the effect of evolving bedrock elevation in regime 3 is therefore not accurate. If we consider $\Delta t = t_2 - t_1$, on the other hand, the column of ice does contribute to sea level because the then-floating ice is now grounded (equivalently, our regime 2), and we will have to consider the effect of $\Delta B$ and $\Delta S$ explicitly while estimating the sea-level contribution of the ice column over this shorter period. This is consistent with the interpretation made by Goelzer et al. (2020). Although there is consistency between the two methods for particular cases such as the one discussed above, it is important to note that our formalism captures the additional effects of a spatially variable mean sea-level (and gravitational fields in general) induced by any geophysical processes. **We will clarify this in the description of the three regimes in the manuscript.**

[Figure]

Figure 1: Top: Figure 2a of Goelzer et al. (2020). Bottom: Figure 2b of our paper.

While the first few words in the abstract seems to say that this paper is about representing ice sheets in models, the mention of geodesy later may suggest that observations of ice sheets are equally addressed. If the aim is indeed modelling, I think this should be made clear and a clearer distinction be made from observations. If both should be addressed at the same time, I suggest to make sure that the presented formulations make sense in both realms.

By design, the proposed formalism is very generic and it can be used for applications with modeling, observations or combination of models and observations. **We will provide a specific example for each case in the revised manuscript.** Many aspects of our formalism including a generic level set for coastlines and grounding lines (see our response to comment #P4.l6 on page 9) can be used for both modeling and observational applications. However, due to lack of observed data for some fields (e.g., change in bedrock elevation beneath the ice sheet), some other aspects of the formalism (e.g., Eq. 7) are more suited for model-based studies. **We will mention this explicitly in the manuscript.**

While I appreciate the formal description of the case, I miss better guidance of the reader through what is a difficult problem to understand and visualise. Recurrent redefinitions of variables (see example S(w,t) below) should be avoided, individual terms in the equations should be better explained and examples should be given where possible. This particularly applies to cases where the formalism uses familiar concepts and applies them to something else (e.g. floatation condition for ice applied to define the ice free coastline).

Based on this and similar comments from two other reviewers, we desire greater clarity in defining some of the variables and terminologies. As you will find in our responses below (see also our response to two other referees), **we will present more precise definitions of the key terminologies and expand descriptions of individual terms appearing in the equations.** We appreciate this comment, as it greatly helps to improve the focus and clarity of the manuscript.

For understanding and reproducing the results it would be useful to provide access to the data and tools used to produce the results and plots in Figure 3. Please consider making the geometry and scripts available.

Agreed. **We will add the following in the end of the manuscript.**

Code and data availability: Data and a Matlab script used to produce Figure 3 are available online at *DATA LINK*.

**Specific Comments**

P1.l1 Not all ESMs include ice sheet components. Reformulate.

Agreed. **We will rephrase the sentence:**

Polar ice sheets are important components of the Earth System.

P1.l2 The connection between ESMs and geodesy is not clear to me. E.g. observation don't exist for ESM paleo simulation where the formalism should also hold.

ESMs are intimately tied to geodesy, even if these models are not constrained by paleo data. Some of the fields considered in our formalism such as mean sea-level, geoid and bedrock topography are defined relative to a common geodetic datum, such as the International Terrestrial Reference Frame. These fields essentially describe the "shape" of the Earth, and modeling and measuring of which is the very definition of geodesy. Also note that some geodetic observing systems relevant for ESMs have been in place for more than a century. Earth rotation and gravity fields were measured during the first half of the 20th century, for example.

P1.l5 "grounded and floating masks" suggests a modelling perspective, but "as viewed from space" relates to an observational dataset. What is the perspective of this paper?

As noted above, our formalism is generic and applied to both modeling and observational realms, although some aspects of it (e.g., Eq. 7) are more suited for model-based studies due to the apparent lack of observational data. **We will rephrase the sentence:**

> ... requires both geodetically consistent and mass conserving descriptions of evolving land and ocean domains, grounded ice sheets and floating ice shelves, and their respective interfaces.

P1.l5 "Here we present ...". The subject in this sentence is not clear. Reformulate.

As we propose to rewrite the second half of the abstract, **this sentence will not appear anymore.** See our response to Referee #2's comment #p.1.Abstract on pages 18-19.

P1.L13-15 This is clearly true for simulations of the Antarctic ice sheet, but not really for the Greenland ice sheet, which is dominated by surface mass balance changes. Similar cases may also exist during other climates with ice sheets that were mainly land based. Suggest to reformulate.

Agreed. **We will explicitly include "marine ice sheet" in the sentence:**

> Indeed, how well a numerical model of marine ice sheet projects its future sea level contribution largely depends on ...

P2.l2 "Defining geometry". Do you mean "Defining the bedrock"? How could the geometry be defined upfront for an intercomparison when the models are supposed to produce an evolving geometry?

We agree that this statement is ambiguous. **We will delete it altogether.**. The removal of this sentence does not break the flow of the write-up.

P2.l5 What do you mean with "basic configuration setup" and "Similar setups"? Could you describe this in more detail?

**We will rephrase these sentences in a more clear way:**

> General description of mechanical analysis of ice sheet evolution at the ice-bedrock-ocean interfaces has been given for a set of simplified geometries... A similar geometric approach is also familiar in the development of glacial isostatic adjustment (GIA) theory...

See our response to Referee #2's comments #p.2,l.8 (pages 19-20) and #p.4,l.23 (page 23).

P2.l12 Why are "floating ice shelves" and "retrograde bedrock slopes" complex features? They occur in the very simple flowline model setups you may be referring to above. Clarify.

Agreed. We will replace "floating ice shelves" and "retrograde bed slopes" by "rugged fjord geometries" and "uneven bed topography". These features are "complex" relative to the simplified geometries considered in previous studies of mechanical analysis of grounding line migration (e.g., Hutter, 1983; Lambeck et al., 2003). We will avoid the usage of word "complex" in the manuscript. **We will revise the statement as follows:**

> ...kilometer-scale geometric features, such as ice rises and rumples, rugged fjord geometries, and uneven bedrock topography. These features complicate the required geometrical simplifications used in previous studies for

ice-bedrock-ocean interface changes...

P2.l13 What is the "traditional theory for ice-bedrock-ocean interface changes"? Clarification needed.

We meant to imply the previous studies that considered simplified geometries (e.g., Hutter, 1983; Lambeck et al., 2003; Mitrovica and Milne, 2003), lacking kilometer-scale features such as pinning points and rugged fjords. **See also our response to Referee #2's comments #p.2,l.8 on pages 19-20.** This phrase, particularly "traditional theory", will be excluded in the revised manuscript.

p2.l23 "setup" → maybe "interpretation"?

**This word will not appear in the revised manuscript.** We will simply write:

> To begin our discussion, we consider a spherical planet...

p2.l24 What is the difference between "glaciers" and "ice sheets" in your description? Clarify if the two terms are interchangeable or distinct. If the latter, what sets them apart in your formalism?

The formalism does not need to distinguish between glaciers and ice sheets. In fact, we prefer a generic term such as "distributed ice domains" (line 24). We use glaciers and ice sheets to give a sense to the readers of what we meant by distributed ice domains.

p2.l24 I found the upfront separation between land and ocean confusing for your context, because it is not intuitive where that separation is to be made for a marine-based ice sheets. The definition what is to be considered land and ocean comes too late. I suggest to make that clearer much earlier.

**We will insert the following sentence:**

> We generally consider ice domains as part of the land, except where they float on the oceanic water (e.g., ice shelves).

P2.l25 Same for S(w,t), defined here first as the sea-surface elevation. How should we think of S for an ice shelf? Why not start with defining S as the geoid as you do later.

We agree that the definition of $S(\omega, t)$ is indeed confusing. **We will clarify it by including the following sentences:**

> We first introduce two fields, $B(\omega, t)$ and $S(\omega, t)$, to be the solid Earth surface (i.e., land surface or sea floor) and the mean sea level (MSL), respectively. These fields must be defined relative to the same reference ellipsoid, preferably consistent with the International Terrestrial Reference Frame... Our definition of MSL complies with that given by Gregory et al. (2019): Time-mean of sea surface over a sufficiently long period so that the effects of waves, tides, or meteorologically-driven high-frequency fluctuations are eliminated... Specifically, the change in global-mean of MSL in the present context is equal to the so-called barystatic sea-level change that defines global-mean sea-level

change due to a sustained exchange of water between the land and the ocean. The evolving spatial pattern of MSL, on the other hand, is strictly dictated by the GRD response of solid Earth to land-ocean mass exchange. This definition of evolving MSL is familiar in GIA modeling wherein there is a requirement to solve the so-called "sea level equation" (Farrell and Clark, 1976). MSL as defined above represents an equipotential surface whose spatial pattern matches the geoid (Tamisiea, 2011), and is therefore a globally-defined field.

P2.l26 I admit, I had to look up what the ITRF is. For other readers not familiar with it, you may want to add a sentence or two to say what the ITRF provides. In practice, if I use Bedmachine data, is it registered on the ITRF or are you suggesting this is something the user would have to take care of herself?

ITRF is the standard reference frame defined by the International Earth Rotation and Reference Systems Service (www.iers.org). It is being updated every few years. Glaciological datasets (e.g., surface and bedrock DEMs) are intimately tied to a reference frame. BedMachine v3 data, for example, are referenced to the WGS 84 ellipsoid. This and ITRF's recommended ellipsoid, GRS80, should not differ from each other by more than a few centimeters. This difference is much smaller than the uncertainties in BedMachine data. It is also possible to project data from one ellipsoid to another. **We will add a brief description of ITRF and its relation to glaciological dataset in the manuscript.**

P2.l31 So far $S(\omega, t)$ is defined as sea-level. As such, any case $B > S$ is not well defined. The interpretation of S as the geoid must come earlier for this to make sense.

Mean sea level $S(\omega, t)$ is a global field. **We will include the following statement:**

MSL as defined above represents an equipotential surface whose spatial pattern matches the geoid (Tamisiea, 2011), and is a globally-defined field.

P2.l32 You say here that $S(\omega, t)$ includes high-frequency noise and variability, but on the next page you want $S(w, t)$ to refer only to the quasi-static component of the sea-surface. Why not introduce $S$ directly as the quasi-static component of the geoid, rather than going through three redefinitions along the way (sea surface → quasi-static sea-surface → sea-level → geoid).

We appreciate the comment. This issue is already addressed above. **See our response to comment #P2.l25 on page 6.**

P3.l1 Remove "changing" before interactions?

Agreed.

p3.l7 What is the "interior of marine ice sheets"? Clarify

We meant to imply that $S(\omega, t)$ is a globally-defined field. **This sentence will not appear in the revised manuscript.** See our response to comment #P2.l31 on page 7.

P3.l9 Say and explain what $F(\omega, t)$ means. Traditionally it determines if the ice is floating.

But you seem to extend it to locations with $H = 0$? Maybe it would be worth it to mention that.

The function $F(\omega, t)$ is indeed a generalization of the concept of flotation condition for ice. **We will include the following in the revised manuscript:**

> We develop our formalism based on a familiar concept in glaciology: the flotation condition for ice. Since $H(\omega, t)$ is defined on the entire planetary surface, this concept can be generalized to deduce a criterion for delineating boundaries between the land and ocean, and the floating and grounded ice.

p3.l13 Please define what "open ocean" means and what "contact with the open ocean" means. This definition comes too late in the manuscript. Does $w$ have to be part of $\mathcal{O}$? Maybe we need the definition of $\mathcal{O}$ already before this part not on page 4?

We will simply write "ocean" consistently throughout the manuscript. **We will define it up front in the section as follows:**

> ...we consider a spherical planet whose surface is divided into complementary domains of land and ocean. Ocean may be thought of as an interconnected system of oceanic waters, just like Earth's ocean that includes fjords and marginal seas (e.g., Mediterranean Sea).

To avoid the ambiguity, we will provide a mathematical description of a generic level-set for seamless delineation of land-ocean-ice interfaces. **See our response to comment #p4.l6 on page 9.** The quoted words will not appear in the manuscript.

**The mathematical definition of $\mathcal{O}$ will remain where it is**, because it relies on the definition of $F(\omega, t)$.

p3.l14 A more obvious definition of a generalised coastline for me, that also exists in presence of a marine-based ice sheet would be the point where the bedrock and the geoid intersect (1/2 in Fig2). That doesn't help for your formalism, but it goes to show that it is not immediately obvious to think the coastline at the grounding line. Better guidance needed.

**We would like to respectfully disagree with the reviewer on this point.** We believe that our formalism, which treats grounding lines and coastlines as a seamless level set of $\mathcal{F}(\omega, t) = 0$, makes it appealing for Earth System models. See our response to comment #P4.l6 on page 9. The reviewer is suggesting the boundary between regime 1 and regime 2 to be the coast line (see Figure 2a of the paper). Unlike grounding line, tracking migration of this boundary neither improves our understanding of ice sheet dynamics nor helps us determine an evolving ocean surface area, for example.

P3.l8-15 I found this paragraph difficult to follow. You start with reference to Fig. 2a, where our focus is on the left hand side and with $F(\omega, t)$ which suggests it is about ice. But then you describe $R(w, t)$ and the coastline, which are difficult to visualise in a place with an ice shelf. It may help to guide the reader by being explicit about the two "generalisations" that take place here: floatation criterion (for ice) $\rightarrow$ definition of the

coastline (everywhere). grounding line (for ice) → coast line (everywhere).

We have already addressed most of the issues noted here. **See our responses to comments # p3.l9, p3.l13 and p.4l1 on pages 7-9. We will move the definition of $R(\omega, t)$ to the new section "Mass conservation in the Earth System".** With these changes, we believe that the paragraph should be clearer and easier to follow.

P4.l1 I don't see why there could not be a grid point in a model with $B = S$ and $H > 0$. A glacier terminating on land or on a sill exactly at sea-level? Please clarify.

We are glad that the reviewer raised this point. Indeed, there is a subtle assumption that goes into our interpretation of equation (1). **We will write it explicitly in the revised manuscript.**

> To simplify our analysis, we ensure the absence of marine ice cliffs that have larger thickness than the flotation height (negative of the second term on the right side of the equation) by assuming that $F(\omega, t) \leq 0$ holds at the "ice front". The ice front that satisfies the equality (inequality) in the mentioned relation represents the calving face of a tidewater glacier (an ice shelf). Along the same lines, we assume that terrestrial ice cliffs do not present where $B(\omega, t) = S(\omega, t)$. These assumptions are required for an unambiguous definition of land-ocean boundary, and are generally valid due to the diffusive flow of ice.

P4.l6 The fact that neither O nor L are defined at the grounding line seems problematic. How can your formalism be mass conserving when grounding line grid points in an ice sheet model are not part of these masks? How do you track the grid cells that fulfil this condition, do they form a separate category? Why would it not matter to consider them?

$O$ and $L$ are complementary fields such that their surface areas make up the total area of Earth's surface. Coastlines and grounding lines form their own level set, with no surface areas, which can be tracked straightforwardly. As such, there should not be any "cells" outside both $O$ and $L$, and therefore there should be no problem regarding mass conservation. We agree that we did not provide a (unified) mathematical description of the coastlines and grounding lines. **We will add the following in the manuscript:**

> We now define a level set of the function $F(\omega, t)$ such that:
>
> $$T(F) = \{(\omega, t) \mid F(\omega, t) = 0\}.$$
>
> Note that $T(F)$ consists of several simple curves, $T_i(F)$, that form several non-overlapping regions, $\Omega_i(F)$. These regions together cover the entire planetary surface. Let $\Omega_i^-(F)$ denote the regions in which the function $F(\omega, t)$ takes negative values (candidates of the ocean domain). Since we consider ocean to be an interconnected system of oceanic waters, only the largest amongst $\Omega_i^-(F)$ forms the ocean domain. Smaller $\Omega_i^-(F)$, if there are any, and their boundaries $T_i(F)$ are considered to be part of the land. Let $\Omega_{\mathcal{L}}^-(F)$ be the union of all these small newly-considered land regions and $T_{\mathcal{L}}(F)$ be the union of

corresponding boundaries. We define a new function

$$\mathcal{F}(\omega, t) = |F(\omega, t)| + \epsilon \quad \text{if } \omega \in \Omega_{\mathcal{L}}^-(F) \text{ and } \omega \in T_{\mathcal{L}}(F)$$
$$= F(\omega, t) \qquad \text{otherwise,}$$

so that the level set

$$\mathcal{T}(\mathcal{F}) = \{(\omega, t) \mid \mathcal{F}(\omega, t) = 0\}$$

represents the land-ocean boundaries. Here $\epsilon$ is a positive number to ensure $\mathcal{F}(\omega, t) > 0$ at $\omega \in T_{\mathcal{L}}(F)$.

Also, **see our response to Referee #2's comment #p.4,l.7 on page 22** to find out the new definition of $\mathcal{O}$. We believe that the definition is free of any ambiguities.

P4.l9 Remove "deep" and "well". I suppose the condition could also be true for shallow troughs with bathymetry moderately below sea-level.

Agreed.

P4.l10 While I understand the use of this connectivity concept in your formalism, I find it problematic in practice. It means that small changes in ice or bedrock can lead to very large changes in O and L. In an unfavourable configuration, the short term grounding and ungrounding of a critical point could e.g. switch an entire system of connected fjords on and off.

**With a newly-added mathematical description of level set function $\mathcal{F}(\omega, t)$ (see our response to comment #p4.l6 on page 9), the (relatively subjective) concept of "connectivity" does not need to be invoked anymore.** However, our formalism still supports the idea of potentially having large changes in $O$ and $L$ due to relatively small changes in bedrock or mean sea level. And, this is precisely what happens in reality (at least from mass conservation perspective) and it should be an essential feature in paleo simulation of a system of ice sheets, solid Earth and sea level.

p4.l18 With the above, combining the grounded and floating ice masks leaves a hole at the grounding line. Is this desired?

Again we understand the confusion created by the lack of mathematical description of grounding line level set, **which will be included in the revised manuscript (see our response to comment # p4.l6 on page 9).** Adding grounded and floating ice masks to grounding line level set would leave no hole in the ice domain.

P5.l13 The geoid typically changes first, then the bedrock. Maybe re-order in the sentence.

**We would like to respectfully disagree with the reviewer.** For a given relaxation frequency, both geoid and bedrock elevation evolve simultaneously.

p5.l22-23 I am confused about this sentence. Isn't "quantifying the fraction of ice mass change that contributes to sea level" exactly what you are doing by defining $\Delta H_S$ below? Reformulate?

We meant to imply that not all of ice mass loss makes it to the ocean. A fraction of it, for example, may get impounded in proglacial lakes. However, the quoted statement is in direct conflict with the assumption that "the net change in ice sheet mass directly affects the ocean mass" (line 21 of the original manuscript), **so we will simply delete it.** We thank the reviewer for catching this apparent inconsistency. The reviewer is correct that $\Delta H_S$ provides the accurate estimate of ice mass loss.

p5.23 "the assumption" appears three sentence back, maybe refer to it more specifically.

**The in-between sentences will be deleted (see our response to the previous comment).** This should fix the issue noted by the reviewer.

p5.23 Remove "all the time".

Agreed.

p5.26 "As we show below". This has been shown before by others (see references). Reformulate to avoid confusion.

Agreed. **We will simply remove the quoted phrase.**

p5.l30 Can you please explain what the three terms mean physically. E.g. the first term accounts for thickness changes of ice that is and remains grounded ...

Sure. But, we realize that the third term does not belong to this equation, especially when we interpret $\Delta H_S$ as a contributor to the barystatic sea level change $\overline{\Delta R}$ (see Eq. 8 of the manuscript). By definition, $\overline{\Delta R}$ is the global mean sea level change due to water mass exchange between land and ocean (Gregory et al., 2019). The third term appearing in Eq. (7) certainly contributes to sea level change by contributing not to the ocean mass but to its volume. So, we will treat this term separately, and **its physical interpretation is discussed in our response to the next comment (#p5.l30 on pages 11-12). We will modify Eq. (7) and include the following interpretation in the manuscript:**

$$\Delta H_S(\omega, \Delta t) = \Delta H(\omega, \Delta t)\ \mathcal{L}(\omega, t)\ \mathcal{L}(\omega, t + \Delta t) +\ |\Delta H_F(\omega, \Delta t)|\ \Delta \mathcal{L}(\omega, \Delta t)$$

While a detailed physical interpretation of the right-hand side terms is given in the following paragraph, we briefly note that the first term accounts for ice thickness change in the regions that remain grounded at both times (Regime 1 in Figure 2b) and the second term accounts for the absolute change in height-above-flotation in the regions that transit from the grounded to floating state, or vice versa, over the period $\Delta t$ (Regime 2).

p5.l30 Could you explain why H0 appears as an *absolute* contribution in the third term compared to considering *changes* in $H_S$ and $H_F$ in term one and two?

The third term appearing in Eq. (7) accounts for a missing piece of the so-called density correction, in which ice mass is converted into freshwater (having density $\rho_w$) rather than seawater ($\rho_o$) as is done in Eq. (8). It is far easier to explain why we need $H_0$ at time $t + \Delta t$ to derive the third term in Eq. (7) using a simple sketch (see Figure 2 below).

[Figure]

Figure 2: Physical interpretation of the third term appearing in Eq. (7) of the manuscript.

In this figure, we have a grounded ice column having its initial thickness H(1) at time 1 (see A). We remove $\Delta H$ such that the ice column (at time 2) is now at the flotation point (see B), i.e., ice column is in hydrostatic equilibrium with the ocean water. The column of ice is now considered a part of the ocean. However, if the column of ice is melted, its freshwater equivalent may be visualized as the column C. So, clearly the level of the column is larger than that of sea level, and the excess layer (hatched portion) should be distributed to the ocean. The third term in Eq. (7) represents the height of the hatched portion (see C), in terms of ice-equivalent-height. In D and E we show the density-correction applied to $\Delta H_S$, which is performed in Eq. (8).

**We will include this material in the manuscript as a Supplementary Figure.** As noted in our response to the previous comment, however, we will treat this contribution separately (not as part of Eq. 7).

p6.l6 Not clear what "holding in the areas of on-land ice margin migration" means. Reformulate.

We realize that the quoted statement is in direct conflict with the assumption that "the net change in ice sheet mass directly affects the ocean mass" (line 21 of the manuscript), **so we will simply delete it.** It implies that all $\mathcal{L}$'s appearing in equation (7) are interchangeable with corresponding $\mathcal{G}$'s. **We will clarify this in the manuscript.** We thank the reviewer for helping us catch this minor inconsistency.

p7.l15 I suppose you mean that the nonzero $\Delta H_F$ is compensated by other terms in Eq. 7. Which ones? This is important to understand.

No, but thanks for the opportunity to clarify. What we mean is that externally forced $\Delta B$ or $\Delta S$ may possibly modulate ice dynamics in this regime, even if they do not contribute to $\Delta H_F$ or $\Delta H_S$. We will rephrase the sentence and expand it as follows:

> ... it may alter bedrock slope or gravitational driving stress and possibly

modulate the ice flow dynamics. While the effects of far-field ice melting and associated ocean loading may be negligible due to their relatively long-wavelength imprints, $\Delta B(\omega, \Delta t)$ due to large earthquakes beneath the ice sheet may have some impact on ice dynamics.

p7.l22-23 This sentence may be confusing because Goelzer et al. (2020) consider not only transitions between grounded and floating ice, but all three regimes. Reformulate.

This is correct. Now that **we will include a summary of the Goelzer paper and contrast it against our goals in the Introduction (see our response on page 2)** and that their paper is cited on p7.l7 of the manuscript and in other relevant places, we really do not need this citation here. **We will simply delete the sentence.**

p7.l26 Important to note that in regime 3 bedrock changes can contribute to sea-level change, even if you are not considering them as part of the ice sheet change. Also important to realise that ice floating at $t$ and $t + \Delta t$ may have been grounded at some point in between. I think this makes your solution dependent on the time stepping. See Fig 2a in Goelzer et al. (2020) for an example of such a case.

We acknowledge that the bedrock change in this regime may contribute to spatial pattern of sea level. By referring to equation (7), all we are saying here is that there is no barystatic sea-level contribution to the ocean from this regime of the ice sheet. **We will rephrase it accordingly:**

Despite the change in $\Delta B(\omega, \Delta t)$ and $\Delta S(\omega, \Delta t)$, there is no barystatic sea level contribution from this regime of the ice sheet over the period $\Delta t$.

Once $\Delta H_S$ is defined (Eq. 7 of the original manuscript), which is what our focus is at this point, a self-consistent GRD (gravity, rotation, deformation) model of solid Earth yields self-consistent solutions for bedrock elevation change and sea level change. This is a whole different story, and **will be expanded in the new section "Mass conservation in the Earth System".**

Regarding the last question referring to Fig 2a of Goelzer et al. (2020): **Given our explanation on pages 2-3 (see Fig. 1) on the topic, we believe that there is no confusion left in this regard.** Nevertheless, here is our brief response to the questions listed above. **Yes, ice could have been grounded in this region at some point in time** between $t$ and $t + \Delta t$. In this instance, the part of the region that is grounded belongs to regime 2 (not 3; see Fig. 1b) and it should be interpreted accordingly. **Yes, our solution is time dependent and this is what it should be.** See Fig. 1a. There is no net mass contribution to sea level from the ice column during the period $[t_1, t_4]$, while there is clearly non-zero mass contribution from ice during the sub-period $[t_1, t_2]$.

p8.l7 The change in $\Delta H_S$ and $\Delta R_C^I(\omega, t)$ is not only due to ice mass changes, but also due to bedrock changes under the ice. It may be good to mention that here.

We think that the reviewer got confused between "ice **mass** change" and "ice **thickness** change". $\rho_i \Delta H_S$ essentially describes the change in ice mass per unit area (units of kg/m$^2$).

And, this field is contributed by ice thickness change, bedrock elevation change and mean sea level change, which we have thoroughly discussed in the original manuscript (see pages 5-7). **We do not wish to list all of these contributors explicitly again.**

p8.l8 For clarity, could you mention why potential changes in ocean area from $O(\omega, t)$ to $O(\omega, t + \Delta t)$ do not matter for $\Delta R_C^I(\omega, \Delta t)$ in Eq 8? I suppose the underlying assumption is that we should be interested in sea level at time $t + \Delta t$?

The change in ocean area from $O(\omega, t)$ to $O(\omega, t + \Delta t)$ does matter. The ocean area at time $t + \Delta t$ (i.e., the integral in denominator of Eq. 8) is the sum of the ocean area at time $t$ and change in the ocean area over the period $\Delta t$. **We will clarify this in the manuscript.**

p8.l12 $\mathcal{G}(\omega, t)$ is not explicit in Eq. 7 only implicitly by evoking Eq. 6. This could be mentioned.

Agreed. **We will add the following sentence:**

> Note that $\mathcal{G}(\omega, t + \Delta t)$ is implicit in equation (7) via $\Delta H_F(\omega, \Delta t)$, which can be derived from equation (6).

p8.l14. Maybe remind us what R(w,t) is here, as it is only introduced inline and back at p3.

**We will actually move the definition of $R(\omega, t)$ from page 3 of the original manuscript to around here.** See our response to comment # P3.l8-15 on page 8.

p8.l15 To make this equation more digestible, maybe start with combined symbols. E.g. by combining all the barystatic components like you do in the text.

Other reviewers also have similar concerns. **To address these, we will introduce a new section "Mass conservation in the Earth System" at around p8.l6 of the original manuscript.** Here is the tentative structure of the section:

1. We will introduce $R(w, t)$ and a suite of processes that contribute to it (equation 9).

2. We will specify the component that is of interest for the present-study, i.e. $R_C^I$.

3. We will define barystatic sea level (equation 8) and elaborate on how $R_C^I$ is computed on a self-consistent mass conserving GRD model of the solid Earth (equation 10). New materials (2-3 paragraphs) will be added to elaborate the discussion.

4. We will highlight the importance of generic coastline level set (see our response to comment #P4.l6 on page 9 ) and $\Delta H_S$ (equation 7 of the original manuscript) in light of sea level and mass conservation in the Earth System.

See also **our response to Referee #2's comment #P.1,Title on pages 17-18.**

p9.l1-24 My experience with the paragraph including Eq. 9 is that a lot of new concepts are suddenly thrown in here without much preparation. Especially the idea to separate the effect of the past from the contemporary would profit from some more introduction. May be a new section with a few introductory sentences could be started p8.l6 to prepare the ground for this discussion.

We appreciate the feedback. We will restructure the materials presented on page 8/line 6 through page 9/line 23. **See our response to the previous comment.**

p9.l3 Since $\Delta R_V(\omega, \Delta t)$ may be the component that takes into account bedrock changes in regime 3 and is complementary to the bedrock changes happening under the ice, I would be interested to see how it is calculated and how the masking works in that case.

One of the strengths of our formalism is that $\Delta H_S$ as defined by equation (7) considers the effects of $\Delta B$ and $\Delta S$ perturbed by all kind of processes (e.g., glacial/oceanic loads or tectonics). This is clearly stated on page 5 (lines 19-20) of the manuscript. Proposed new materials (see our response to comment #p7.l15 on page 12) only strengthen this fact. Our partitioning of the ice domain into 3 distinct regimes is for a convenient interpretation of $\Delta H_S$. **Regime 3 & other 2 regimes for that matter are defined by considering the effects of $\Delta S$ and $\Delta B$, including $\Delta B_V$** – the component of bedrock elevation change that corresponds to $\Delta R_V$. Equation 9 summarizes the processes that contribute to $\Delta R$. **Providing a detailed theoretical/numerical description of each term, including $\Delta R_V$, is beyond the scope of this study. In terms of masking, both $\Delta B$ and $\Delta R$ are globally-defined fields and there is no need for masking.**

p9.l28 By your definition, the grounding lines are neither part of the grounded ice nor part of the floating ice domain. What is this category called and how are you accounting for it?

**We will address this by including new materials** that define a generic level set for coast lines and grounding lines. **See our response to comment # P4.l6 on page 9.**

p10.l4-5 How does your approach compare to/ differ from that proposed by Goelzer et al. (2020).

In a nutshell, they provide correction terms associated with an evolving bedrock and an externally forced (spatially uniform) sea level for a widespread method of estimating ice sheets' contribution to the barystatic sea level change (BSLC). Their focus is the marine portions of stand-alone ice sheet models. In contrast, we provide a new field altogether, $\Delta H_S$, that yields BSLC by accounting for spatially variable $\Delta B(\omega, t)$ and $\Delta S(\Delta t)$ caused by any geophysical processes. Our method is applied to all plausible settings of ice sheets (e.g., terrestrial- vs. marine-based and grounded vs. floating). In addition, our goal is also to track grounding lines and coastlines in a seamless manner in order to conserve mass in the Earth System and facilitate complex interactions between ice sheets, solid Earth and sea level. While we do not wish to be critical of the Goelzer paper, **the new materials that summarizes their work and contrast it against our goals in the Introduction (see our response on pages 1-2) should provide enough information in this regard.**

**Figures**

Figure 1. It seems confusing to introduce lakes, subglacial lakes and proglacial lakes and then not consider them at all. It would make the figure much clearer to remove them.

Our goal is to make our formalism as generic as possible. Given the complexities of Earth's

surface features, it is an extremely challenging task. Figure 1 depicts some of the relevant features (and associated modeling challenges), and in the caption we have clearly stated what is considered and what is not considered in our analysis. **We think this is important. We prefer to keep the figure as is.**

Figure 2. The geoid (and $S$) is also defined over land. Please add in both panels.

It is true that $S$ is not drawn over land. **We will do that in the revised figure.**

Figure 3. Please include a panel with $\Delta H_F$. This is important as it is discussed as the conventional method and appears in the difference in panel c. Please also add contour lines to delineate the regions 1-3. Mention that regime 4 does not exist in this region if that is true. Otherwise, delineate regime 4. If different from results in Goelzer et al (2020) it would also be interesting to add a comparison as figure here.

**We will include a new panel in the figure as suggested. We will also add contours to all panels to show 3 regimes.** We find that 4 regimes defined in Fig. 2a to interpret $H_F$ may be confused with 3 regimes defined in Fig. 2b to interpret $\Delta H_S$. We will label the formers as Sectors A, B, C, and D. In principle, the Goelzer method should yield similar, if not the same, result as ours. **We will calculate, compare, and report the results appropriately,** either as a new figure panel or in the figure caption.

**Referee #2**

**Summary**

This paper presents a framework for describing the geometry of an evolving ice sheet margin in Earth system models, in which the geometry of the bedrock and mean sea level also can be dynamic. The authors define relevant terms and give a mathematical description of the way different quantities are related. In particular, they quantify the portion of ice thickness change that contributes to changes in ocean mass and global mean sea level. As Earth system modelers work to integrate dynamic ice sheet models with solid-Earth and sea-level models, this paper will be a useful reference.

In general, the paper is well written, and the figures are very helpful. Sometimes, however, technical terms related to sea level are used without precision, or are introduced without giving enough background information. Some of the equations contain terms that are not clearly defined or explained physically. In the comments below, I suggest where the exposition could be improved, especially for readers approaching these concepts for the first time.

We thank the reviewer for a positive and constructive review. We will clarify concepts and terminologies that are ambiguous, supply additional background materials as required, and explain physical meanings of key mathematical terms that are missing. Find our response to individual comments below. **For questions that are similar to those from Referee #1, we kindly refer to corresponding responses that are already documented.** Proposed changes to the manuscript are in red font.

Also, the text refers to "traditional" or "customary" approaches of estimating sea-level contributions from ice sheets based on the change in height above flotation, in contrast to the approach described here. It would be helpful to see some specific examples of customary approaches from published papers, with estimates of the magnitude and sign of the associated errors. This would enable readers to better assess the value of the proposed formalism.

Most of stand-alone ice sheet models use $\Delta H_F$ to estimate ice sheet's contribution to barystatic sea level change (BSLC). Many of participating models of SeaRISE or ISMIP6 projects are some examples.Unless we compare these BSLC estimates with those derived from better/improved methods (e.g., our methods or Goelzer et al., 2020), **we cannot quantify/report errors associated with these estimates.** This is why the results presented in Fig. 3, based on a particular model run by Larour et al. (2019), is useful.

**Major comments**

p. 1, Title: The title includes the words "mass conserving", but in the text I did not find an operational definition of what this means in an ESM with evolving ice sheets. Please provide such a definition, and perhaps an example of how mass conservation would be violated.

We agree that we did not provide much information about "mass conservation" in the

manuscript, except for Eq. 10. **We will state the following in the Introduction:**

> Our formulation ensures mass conservation in the Earth System by exchanging mass between land and ocean, accounting for induced GRD [gravitational, rotational, deformational] response of the solid Earth, and adjusting the ocean area through migration of grounding lines and coastlines simultaneously.

After Eq. 10 where we define a mass conserving field $\Delta M$, **we will write:**

> Stand-alone ice sheet models (e.g., many of ISMIP6 participating models) or the models that account for local/regional isostatic adjustment of bedrock (e.g., Le Meur and Huybrechts, 1996; Pattyn et al., 2017) do not fully capture $\Delta R_C^I$ and hence violate mass conservation in the Earth System.

Also, the title implies that there will be a detailed analysis of ice-sheet interactions with solid Earth and sea level, but the actual scope seems narrower: to accurately compute the contribution of dynamic ice sheets to barystatic sea level rise (i.e., the sea-level component associated with a redistribution of mass between land-based ice and the ocean) in ESMs. I suggest a revised title that better reflects the scope.

We fully understand the reviewer's concerns, and **we are open about modifying the title suitably**. However, our wish here is to improve the manuscript by constructing the formalism in a broader context of sea level and mass conservation in the Earth System. We believe that the central focus of our paper is **a new field** $\Delta H_S(\omega, \Delta t)$ for estimating ice sheet's contribution to sea level (Eq. 7 of the manuscript) and **a newly-added generic level set** $\mathcal{T}(\mathcal{F})$ for tracking land-ice-ocean interfaces in the Earth System (see our response to comment #P4.l6 on page 9). Estimation of $\Delta H_S(\omega, \Delta t)$ and $\mathcal{T}(\mathcal{F})$ require knowledge of $\Delta H(\omega, \Delta t)$, $\Delta B(\omega, \Delta t)$, and $\Delta S(\omega, \Delta t)$ along with accurate geometry of land-ice-ocean interfaces. **These fields and level-set are intertwined with each other, and only by using a mass conserving Earth System model that can capture ice sheet dynamics, solid Earth deformation, and sea level adjustment may we find self-consistent solutions.** As noted in our response to comment #p8.l15 on page 14, **we will introduce a new section "Mass conservation in the Earth System"** at p.8,l.6 of the original manuscript in order to accommodate (and expand on) aforementioned discussion and put $\Delta H_S(\omega, \Delta t)$ and $\mathcal{T}(\mathcal{F})$ in a broader context of sea level change and mass conservation in the Earth system.

p. 1, Abstract: The abstract is very general and does not provide a clear sense of the scope of the paper. If a central goal is to describe how to accurately compute the ice sheet contribution to barystatic sea level rise, then this goal should be clearly stated.

We agree that our abstract is a bit vague and we are aiming at a more impactful abstract. **We propose the following:**

> Polar ice sheets are important components of the Earth System. As the geometries of land, ocean, and ice sheet evolve, they must be consistently captured within the lexicon of geodesy. Understanding the interplay between the processes such as ice sheet dynamics, solid Earth deformation, and sea level

adjustment requires both geodetically consistent and mass conserving descriptions of evolving land and ocean domains, grounded ice sheets and floating ice shelves, and their respective interfaces. Here we present a mathematical description of generic level-set that can be used to track both the grounding lines and coastlines in a seamless manner in light of ice-ocean mass exchange and complex feedbacks from solid Earth deformation and sea level adjustment. In order to improve estimates of ice mass that contributes to ocean mass change, we next present a new field based on the change in ice thickness, bedrock elevation and mean sea level caused by any geophysical processes. Our formalism is applied to arbitrary geometries of land-ocean-ice interfaces, and at all time scales. It is suited for a range of modeling architectures: from stand-alone ice sheet models to those that account for a self-consistent GRD (gravitational, rotational, deformational) response of the solid Earth. In the latter set of models, our formalism conserves mass in the Earth System.

p. 1, l. 1: Although I am fully in favor of including dynamic ice sheets in ESMs, I would not go so far as to suggest that "any Earth System model" already includes them. For some ESMs, ice sheets are still on the back burner.

Agreed. See our response to the previous comment.

p. 2, l. 2: "Defining geometry". I am not sure what this means – something like defining geometric concepts that are relevant to models with dynamic ice sheets?

Referee #1 also had a concern about the quoted phrase. We have decided to remove the entire sentence altogether. **See our response to comment #P2.l2 on page 5.**

p. 2, l. 4: "future debate and reconciliation." Ideally, the concepts are set forth clearly in a way that leads to greater mutual understanding and less debate.

This phrase and "defining geometry" (see previous comment) both appear in the same sentence, **which we have decided to remove from the revised manuscript.**

p. 2, l. 5: "basic configuration setup". I am not sure what this means. It seems an odd way to describe what seem to be theoretical or analytical frameworks.

Referee #1 also found this phrase unclear. We will reformulate the sentence in much clearer way. **See our response to comment #P2.l5 on page 5.**

p. 2, l. 8: I am not clear why these previous analyses are referred to as "traditional configurations". The word "traditional" suggests a contrast with something novel and untraditional to be introduced here. However, I don't see this work as heading off in a different direction from the cited papers, but rather as clarifying concepts that are particularly relevant for ESMs. As above, "configuration" doesn't seem to be the right word.

We will not label the previous analyses as "traditional", and **will avoid its usage in the revised manuscript along with the other word "configuration". We will simply delete this sentence.**

We labeled the previous studies that considered simplified geometries of ice/bedrock/ocean system (e.g., Hutter et al., 1983; Lambeck et al., 2003; Mitrovica and Milne, 2003) as "traditional" because these studies do not capture kilometer-scale geometric features, e.g. ice rises and rumples and rugged fjord geometries, that are critical to understand grounding line dynamics. One of our key goals is to present a method that can track grounding lines and coastlines of arbitrary geometries within a system of ice sheet, solid Earth and sea level models. To clarify this, **we will include the following in the revision:**

> ...kilometer-scale features, such as ice rises and rumples... These features complicate the required geometrical simplifications used in previous studies for ice-bedrock-ocean interface changes... Here we generalize the flotation condition for ice to present a method that facilitates a precise tracking of both grounding lines and coastlines in light of ice-ocean mass exchange and induced gravitational, rotational and deformational (GRD) response of the solid Earth and associated sea level adjustment (Gregory et al., 2019). The method is applied to an arbitrary geometry of interfaces and at all time scales.

p. 2, l. 13: Similarly, what is meant by "traditional theory for ice-bedrock-ocean interface changes"?

We believe that our response to the previous comment fully addresses this question.

p. 2, l. 25: "sea surface elevation". This is an ambiguous term; it can refer either to the mean (on some appropriate time scale) or to a quantity that varies on short time and spatial scales. There is some explanation below, but it is better to be as clear as possible when introducing the quantity $S(\omega, t)$. I think that what is meant here is what Gregory et al. (2019) call "mean sea level", a term they recommend in place of the deprecated term "mean sea surface". If $S$ is actually meant to represent the geoid, which is not quite the same as mean sea level, then this should be stated clearly.

In general, Gregory et al. (2019) is a comprehensive, carefully written reference. I suggest that the authors adopt similar terminology, paraphrasing and referring to that paper as appropriate.

We agree that Gregory et al. (2019) is the key reference here. **We will comply with their definitions throughout the manuscript.** $S(\omega, t)$ in our context generally complies with "mean sea level (MSL)" as defined by Gregory et al. (2019), but it does not account for steric and dynamic components of MSL. Specifically in our case, global mean value of $S(\omega, t)$ is given by barystatic sea level, and its spatial pattern is dictated by GRD (gravitational, rotational, deformational) response of the solid Earth to land-ocean mass exchange. **We will include the following in the revision:**

> We first introduce two fields, $B(\omega, t)$ and $S(\omega, t)$, to be the solid Earth surface (i.e., land surface or sea floor) and the mean sea level (MSL), respectively.

> Our definition of MSL complies with that given by Gregory et al., 2019): Time-mean of sea surface over a sufficiently long period so that the effects of waves, tides, or meteorologically-driven high-frequency fluctuations are

eliminated. The period of time-mean may be on the order of 20 years or longer, a timescale over which interactions between sea level and ice sheet may become important (Hillenbrand et al., 2017; Larour et al., 2019). Specifically, the change in global-mean of MSL in the present context is equal to the so-called barystatic sea-level change that defines global-mean sea-level change due to a sustained exchange of water between the land and the ocean. The evolving spatial pattern of MSL, on the other hand, is strictly dictated by the GRD response of solid Earth to land-ocean mass exchange. This definition of evolving MSL is familiar in GIA modeling wherein there is a requirement to solve the so-called "sea level equation" (Farrell and Clark, 1976).

p. 2, l. 26: What is the International Terrestrial Reference Frame, and how is it defined? Is it similar to what Gregory et al. (2019) call the reference ellipsoid?

Referee #1 also had a similar question. **See our response to comment #P2.l26 on page 7. Yes, the ITRF solutions (e.g., GRS80) are reference ellipsoids.**

p. 2, l. 32: It is stated first that $S$ is highly variable in space and time, and then it is stated that $S$ is quasi-static and does not, in fact, include short-term dynamic processes. Please use $S$ only to refer to quasi-static mean sea level, and use a different term when discussing short-term dynamics.

These sentences will not appear in the revised manuscript. We will provide a precise definition of $S$, complying with Gregory et al. (2019). **See our response to comment #p.2,l.25 on pages 20-21.**

p. 3, l. 5: "Sea level" is another ambiguous term, as discussed by Gregory et al. I suggest "mean sea level."

**This sentence will not appear in the revise manuscript.** Also, see our response to comment #p.2,l.25 on pages 20-21.

p. 3, l. 5: "represents an equipotential surface whose spatial pattern mimics the geoid." This is confusing. First, how is the geoid defined? Gregory et al. define it as the geopotential surface chosen so that the volume between the geoid and the sea floor is equal to the time-mean volume of sea water (including the liquid-water equivalent of floating ice) in the ocean. Second, what is meant by "mimics" the geoid? Does "mimics" mean "is equivalent to", or "is similar to"? If the latter, in what way does $S(\omega, t)$ differ? If $S$ is mean sea level, then it is not an equipotential surface; for instance, mean sea level has a higher geopotential on one side of the Gulf Stream or ACC than the other. (Though it could be convenient to define $S$ as an equipotential surface in areas not covered by ocean.)

$S(\omega, t)$ as we will define in the revised manuscript is familiar in GIA modeling, which requires to solve the so-called "sea level equation" (see our response to comment #p.2,l.25 on pages 20-21). This particular definition of $S$, which complies with general definition of MSL (Gregory et al., 2019), represents the equipotential surface and it differs from the geoid only by a spatial invariant for the sake of mass conservation in sea-level equation (see Tamisiea, 2011). **We will rephrase the sentence as follows:**

This definition of evolving MSL is familiar in GIA modeling wherein there is a requirement to solve the so-called "sea level equation"... MSL as strictly defined above represents an equipotential surface whose spatial pattern matches the geoid (Tamisiea, 2011), and is therefore a globally-defined field.

p. 3, l. 11: Since ocean and ice have variable density, it would be clearer to refer to $\rho_o$ and $\rho_i$ as reference densities.

Agreed. **We will rewrite it more clearly as follows:**

Here $\rho_i$ and $\rho_o$ are average densities of ice and ocean water, respectively.

p. 3, l. 12: "sea surface relative to the seafloor". I suggest "local mean sea level relative to the seafloor".

This sentence will be moved to a new section "Mass conservation in the Earth System". See our response to comment #P3.l8-15 on pages 8-9. Nevertheless, **we will rephrase the sentence as suggested:**

$R(\omega, t)$ is the local mean sea level relative to the sea floor...

p. 4, Eq. (3): Why are the ocean and land functions undefined at coastlines and grounding lines? Is it problematic not to include them in one domain or the other?

Coastlines and grounding lines are neither part of land nor of ocean. They are the interfaces between the two. However, we understand the reviewer's concern given the lack of a precise description of these interfaces. **We will now include a mathematical description of a generic land-ocean boundary, including grounding lines.** Reviewer #1 also had a similar concern. **See our response to comment #P4.l6 on page 9.**

p. 4, l. 7: Please say precisely what is meant by "connected to the open ocean". I would guess that the connected regions include marginal seas (e.g., the Mediterranean) but not inland lakes (e.g., the Great Lakes). Also, one needs an operational definition of the open ocean before defining a connection to the open ocean.

With new materials that present a mathematical description of level set function $\mathcal{F}(\omega, t)$ and its zero level set $\mathcal{T}(\mathcal{F})$ (see our response to comment #P4.l6 on page 9), **we do not need to invoke the (relatively subjective) concept of "connectivity".** The ocean function (Eq. 3 of the original manuscript) for example will be written as follows:

$$\mathcal{O}(\omega, t) = 1 \quad \text{if} \ \ \mathcal{F}(\omega, t) < 0;$$
$$= 0 \quad \text{otherwise, except when } \omega \in \mathcal{T}(\mathcal{F}).$$

Note that $\mathcal{O}$ is not defined at $w \in \mathcal{T}(\mathcal{F})$, i.e., the ocean (or land) is not defined on coastlines or grounding lines. Also, see our response to the previous comment.

p. 4, l. 17: "is connected to" is more precise than "is in direct contact with"?

As noted in our response to the previous comment, the concept of "connectivity" will be discarded altogether. **Relevant phrases will not appear in the revised manuscript.**

p. 4, l. 23: Are there published examples of frameworks (i.e., "traditional theory") that cannot handle pinning points? It seems natural to define the ice domain as in Eq. (4), and I'm not aware of frameworks with a different or less natural definition.

No doubt that Eq. 4 is the simplest and most generic definition of ice domain. And, there is nothing new about it here, compared to the previous studies. To avoid a confusion, **we will simply delete this sentence.** However, what we meant to imply referring to the previous (aka traditional) studies is that they consider simplified geometries of land, ocean and ice sheets, perhaps due to the lack of constraining data or computational resources. Consequently, these studies do not capture kilometer-scale features such as rugged fjord geometries and pinning points that are critical for accurate modeling of grounding line migration and hence ice sheet dynamics. **We will clarify it in the Introduction:**

> ... [previous] analysis of ice sheet evolution at the ice-bedrock-ocean interfaces has been given for a set of simplified geometries, owing to the lack of constraining data or computational resources,...

> A similar geometric approach is also familiar in the development of glacial isostatic adjustment theory... with migrating grounding lines and coastlines.

> ... kilometer-scale geometric features, such as ice rises and rumples, rugged fjord geometries, and uneven bedrock topography complicate the required geometrical simplifications used in previous studies...

Also, **see our response to comment #p.2,l.8 on pages 19-20.**

p. 4, l. 24: Which "employed assumption" is being referenced here? Maybe the assumption that only ice that is part of the ocean domain is included within the floating ice mask?

Correct. **We will clarify the sentence as follows:**

> Our operative definition of a floating ice mask, however, limits us from capturing the floating ice on subglacial and proglacial lakes that are not part of the global ocean (see Figure 1).

p. 4, l. 27: "the first generation of Earth system models". I'm not sure we are still in the first generation, since ESMs have been around for about a decade. Maybe "current Earth system models".

Ok. **We will use the suggested phrasing.**

p. 5, Eq. (6): Why is the grounded mask needed in this expression, if it is true that $H = H_0$ for floating ice shelves?

Given our definition of flotation height for ice (Eq. 5), ice shelves actually have smaller thickness than the flotation height. This implies that ice shelves can potentially contribute to barystatic sea level change, which obviously is not correct. Therefore we must invoke the grounded ice mask in Eq. (6). **We will state this explicitly in the manuscript.**

p. 5, l. 8: What is the referent in "it can be negative"?

It is $H_F$. **We will clarify it in the revised manuscript:**

> For grounded portions of the marine ice sheet, $H_F(\omega, t) < H(\omega, t)$. In fact, $H_F(\omega, t)$ can take on negative values...

p. 5, l. 21: What is meant by "directly affects"? Is this just the (fairly trivial) statement that some, but not necessarily all, of the net change in ice mass results in a change in ocean mass?

Yes to your second question. **We will rephrase the sentence to clarify this:**

> We assume that the net change in ice sheet mass results in the equivalent change (but of opposite sign) in ocean mass, ensuring mass conservation in the Earth System.

p. 5, ll. 21-22: "Quantifying the fraction of ice mass change that contributes to sea level" I thought that this was the main point of the section. In what sense is it analytically unapproachable and beyond the scope of the study?

We agree that these sentences are ambiguous, as also pointed by Referee #1. See our response to comment #p5.l22-23 on pages 10-11. **We will delete these sentences.**

p. 5, l. 23: "Despite the assumption"? Which assumption? If the reference is to the assumption that "the net change in ice sheet mass directly affects the ocean mass", I'm not sure there is a contradiction, but I'm not clear on the precise meaning of the assumption.

Yes, it was in reference to the quoted assumption. We agree that there is no contradiction either. **We will delete the quoted phrase and revise the sentence as follows:**

> Not all of $\Delta H(\omega, \Delta t)$ contributes to ocean mass change.

p. 5, l. 27: "...yields some error." After reading this statement, I was expecting to see quantitative error estimates later in the text. There is an illustration in Fig. 3, but is it possible to state the typical order of magnitude of the error? For instance, is it closer to 1% or 10%?

Good point. **We will include error estimates based on the results shown in Fig 3.**

p. 5, l. 28: Please say precisely what is meant by "contributes to sea level". For example, if an ice sheet loses mass, the geoid will change because of gravitational and rotational effects, but these changes aren't part of $\Delta H_S$. What is meant, I think, is the part of the ice loss that adds to the mass of the ocean, i.e. the barystatic sea-level component. If so, then barystatic SLR and related terms should be defined here or earlier.

We agree with the reviewer's assessment. **We will rephrase this sentence:**

> We define $\Delta H_S(\omega, \Delta t)$ to be a portion of ice thickness that contributes to barystatic sea level change over the period $\Delta t$.

Note that **barystatic sea level change will be defined in the beginning of Section 2**. Also, see our response to comment #p.2,l.25 on pages 20-21.

p. 5, l. 30: Since this is a central equation in the paper, I would like to see a clearer description of the physical meaning of each term, and when appropriate a derivation. I convinced myself that the first two terms on the RHS are correct, but I was not able to derive the third term or understand the physical motivation. In the text, the closest thing to an explanation is on p. 7, l. 20: "The last term in the equation accounts for the fact that fresh water density evolves during the accretion and ablation of ice, whereas the average ocean water density in the vicinity of the grounding line acts to determine the ablation height." This is confusing, in part because freshwater density $\rho_w$ is a physical constant that does not evolve. Please provide a clearer explanation and, if possible, a supporting figure.

Three bullets on page 7 of the manuscript were meant to provide such interpretations. However, we are happy to provide more information and expand on the description of those terms. Referee #1 also requested for a clearer explanation. **See our response to two of her comments #p5.l30 on page 11-12**. The interpretation of last term appearing in Eq. 7 is given in full details, with a supporting figure (Fig. 2). Given the expanded explanation, **the quoted sentence will not appear in the manuscript.**

p. 6, Figure 2: The figure and caption are helpful, especially panels a and b with the four different regimes.

Thank you.

p. 7, ll. 3ff: The text refers to three distinct "regimes", whereas Fig. 2 refers to four regimes that are defined differently from the regimes in the text. Please use the term "regimes" consistently. In the text, paragraph 1 corresponds to the first term on the RHS of Eq. (7), and paragraph 2 to the second and third terms. But as stated above, the explanation of the third term is not clear. Perhaps revise so that paragraph 2 addresses the second term and paragraph 3 the third term. Then the current paragraph 3 would become a short paragraph 4.

Indeed, we also realize that the 4 regimes defined to interpret $H_F$ (Figure 2a) may be confused with the 3 regimes defined to interpret $\Delta H_S$ (Figure 2b). To avoid this confusion, **we will label the formers as Sectors A, B, C and D, and the latter as Regimes 1, 2, and 3.**

**The description of Regimes 1, 2, and 3 in the text will be restructured as suggested. The explanation of third term appearing in Eq. 7 will be improved with a supplementary figure.** See our response to comments #p.5,l.30 on page 25.

p. 7, l. 19: Could you give an example of when the magnitude of the change in $H_F$ would be equal to the magnitude of the change in H, and when it would be less?

We appreciate the comment. Since it is much easier to demonstrate such scenarios with help of sketches, **we will provide a supplementary figure** that clearly shows the circumstances at which the equality or inequality holds in this relationship. For now, we use a sketch that we prepared in response to Referee #1 **(see comment #p5.l30 on pages 11-12)** to give example scenarios. In Figure 2 (page 12) consider that column A represents the state of ice at time $t$ and column B at $t + \Delta t$. By design, $\Delta H = \Delta H_F$. Since

ice column A is grounded and B is floating, **we can conclude that equality may hold between the two fields in Regime 2.** It is easy to demonstrate that $|\Delta H_F| < |\Delta H|$ holds in this regime. In this particular sketch (between columns A and B) $\Delta H_F$ is fixed, irrespective to $\Delta H$. If $|\Delta H| < |\Delta H_F|$ the column B would still be grounded (representing a scenario in Regime 1), **implying that $|\Delta H_F| \leq |\Delta H|$ must hold in Regime 2.** Absolute values ensure that this relationship holds in both cases: when ice transits from grounded to floating state, or other way around.

p. 7, ll. 29ff: I am confused about the difference between events in regimes 1 and 2. My understanding is that regime 1 consists of regions that are grounded at both the start and end of the simulation, whereas regime 2 consists of regions that transition from grounded to floating during the simulation. If so, this should be stated clearly. For regime 1, it is stated that $\Delta H_S$ is different from $\Delta H_F$ because of "evolving bedrock and sea level". Are bedrock and sea level not evolving in regime 2? Or is the point rather that in region 1, the ice remains grounded throughout the simulation, and therefore the entire DeltaH contributes to $\Delta H_S$, whereas $\Delta H$ differs from $\Delta H_F$ because of bedrock changes? For regime 2, it is stated that the discrepancy is due to the ?missing fraction of newly grounded or newly floating ice.? I am not sure what this means. I understand why $\Delta H_S$ differs from $\Delta H$ in this region, but not why $\Delta H_F$ differs from $\Delta H_S$.

The reviewer is correct about how we separate three regimes in Figure 3. **We will include the following sentence explicitly.**

> In order to facilitate our interpretation of the results, we separate the model domain into 3 regimes as in Figure 2b: Regime 1 consists of regions that are grounded at both the start and end of the simulation; Regime 2 consists of regions that transition from grounded to floating state over the course of simulation; and Regime 3 consists of regions that are floating at both the start and end of the simulation.

It is not that we consider the effects of evolving bedrock and mean sea level in one regime, and not in others. As shown in Eq. 7, we first define $\Delta H_S$ by accounting for $\Delta H$, $\Delta B$ and $\Delta S$. Only then we define 3 regimes for the ease of interpretation. **We agree that our interpretation of the results, particularly Figure 3c, is too brief and perhaps confusing.** The reviewer's second interpretation about the results in Regime 1 is accurate. **We will expand on our interpretation for Regime 1 as follows:**

> Only $\Delta H(\omega, t)$ contributes to $\Delta H_S(\omega, t)$ in Regime 1 (see equation 7). In this regime, $\Delta H(\omega, t)$ and hence $\Delta H_S(\omega, t)$ differs from $\Delta H_F(\omega, t)$ due to the effects of evolving bedrock and mean sea level on the latter field.

Mathematically, the difference between $\Delta H_S$ and $\Delta H_F$ in Regime 2 is simply the third term appearing in Eq. 7. **We will rewrite the sentence as follows:**

> The difference between $\Delta H_S(\omega, t)$ and $\Delta H_F(\omega, t)$ in Regime 2 essentially shows the effects of the third term appearing in equation (7). This term accounts for the misrepresentation of ocean volume due to differing densities between

freshwater and ocean water, as illustrated in the Supplementary Figure X.

The "Supplementary Figure X" will look like **Figure 2 on page 12** of this document.

p. 7, l. 30: Please say more precisely what is meant by the "customary approach of using $\Delta H_F$". Can you cite specific examples in the literature in which the ice-sheet contribution to SLR was derived from $\Delta H_F$, yielding a significant error? In the literature (beyond this specific example from Larour et al. (2019)), do the errors have a systematic sign? Are these errors prevalent in ice sheet models that include isostatic adjustment (i.e., where $\Delta H_S$ could have been computed accurately, but $\Delta H_F$ was reported instead)? Or is the problem that most ice sheet models ignore isostatic adjustment, so that they are missing a key term needed to compute $\Delta H_S$?

We meant to imply the methods that use thickness-above-flotation to estimate sea level contribution (as clearly stated on line 25 of page 7 in the manuscript). For clarity, **we will rephrase the sentence:**

> Notice the systematic error associated with the method that uses $\Delta H_F(\omega, \Delta t)$ to quantify barystatic sea level change...

Regarding second question: Most of the stand-alone ice sheet models (e.g., those contributing to SeaRISE and ISMIP6 projects) derive SLR from $\Delta H_F$. **We will explicitly mention it in the manuscript.** Unless we compare these estimates with those derived from better/improved methods (e.g., our methods or Goelzer et al., 2020), we cannot quantify/cite errors associated with these estimates. This is why Fig 3 based on a particular model run by Larour et al. (2019) is useful.

Regarding third question: As noted above there are no error estimates available in the literature, so **we cannot comment on whether they have a systematic sign.**

Regarding 4th: **It may be possible.** Goelzer et al. (2020), for example, write: "In our own ice-sheet modelling experience and from exchange with colleagues in different groups, it is not always clear how the sea-level contribution should exactly be calculated and what corrections need to be applied."

Regarding 5th: **The reviewer's speculation is absolutely correct.** Most of participating models of SeaRISE and ISMIP6 experiments are stand-alone ice models.

Also, can you state the magnitude of the systematic error? That is, what is the magnitude of the integrated error in Fig. 3 panel c, relative to the integrated value of $\Delta H_F$?

Ok. **We will report the errors in the revised manuscript.**

p. 8, Fig. 3: In addition to the 2D fields, it would be useful to show a graph consisting of time series of the area-integrated values of $H_S$ and $H_F$. This graph could show not only the total values, but also the values computed separately for regions 1, 2, and 3. Also, please cite Larour et al. (2019) in the caption.

We appreciate the comment. **We will provide such a graph as a new panel in**

**Figure 3. We will also cite Larour et al. (2019) in the caption.**

p. 8, l. 6: Here, barystatic sea level change is finally defined. I suggest introducing and defining this concept earlier in the paper. Also "ocean mass-related" is a bit vague; I suggest phrasing similar to that of N19 in Gregory et al. (2019): e.g. "the part of global-mean sea-level rise which is due to the addition to the ocean of water mass that formerly resided within the land area as land water storage or land ice." Then $\Delta R^I$, introduced below, would be the land-ice contribution, and $\Delta R^L$ would be the contribution from other land terms.

We will define this term in a much clearer way early in the manuscript. See our response to comment #p.2,l.25 on pages 20-21.

p. 8, Eq. (8): The term on the LHS includes a subscript, a superscript, and an overbar, without immediately saying what these things mean. I suggest a more gradual and systematic introduction to the notation. Also, could you explain why the denominator contains $\rho_w$ instead of $\rho_o$? At first, I assumed that the denominator represents the mass of the ocean, but I think the reason for $\rho_w$ is that we are converting a mass of fresh ice into an equivalent ocean volume, ignoring halosteric effects. Again, a more detailed physical explanation would be helpful.

We will restructure the materials (p.8,l.6 through p.9,l.23 of the original manuscript) under the new section "Mass conservation in the Earth System". We will go from general description of $\Delta R$ and its components (Eq. 9) to specific in-depth discussion of the component of interest, i.e. $\Delta R_C^I$. **See our response to comment #p8.l15 on page 14.**

$\rho_w$ is used to convert the net change in ice mass to freshwater volume that is added to the ocean. **We will state this explicitly in the revised manuscript:**

> Note that we divide the net change in ice mass by $\rho_w$ to quantify freshwater volume that is added to the ocean.

p. 8, Eq. (9): This equation introduces several more terms without preamble, and the reader has to study the following paragraph carefully to translate each term. Please rewrite in a way that is gentler for the reader.

We appreciate the comment. **We will restructure the relevant materials as advised.**

p. 9, ll. 7-9. I am not clear on the meaning of the third and fourth ("past") terms, and how these terms change the ocean mass. I understand that past ice-sheet changes affect sea level through ongoing glacial isostatic adjustment, but isn't GIA included in term 6, the vertical-land-motion term?

Third and fourth terms are indeed related to GIA processes. They modulate not only the present-day bedrock elevation but also the geoid field, hence their imprints are imbedded both in $\Delta B$ and $\Delta S$. The last term $\Delta R_V$ is reserved for vertical land motion due to non-GIA processes, such as earthquakes, landslides. **We will make it explicit:**

> $R_V(\omega, \Delta t)$ represents a component due to vertical land motion caused by

non-barystatic processes, such as earthquakes and landslides.

A more general comment: It could be easier for the reader if the text were organized from general to specific, instead of specific to general. That is, first define the various kinds of sea level rise, introduce notation, and state the various source terms. Then state that this paper is focused on $R_C^I$, as computed in equation (8) based on $\Delta H_S$. Finally, show how to compute $\Delta H_S$. This would be a fairly major rewrite, and I don't want to be too prescriptive, but it is challenging for readers to introduce basic concepts just a page or two before the conclusions.

We really appreciate this comment. We believe that the central focus of our paper is a new field $\Delta H_S(\omega, \Delta t)$ for estimating ice sheet's contribution to sea level (Eq. 7 of the manuscript) and a newly-added generic level set $\mathcal{T}(\mathcal{F})$ for tracking land-ocean interfaces in Earth System (see our response to comment #P4.l6 on page 9). **We are therefore reluctant to restructure the entire manuscript as suggested. However we will introduce a new section "Mass conservation in the Earth System" at p.8,l.6 of the original manuscript in order to put $\Delta H_S$ and $\mathcal{T}$ in a broader context of sea level change.** See our response to comment #p8.l15 on page 14.

p. 9, l. 16: When $\Delta M$ is described as a mass-conserving field, is this equivalent to saying that its global integral is zero? Also, is it strictly true that $\rho_i \Delta H_S$ is equal to the change in ice mass at each location? Here, I'm wondering about the third term in Eq. (7); is that term associated with a change in the local mass per unit area?

**The answer to all of these questions is "yes".** The second term represents ice mass change per unit area, whereas the third represents the mass change per unit area within the ocean domain. **We will explicitly mention that the global integral of $\Delta M$ is zero:**

We may define a mass conserving field

$$\Delta M(\omega, \Delta t) = \rho_i \ \Delta H_S(\omega, \Delta t) + \rho_w \ \Delta R_C^I(\omega, \Delta t) \ \mathcal{O}(\omega, t + \Delta t),$$

such that its global integral is zero.

p. 9, Conclusions: As mentioned above, it would be helpful to quantify the benefits of the new methods, e.g. by estimating the errors associated with the older methods.

**We will include the following:**

For the case considered in this study, we find that the existing method underestimates the barystatic sea level change by about X%.

**Minor corrections**

p. 1, l. 6: "and include the ice shelves and adjacent ocean mass." The phrasing is awkward. Please use a parallel grammatical construction.

We will completely rewrite the second half of the Abstract, highlighting the main goals of the paper and their implications. **This sentence will not appear in the revised version. See our response to comment #p1.Abstract on pages 18-19.**

p.1, l. 7: "is" → "can be"?

p. 1, l. 15: "grounding line" → "grounding lines"

p. 1, l. 17: Delete "involved"

p. 2, l. 2: "first order" → "first-order"

p. 2, l. 28: Delete "for"

p. 3, l. 5: "refer" → "refer to"

p. 4, l. 30: "ice sheet driven" → "ice-sheet-driven"

p. 4, l. 32: "far field" → "far-field"

p. 5, l. 2: Add "the" after "estimate"

p. 5, l. 20: "farfield" → "far-field"

p. 5, l. 26: Add a comma after "below"

p. 6, l. 7: "predict" → "predicts"

p. 9, l. 15: Insert "the" before "ice sheet"

p. 10, l. 6: "analyses" → "analysis"

We will implement all of these suggestions.

**Referee #3**

The paper by Adhikari et al. presents a formalism to calculate the contribution of dynamic ice sheets to mean sea level by considering a changing sea-level, bedrock and land-ice-ocean mask. Compared to standard methods the contribution to mean sea level is computed from ice thickness and not from the height above flotation. The formalism is valid on all timescales, but the full benefit is on longer timescales. As there are currently huge efforts to include dynamic ice sheets in Earth system models the presented paper is a good reference.

Generally, I found the paper well written and structured with illustrative and clear figures. The paper is worth publishing in The Cryosphere but as the focus is rather technical it would fit much better to GMD. The current form of the manuscript lacks a bit in the presentation and clarity. Therefore, I have a few suggestions that should be addressed in a revised version.

We appreciate the positive and constructive review by the referee. Find our response to individual comments below. **For questions that are similar to those from other referees, we kindly refer to corresponding responses that are already documented.** Proposed changes to the manuscript are in red font.

1) I found the title a bit misleading to the content of the paper. I am missing the definition of "mass conservation" in the text. I also would like to see (e.g. with an example), if the formalism is mass-conserving (or better mass-conserving than traditional or other methods). Also, the only example in the manuscript was about calculating SLE relevant thickness changes rather than mass-conservation.

**We will suitably modify the title of the manuscript.** Referee #2 also had a similar concern. **See our response to comment #p.1.Title on pages 17-18.** As promised there, **we will state what mass conservation means in the context of our formalism, and will provide example cases where mass conservation is violated.** In terms of results, there is nothing really to show to justify mass conservation except to state that Eq. 10 takes $\Delta H_S$ from Eq. 7 and $\Delta R_C^I$ is computed using a global GRD (gravitational, rotational, deformational) solid Earth model, such that the global integral of $\Delta M$ equals zero. See our response to comment #p.9,l.16 on page 29. **We will add a mathematical overview of physical basis of GRD that determines the spatial pattern of** $\Delta R_C^I$ (i.e., spatial pattern of $\Delta B$ and $\Delta S$ and hence other derived fields and level sets). See our response to comment #p8.l15 on page 14 as well.

2) The Introduction should refer to Goelzer et al. (2020). You do very quick comparisons to Goelzer et al. (2020) on page7,line6 and 23, but I think the Introduction should clearly say what you are doing differently and why. This could be a motivation to release your new formalism. An appropriate discussion to Goelzer et al. (2020) is also missing. Additionally, from the Introduction it was not really clear to me what is actually wrong with the traditional methods, the order of error on SLE they could introduce and what you are now aiming to improve.

We will summarize the work of Goelzer et al. (2020) in the Introduction. We will also

point out the key limitations of existing methods. **See our response to General Comments by Referee #1 on pages 2-3.** As for errors associated with the existing methods, **kindly see our response to Referee #2's Summary on page 17.**

3) Not sure if this could be really addressed, but it would be interesting to estimate the errors (i.e. traditional versus your formalism) of current projections of SLE from the two big ice sheets e.g. within the ISMIP6 framework (Antarctica: Seroussi et al., 2020; Greenland: Goelzer et al., 2020b). Or from current remote sensing products like IMBIE (Shepherd et al., 2019). My point here is, that I would get a better feeling for the error on e.g. different timescales, regional settings and how it differs for Greenland and Antarctica. The example based on the Larour et al. (2019) simulation is very helpful (see also my comment to P7,l29ff) but very specific – and, as I understood – not in line with current projections efforts. I do not strictly insist that you show an error for ISMIP6 or IMBIE, but as also commented below, I would like to have a better error estimate and its impact on current research (compare Fig. 3 in Goelzer et al. (2020)).

We appreciate the comment. This sounds like a great idea, but due to the lack of enough information in these dataset and model results, i.e. $\Delta B(\omega, t)$ and $\Delta S(\omega, t)$, we really cannot evaluate $\Delta H_S(\omega, t)$ (see Eq. 7). Computing $\Delta B(\omega, t)$ and $\Delta S(\omega, t)$ is out of the scope of this theoretical paper. That said, based on Larour et al. (2019), we have shown the results to highlight the improvement that our method makes over the existing method (Fig. 3). **We will add a new panel that compares time series of barystatic sea-level change (see Eq. 8) due to $\Delta H_S$ (our method) versus $\Delta H_F$ (existing method).** See also our response to comment #p.8,Fig.3 on pages 27-28.

**Minor comments:**

P1,l8: I have not found in the text, which computational strategies you have simplified. What do you mean with computational strategies?

We will completely rewrite the latter half of the Abstract, and this sentence will not appear anymore. **See our response to comment #p.1.Abstract on pages 18-19.**

P2,l11: Why are "floating ice shelves, ice rises and rumples, and retrograde bedrock slopes" complex features?

We will slightly modify the phrasing:

> ...kilometer-scale geometric features, such as ice rises and rumples, rugged fjord geometries, and uneven bedrock topography...

We will not call these features complex. What we really meant to imply in this sentence, however, is that **these kilometer-scale geometric features are not generally captured in the previous studies of evolving ice-bed-ocean interfaces,** although they are absolutely essential to understand the dynamics of marine-based ice sheet. See our responses to comment #P2.l12 on page 5 and comment #p.2,l.8 on pages 19-20.

P2,l13: What is "traditional theory for ice-bedrock-ocean interface changes"?

**This question is fully addressed in our response to comment #p.2,l.8 on pages 19-20.** See also our response to comment #P2.l13 on page 6.

P2,l18: "... that can be straightforwardly employed in any Earth System model"?. I think it would be worth to mention (somewhere), if this new formalism could be adopted to other disciplines (e.q. remote sensing, standalone ice sheet modelling). In the current form, it sounds the formalism in only valid/applicable in ESMs.

By design, the proposed formulation is very generic and we do not see why it cannot be applied for both (stand-alone) modeling and observational data. For example, definitions of land, ocean, and ice domains, as well as their interfaces (see newly added materials on P4.l6 on page 9) are valid in both realms. However, due to lack of modeled/observed data for some fields (e.g., $\Delta B$ and $\Delta S$) some aspects of the formalism (e.g., Eq. 7) are best suited for ESMs. **We will mention it explicitly in the manuscript.** Also see our response to Referee #1 on page 3.

P4,l23: I cannot see from Eq. 4 that your new setup diverges from traditional approaches. Eq. 4 is a very common equation to define an ice-mask. On page5,line 25 you give another example of how the traditional setup differs from your setup. Maybe outlining the differences could be gathered together.

Agreed that Eq. 4 is the simplest and the most generic mathematical description of ice domain. **We will simply delete this sentence.** Also see our response to comment #p.4,l.23 on page 23.

Regarding "traditional" versus our setup: **See our response to comment #p.2,l.8 on pages 19-20.**

P5,l8: I don't understand this sentence. What is negative?

$H_F$ is negative. We will clarify it in the revised manuscript. **See our response to comment #p.5,l.8 on pages 23-24.**

P5,l9: "...hence contribute to sea level inversely." Maybe say sea-level drop/fall to avoid confusion.

Agreed. **We will rephrase the sentence as suggested:**

> ...this region, when physically connected to the ocean by oceanic water, can take up water and contribute to sea level fall.

P5,l10: I am not a native speaker, but are both "evolving" really needed?

**We will rephrase the sentence.**

P5,l25: Can you add a reference to a Figure after "show"?

We believe that line 26 is being refereed here, not 25. Considering this and Referee #1's comment (#p5.26 on page 11), **we have decide to remove the phrase containing the quoted word. The new sentence will read:**

This simplistic approach yields some error, particularly when evolving bedrock and mean sea level are considered (Larour et al., 2019; Goelzer et al., 2020).

P7,l4: "...and the elevations"?

Should be "at the elevations" as is in the manuscript. But, we understand the reviewer's confusion. **We will simplify the sentence as follows:**

Where ice remains grounded at both times $t$ and $t + \Delta t$, all of $\Delta H(\omega, \Delta t)$ contributes to barystatic sea level change.

P7,l29ff: I would first describe the Larour et al. (2019) setup and then present the results. Can you give an integrated value (e.g. SLE) for both approaches to get a better feeling for the error? The simulations were run over 500 years. Why do you choose to present the results after 350 years?

Ok. **We will restructure the paragraph as advised.** There is no particular reason as to why we chose 350 years. **We will use results after 500 years of model run** in the revised manuscript. **We will also provide the SLE (aka barystatic sea level change) estimates for both approaches.**

P9,l6ff: The following paragraphs and Equations appeared very suddenly and without introducing their purpose. According to the title, I would expect Eq. 10 (the massconserving field M) is the main point in your paper. But this is not illustrated and somehow contradicts with your statement in the conclusion (p9,l31-32); here you say $\Delta H_s$ is the main point. This is perhaps personal matter, but I found it a bit brutal to stop the results of the paper with these equations. An illustrative example on "implication of this new geometrical setup for sea level and solid Earth loading studies" (your comment on p2,l20) would make more sense to me.

We agree with the reviewer's assessment. Other reviewers also had similar opinion. As noted in our response to comment #p8.l15 on page 14, **we will introduce a new section "Mass conservation in the Earth System"** at p8.l6 of the original manuscript. We will begin by introducing $\Delta R$ and many contributing processes (Eq. 9). We will then specify that we are interested in $\Delta R_C^I$ and show its relationship to $\Delta H_S$ (Eqs. 8 and 10). We will then highlight the need for mass conserving model of solid Earth in order to compute the spatial pattern of $\Delta R_C^I$ and many other derived fields (including $\Delta H_S$ itself) and level set $\mathcal{T}(\mathcal{F})$. **See our response to comment #p.1.Title on page 17-18.**

In conclusion **we will equally highlight two of our key contributions:** $\Delta H_S(\omega, \Delta t)$ for estimating ice sheet's contribution to sea level and $\mathcal{T}(\mathcal{F})$ for tracking land-ice-ocean interfaces in the Earth System. **We will stress that their self-consistent solutions are only possible through a mass conserving system of ice sheet, solid Earth, sea level models.**

Fig.3: What is the grey line? And in the caption: Is "conventional"=="traditional"? I guess yes. Please use the same wording in the whole text. Eq. 3 and 4: consider rewriting with "latex-cases".

The gray line shows the ice margin. **We will mention it in the caption.** The quoted words are equivalent. **We will accommodate all of suggested changes.**

---

## Author Response (AR1)

June 30, 2020

Dear Editor,

We thank you for the opportunity to revise the manuscript and for generously granting a month-long extension of submission deadline. We thank all three reviewers for their positive and thoughtful comments that have led to a greatly improved manuscript. We are happy to report that we have addressed all of reviewers' comments and revised the manuscript suitably. Much of the reviewers' criticisms are on the clarity and focus of the manuscript, which we have considered sincerely and made some significant reshuffling of the text. Here is the list of key changes we have made in the revisions.

- 1. We have changed the title so that it better aligns with the key goals of the paper.
- 2. We have spelled out the key goals of the paper both in the Abstract and Introduction.
- 3. We have added in the Introduction more description of the existing methods and their limitations and have contrasted our goals against them.
- 4. We have added mathematical description of a generic level-set function that can track both the grounding lines and coastlines in a seamless manner (Section 2.1).
- 5. We have explained the key mathematical terms concisely, with their physical interpretations in light of space observations when possible.
- 6. We have added new materials (about 2.5 pages, with 1 figure) in the Appendix to facilitate the interpretation of some mathematical terms.
- 7. We have introduced a new section (Section 4) to place our formalism in a broader context of sea-level change and mass conservation in the Earth System.
- 8. We have added new figure panels to demonstrate the level of improvements possible by employing the new method of estimating ice sheets' contribution to sea level, which is significantly large (on the order of 10-15%).
- 9. We have inserted several subsections to make the paper more accessible to readers.

Let me know if you have any question or need further information.

On behalf of co-authors, Surendra Adhikari Jet Propulsion Lab, Caltech

**Referee #1**

**Summary**

The paper presents a formalism to geometrically interpret changes in ice sheets and underlying bedrock and their combined effect on the ocean and sea level. The approach defines two distinct domains (land and ocean) that can both intersect areas of ice cover. It then traces ice and bedrock changes and transitions between different domains to determine the sea-level contribution of the ice sheet.

**General comments**

The paper is well written, clearly structured and deals with the important question of how to calculate the sea-level contribution of a marine ice sheet, among others. I believe it would make an interesting contribution to The Cryosphere given that the points raised below are addressed adequately.

We thank the reviewer for a positive and constructive review. Please find our response to individual comments below. The changes made in the revised manuscript are pointed by the **bold text**.

One of the main conclusions of this paper reads very similar to the one in Goelzer et al. (2020), both papers proposing an alternative to conventionally calculating the sea-level contribution of marine ice sheets based on volume above flotation. It seems important to clarify what the similarities and differences are. The results presented by Goelzer et al. (2020) imply that ice and bedrock changes have to be considered together at least in any place where ice could ground over the course of the experiment. This is the direct consequence of the claim that the sea-level contribution calculated from one point in time to another should be independent of what happens in between. Since the example that is put forward (see their Fig 2a and related text) matches with regime 3 here, there seems to be a direct disagreement between the two approaches: bedrock changes are taken into account in their case, but not here. This may point to a flaw in the approach that should be clarified and discussed. If bedrock changes are not considered as part of the ice sheet change in regime 3, what other component is taking it into account (if any) and how does the domain separation between those components work? If both approaches are not compatible, why and under what circumstances do they differ? What would needs to be changed to make the two approaches compatible? Are the two approaches addressing a different modelling framework, which explains the differences?

We break this question into two parts. (1) What are the similarities and differences between our work and the recent work by Goelzer et al. (2020)? (2) Why are Figure 2a of the Goelzer paper and our Regime 3 (as defined in our Figure 2b) inconsistent?

The goal of the Goelzer paper was to provide a correction to a common approach of estimating sea-level contribution from ice sheets. The approach, based on the concept of ice height above flotation (HAF), predicts an incorrect sea-level contribution when the ice-sheet model accounts for evolving bedrock and sea level, especially in the marine portions of the ice sheet. In particular, Goelzer et al. (2020) provide a correction for the effects of bedrock elevation change and externally-forced sea level in stand-alone ice-sheet models (page 2 lines 24-27). Our goal here is two-fold: (1) to formulate a \*new\* field – rather than a correction term – that can be utilized to accurately quantify ice sheets' contribution to sea level by accounting for the effects of evolving bedrock and sea level driven by any geophysical processes (page 2 lines 28-29); and (2) to develop a generic level-set for accurate tracking of both the grounding lines and coastlines in a seamless manner (page 2 lines 16-18). Our formalism can be applied to a whole suite of modeling architectures: from stand-alone ice-sheet models (e.g., many of ISMIP6 participating models), to models that account for isostatic adjustment of bedrock (e.g., Le Meur and Huybrechts, 1996; Pattyn et al., 2017) or a self-consistent GRD (gravitational, rotational, deformational) response of the solid Earth (e.g., Gomez et al., 2013; Larour et al., 2019). In the latter set of models, our formalism ensures mass conservation in the Earth System by exchanging mass between the land and the ocean, accounting for induced GRD response of solid Earth, and adjusting the ocean area through grounding lines and coastlines migration, simultaneously (page 2 lines 29-35).

Regarding the second question, it appears that we were not clear enough in the manuscript in terms of explaining the physical significance of three regimes. To facilitate discussions, we compile relevant figures below (Figure 1). Based on the evolving ice thickness  $\Delta H$ , bedrock elevation  $\Delta B$ , and mean sea level  $\Delta S$ , we classify 3 regimes of the ice sheet. In Regimes 1 and 3, grounded ice remains grounded and floating ice remains floating, respectively, over the considered time period  $\Delta t$ . Regime 2 only captures the portion of the ice sheet that experiences transition from grounded to floating state, or the reverse, over the course of  $\Delta t$ . Now let us interpret Figure 2a of Goelzer et al. (2020) from the lens of the three regimes we've classified. Over the period  $\Delta t = t_4 - t_1$  (see top panel in the figure below), as correctly pointed out by the reviewer the floating ice remains floating (equivalently, our Regime 3) and hence the column of ice does not contribute to the barystatic sea-level change during this period. Even though  $\Delta B$  and  $\Delta S$  do not appear to play any role in this regime (as they do not appear explicitly in equation 10), the fact is that they determine the new grounding line position and hence the domain of Regime 3. The reviewer's remark that we do not consider the effect of the evolving bedrock elevation in Regime 3 is therefore not accurate. If we consider  $\Delta t = t_2 - t_1$ , on the other hand, the column of ice does contribute to the barystatic sea level because the then-floating ice is now grounded (equivalently, our Regime 2), and we will have to consider the effect of  $\Delta B$ and  $\Delta S$  explicitly while estimating the sea-level contribution of the ice column over this shorter period. This is consistent with the interpretation made by Goelzer et al. (2020).

While the first few words in the abstract seems to say that this paper is about representing ice sheets in models, the mention of geodesy later may suggest that observations of ice sheets are equally addressed. If the aim is indeed modelling, I think this should be made clear and a clearer distinction be made from observations. If both should be addressed at the same time, I suggest to make sure that the presented formulations make sense in both realms.

By design, the proposed formalism is very generic and it can be used for applications with modeling, observations or combination of models and observations. Many aspects of our

Figure 1: Top: Figure 2a of Goelzer et al. (2020). Bottom: Figure 2b of our paper.

formalism including a generic level set for coastlines and grounding lines (Section 2.1) can be used for both modeling and observational applications (page 2 lines 18-19). However, due to lack of observed data for some fields (e.g., change in bedrock elevation beneath the ice sheet), some other aspects of the formalism (e.g., Section 3.2) are more suited for model-based studies (page 2 lines 30-35), that too for Earth System models that seek to conserve mass (page 1 lines 9-10).

While I appreciate the formal description of the case, I miss better guidance of the reader through what is a difficult problem to understand and visualise. Recurrent redefinitions of variables (see example S(w,t) below) should be avoided, individual terms in the equations should be better explained and examples should be given where possible. This particularly applies to cases where the formalism uses familiar concepts and applies them to something else (e.g. floatation condition for ice applied to define the ice free coastline).

Based on this and similar comments from two other reviewers, we have provided more precise definitions of the key terminologies and expanded descriptions of individual terms appearing in the key equations. You will find it in our responses below. We appreciate this comment, as it greatly helps to improve the focus and clarity of the manuscript.

For understanding and reproducing the results it would be useful to provide access to the data and tools used to produce the results and plots in Figure 3. Please consider making the geometry and scripts available.

Agreed. We have made the data and Matlab script publicly available at https://doi.org/10.7910/DVN/9LUJTD (page 15 line 1).

**Specific Comments**

P1.11 Not all ESMs include ice sheet components. Reformulate.

Agreed. We simply write "...components of the Earth System." (page 1 line 1).

P1.12 The connection between ESMs and geodesy is not clear to me. E.g. observation don't exist for ESM paleo simulation where the formalism should also hold.

ESMs are intimately tied to geodesy, even if these models are not constrained by paleo data. Some of the fields considered in our formalism such as mean sea level (MSL) or geoid and bedrock topography are defined relative to a common geodetic datum, such as the International Terrestrial Reference Frame. These fields essentially describe the "shape" of the Earth, and modeling and measuring of which is the very definition of geodesy. Also note that some geodetic observing systems relevant for ESMs have been in place for more than a century. Earth rotation and gravity fields were measured during the first half of the 20th century, for example.

P1.15 "grounded and floating masks" suggests a modelling perspective, but "as viewed from space" relates to an observational dataset. What is the perspective of this paper?

As noted above, our formalism is generic and applied to both modeling and observational realms, although some aspects of it are more suited for model-based studies due to the apparent lack of observational data (page 1 lines 9-10). We have revised the referred sentence (page 1 lines 2-5).

P1.15 "Here we present ...". The subject in this sentence is not clear. Reformulate.

Toward more focused and impactful abstract, we have completely rewritten its second half (page 1 lines 5-10). As a result, this sentence does not appear in the revised manuscript.

P1.L13-15 This is clearly true for simulations of the Antarctic ice sheet, but not really for the Greenland ice sheet, which is dominated by surface mass balance changes. Similar cases may also exist during other climates with ice sheets that were mainly land based. Suggest to reformulate.

Agreed. We now write "...numerical model of marine ice sheets..." (Page 1 line 16).

P2.12 "Defining geometry". Do you mean "Defining the bedrock"? How could the geometry be defined upfront for an intercomparison when the models are supposed to produce an evolving geometry?

We agree that this statement is ambiguous. We have deleted it altogether. The removal of this sentence does not break the flow of the write-up.

P2.15 What do you mean with "basic configuration setup" and "Similar setups"? Could you describe this in more detail?

We have rephrased the sentences in a clearer way (page 2, lines 5-9). The quoted phrases do not appear in the revised sentences. See our response to Referee #2's comments #p.2, l.8 on page 17 and #p.4, l.23 on page 19 of this document.

P2.112 Why are "floating ice shelves" and "retrograde bedrock slopes" complex features? They occur in the very simple flowline model setups you may be referring to above. Clarify.

Agreed. We have replaced the quoted phrased by "rugged fjord geometries" and "uneven bed topography". These kilometer-scale features (**page 2 line 11-15**) are "complex" relative to the simplified geometries considered in previous studies of mechanical analysis of grounding line migration (e.g., Hutter, 1983; Lambeck et al., 2003). We have avoided the usage of word "complex" in the revised manuscript.

P2.113 What is the "traditional theory for ice-bedrock-ocean interface changes"? Clarification needed.

We meant to imply the previous studies that considered simplified geometries (e.g., Hutter, 1983; Lambeck et al., 2003; Mitrovica and Milne, 2003), lacking kilometer-scale features such as pinning points and rugged fjords perhaps due to the lack of constraining data or computational resources (page 2 lines 5-9 and lines 13-15). See also our response to Referee #2's comments #p.2,1.8 on pages 17 of this document. We have excluded the usage of the phrase "traditional theory" in the revised manuscript.

p2.l23 "setup"  $\rightarrow$  maybe "interpretation"?

The quoted word does not appear in the revised manuscript. We simply write "To begin our discussion, we consider a spherical planet..." (page 3 line 10).

p2.124 What is the difference between "glaciers" and "ice sheets" in your description? Clarify if the two terms are interchangeable or distinct. If the latter, what sets them apart in your formalism?

The formalism does not need to distinguish between glaciers and ice sheets. In fact, we prefer a generic term such as "distributed ice domains". We use glaciers and ice sheets to give a sense to the readers of what we meant by distributed ice domains (page 3 line 14).

p2.124 I found the upfront separation between land and ocean confusing for your context, because it is not intuitive where that separation is to be made for a marine-based ice sheets. The definition what is to be considered land and ocean comes too late. I suggest to make that clearer much earlier.

Agreed. We now make it explicit upfront in the Section 2 (page 3 lines 11-15).

P2.125 Same for S(w,t), defined here first as the sea-surface elevation. How should we think of S for an ice shelf? Why not start with defining S as the geoid as you do later.

We agree that the definition of mean sea level (MSL) S is indeed confusing. We have clarified it in reference to Gregory et al. (2019) on **page 3 lines 22-33**.

P2.126 I admit, I had to look up what the ITRF is. For other readers not familiar with it, you may want to add a sentence or two to say what the ITRF provides. In practice, if I use Bedmachine data, is it registered on the ITRF or are you suggesting this is something the

user would have to take care of herself?

ITRF is the standard reference frame defined by the International Earth Rotation and Reference Systems Service (www.iers.org). It is being updated every few years. Glaciological datasets (e.g., surface and bedrock DEMs) are intimately tied to a reference frame. BedMachine v3 data, for example, are referenced to the WGS 84 ellipsoid. This and ITRF's recommended ellipsoid, GRS80, should not differ from each other by more than a few centimeters. This difference is much smaller than the uncertainties in BedMachine data. It is also possible to project data from one ellipsoid to another.

We have decided not to specifically mention ITRF in the revised manuscript (whose description would divert the flow of the paper), because the theoretical formalism presented here is valid as long as both the bedrock and MSL are measured with respect to the same reference frame (page 3 lines 20-21).

P2.131 So far  $S(\omega, t)$  is defined as sea-level. As such, any case B > S is not well defined. The interpretation of S as the geoid must come earlier for this to make sense.

The MSL, as we have defined in this study (page 3 lines 22-30), has the same spatial pattern as the geoid and hence is a global field (page 3 lines 32-33).

P2.132 You say here that  $S(\omega, t)$  includes high-frequency noise and variability, but on the next page you want S(w, t) to refer only to the quasi-static component of the sea-surface. Why not introduce S directly as the quasi-static component of the geoid, rather than going through three redefinitions along the way (sea surface  $\rightarrow$  quasi-static sea-surface  $\rightarrow$  sea-level  $\rightarrow$  geoid).

We appreciate the comment. We believe that this issue is fully addressed above. See **page 3 lines 22-33** for the definition of MSL.

P3.11 Remove "changing" before interactions?

Agreed. See page 3 line 24.

p3.17 What is the "interior of marine ice sheets"? Clarify

We meant to imply that MSL is a globally-defined field (page 3 line 33, see also Figures 2a-b). This sentence does not appear in the revised manuscript.

P3.19 Say and explain what  $F(\omega, t)$  means. Traditionally it determines if the ice is floating. But you seem to extend it to locations with H = 0? Maybe it would be worth it to mention that.

 $F(\omega, t)$  is indeed a generalization of the concept of flotation condition for ice based on the principle of hydrostatic equilibrium (page 4 lines 1-6, see also line 9).

p3.113 Please define what "open ocean" means and what "contact with the open ocean" means. This definition comes too late in the manuscript. Does w have to be part of  $\mathcal{O}$ ? Maybe we need the definition of  $\mathcal{O}$  already before this part not on page 4?

We simply write "the ocean" consistently throughout the manuscript. We have defined it upfront in the Section 2 (page 3 lines 11-12). To avoid the ambiguity, we have provided a detailed mathematical description of a generic level set for seamless delineation of land-ocean-ice interfaces (page 4 line 16 – page 5 line 14). Also see our response to your comment #p4.16 on pages 8-9 of this document. The quoted words do not appear in the manuscript. The mathematical definition of  $\mathcal{O}$  (equation 5) remains where it originally was, because it relies on the definition of  $F(\omega, t)$ .

p3.114 A more obvious definition of a generalised coastline for me, that also exists in presence of a marine-based ice sheet would be the point where the bedrock and the geoid intersect (1/2 in Fig2). That doesn't help for your formalism, but it goes to show that it is not immediately obvious to think the coastline at the grounding line. Better guidance needed.

We would like to respectfully disagree with the reviewer on this point. We believe that our formalism, which treats grounding lines and coastlines as a seamless level set of  $\mathcal{F}(\omega, t) = 0$ , makes it appealing for Earth System models. See our response to comment #P4.16 on pages 8-9 of this document. The reviewer is suggesting the boundary between Sector A and Sector B to be the coastline (see Figure 2a). Unlike the grounding line, tracking the migration of this boundary neither improves our understanding of ice-sheet dynamics nor helps us determine an evolving ocean-surface area, for example.

P3.18-15 I found this paragraph difficult to follow. You start with reference to Fig. 2a, where our focus is on the left hand side and with  $F(\omega, t)$  which suggests it is about ice. But then you describe R(w,t) and the coastline, which are difficult to visualise in a place with an ice shelf. It may help to guide the reader by being explicit about the two "generalisations" that take place here: floatation criterion (for ice)  $\rightarrow$  definition of the coastline (everywhere). grounding line (for ice)  $\rightarrow$  coast line (everywhere).

We have already addressed most of the issues noted here. Also see our responses to your comments #p3.19, p3.113 and p.411 on pages 7-8 of this document. We have moved the definition of relative sea level  $R(\omega, t)$  to Section 4 (page 12 line 8). With these changes, we believe that the paragraph should be clearer and easier to follow (See Section 2.1).

P4.11 I don't see why there could not be a grid point in a model with B = S and H > 0. A glacier terminating on land or on a sill exactly at sea-level? Please clarify.

We are glad that the reviewer raised this point. Indeed, there is a subtle assumption that goes into our interpretation of equation (1). This has now been clarified (page 4 lines 10-15). Since the assumption is reasonably valid (page 4 line 14), no "grid point" exits in a model with B = S and H > 0 (page 5, line 16).

P4.16 The fact that neither O nor L are defined at the grounding line seems problematic. How can your formalism be mass conserving when grounding line grid points in an ice sheet model are not part of these masks? How do you track the grid cells that fulfil this condition, do they form a separate category? Why would it not matter to consider them?

O and L are complementary fields such that their surface areas make up the total area of

Earth's surface. The coastlines and grounding lines form their own level set, with no surface areas, which can be tracked straightforwardly. As such, there should not be any "cells" outside both O and L, and therefore there should be no problem regarding mass conservation. This has been now clarified in the manuscript (page 5 lines 21-26). We agree that we did not provide a mathematical description of the coastlines and grounding lines in the original manuscript, which is now included in terms of a level-set function in the revised version (page 4 line 16 - page 5 line 14).

P4.19 Remove "deep" and "well". I suppose the condition could also be true for shallow troughs with bathymetry moderately below sea-level.

**Agreed. See page 4 lines 6-7.**

P4.110 While I understand the use of this connectivity concept in your formalism, I find it problematic in practice. It means that small changes in ice or bedrock can lead to very large changes in O and L. In an unfavourable configuration, the short term grounding and ungrounding of a critical point could e.g. switch an entire system of connected fjords on and off.

With a newly-added mathematical description of a level-set function  $\mathcal{F}(\omega, t)$  (page 4 line 16 – page 5 line 14), the (relatively subjective) concept of "connectivity" does not need to be invoked anymore. However, our formalism still supports the idea of potentially having large changes in O and L due to relatively small changes in bedrock or mean sea level. And, this is precisely what happens in reality (at least from mass conservation perspective) and it should be an essential feature in paleo simulations of a system of ice sheets, solid Earth and sea level.

p4.118 With the above, combining the grounded and floating ice masks leaves a hole at the grounding line. Is this desired?

Again we understand the confusion created by the lack of mathematical description of grounding line level set, which has now been included (page 4 line 16 - page 5 line 14). Adding grounded and floating ice masks to grounding line level set would leave no hole in the ice domain. See our response to your comment #P4.16 on pages 8-9 of this document.

P5.113 The good typically changes first, then the bedrock. Maybe re-order in the sentence.

We would like to respectfully disagree with the reviewer. For a given relaxation frequency that is relevant for the present study (with periodicity of decades or longer), both geoid and bedrock elevation evolve simultaneously.

p5.l22-23 I am confused about this sentence. Isn't "quantifying the fraction of ice mass change that contributes to sea level" exactly what you are doing by defining  $\Delta H_S$  below? Reformulate?

We meant to imply that not all of ice mass loss makes it to the ocean. A fraction of it, for example, may get impounded in proglacial lakes. However, the quoted statement is in direct conflict with the assumption that "the net change in grounded ice mass results in the equivalent change in ocean mass..." (page 7 lines 12-13), so we have simply removed

the sentence. We thank the reviewer for catching this apparent inconsistency. The reviewer is correct that  $\Delta H_S$  (well, in fact, its one component  $\Delta H_M$ ) provides the accurate estimate of ice mass loss (page 7 equation 10 and the text that follows).

p5.23 "the assumption" appears three sentence back, maybe refer to it more specifically.

The in-between sentences have been deleted (see our response to the previous comment). This should fix the issue noted by the reviewer.

p5.23 Remove "all the time".

Agreed. See page 7 line 13.

p5.26 "As we show below". This has been shown before by others (see references). Reformulate to avoid confusion.

Agreed. We have removed the quoted phrase (page 7 lines 24-25).

p5.130 Can you please explain what the three terms mean physically. E.g. the first term accounts for thickness changes of ice that is and remains grounded ...

To facilitate easier interpretation, we have restructured the definition of  $\Delta H_S$  (equation 10) and split it into two parts: the component  $\Delta H_M$  that modulates both the mass and volume of the oceanic water and induces the GRD response of the solid Earth (equation 11), and the component that only modulates the ocean volume (equation 12). The revised description of the equations (page 7 line 26 - page 9 line 31) along with the Appendix (page 15 line 3 - page 18 line 5) should provide enough information to interpret the individual terms appearing in these equations.

p5.130 Could you explain why H0 appears as an \*absolute\* contribution in the third term compared to considering \*changes\* in  $H_S$  and  $H_F$  in term one and two?

We appreciate this comment very much. In fact, it was a mistake on our end. The correct term should be  $(\Delta H - \Delta H_F)$  rather than  $H_O$ , which is now corrected (equation 12) and its physical justification is given in the Appendices A2 and A3 (page 16 line 8 – page 18 line 5; see also Figure A1). This term, in essence, accounts for the difference in volume between the freshwater that would be produced when ice melts and the oceanic water that would be displaced when it floats (page 9 lines 20-23 and 26-29).

p6.16 Not clear what "holding in the areas of on-land ice margin migration" means. Reformulate.

We realize that the quoted statement is in direct conflict with the assumption that "the net change in grounded ice mass results in the equivalent change in ocean mass..." (page 7 lines 12-13), so we have deleted the sentence and written "...it may often be possible to use grounded ice masks in place of the land domains..." (page 14 lines 5-6). We thank the reviewer for helping us catch this minor inconsistency.

p7.115 I suppose you mean that the nonzero  $\Delta H_F$  is compensated by other terms in Eq. 7. Which ones? This is important to understand. No, but thanks for the opportunity to clarify. What we mean is that externally-forced  $\Delta B$  or  $\Delta S$  may possibly modulate ice dynamics in this regime, even if they do not contribute to  $\Delta H_F$  or  $\Delta H_S$ . We have elaborate in the manuscript (page 9 lines 13-16).

p7.122-23 This sentence may be confusing because Goelzer et al. (2020) consider not only transitions between grounded and floating ice, but all three regimes. Reformulate.

This is correct. Now that we have added a paragraph in the Introduction to summarize the Goelzer paper and contrast it against our goals (page 2 lines 26-35), we really do not need this citation here. We have simply deleted the sentence.

p7.126 Important to note that in regime 3 bedrock changes can contribute to sea-level change, even if you are not considering them as part of the ice sheet change. Also important to realise that ice floating at t and  $t + \Delta t$  may have been grounded at some point in between. I think this makes your solution dependent on the time stepping. See Fig 2a in Goelzer et al. (2020) for an example of such a case.

We acknowledge that the bedrock change in this regime may contribute to spatial pattern of sea level. By referring to **equation (12)**, all we are saying here is that there is no mass contribution to the ocean from this regime of the ice sheet (**page 9 lines 27-29**).

Once  $\Delta H_S$  is defined (equation 10), which is what our focus is at this point, a GRD (gravity, rotation, deformation) model of solid Earth yields self-consistent solutions for bedrock elevation and MSL change. This is a whole different story, which has been briefly summarized in Section 4 (page 13 lines 9-21).

Regarding the last question referring to Fig 2a of Goelzer et al. (2020): Given our explanation on pages 3-4 (see Fig. 1) on the topic, we believe that there is no confusion left in this regard. Nevertheless, here is our brief response to the questions listed above. Yes, ice could have been grounded in this region at some point in time between t and  $t + \Delta t$ . In this instance, the part of the region that is grounded belongs to Regime 2 (not 3; see Fig. 1b) and it should be interpreted accordingly. Yes, our solution is time dependent and this is what it should be. See Fig. 1a. There is no net mass contribution to sea level from the ice column during the period  $[t_1, t_4]$ , while there is clearly non-zero mass contribution from ice during the sub-period  $[t_1, t_2]$ .

p8.17 The change in  $\Delta H_S$  and  $\Delta R_C^I(\omega, t)$  is not only due to ice mass changes, but also due to bedrock changes under the ice. It may be good to mention that here.

We think that the reviewer got confused between "ice mass change" and "ice thickness change".  $\rho_i \Delta H_M$ , the mass component of  $\Delta H_S$ , essentially describes the change in ice mass per unit area (units of kg/m2). And, this field is contributed by ice thickness change, bedrock elevation change and MSL change, which we have thoroughly discussed in the manuscript (Section 3.2 and Appendix). We do not wish to list all of these contributors explicitly again.

p8.18 For clarity, could you mention why potential changes in ocean area from  $O(\omega, t)$  to  $O(\omega, t + \Delta t)$  do not matter for  $\Delta R_C^I(\omega, \Delta t)$  in Eq 8? I suppose the underlying assumption

is that we should be interested in sea level at time  $t + \Delta t$ ?

The change in ocean area from  $O(\omega, t)$  to  $O(\omega, t + \Delta t)$  does matter. The ocean area at time  $t + \Delta t$  is the sum of the ocean area at time t and change in the ocean area over the period  $\Delta t$ . The original Eq. 8 does not appear in the revised manuscript. We rather provide explanation in the text (**page 9 lines 30-31**).

p8.l12  $\mathcal{G}(\omega, t)$  is not explicit in Eq. 7 only implicitly by evoking Eq. 6. This could be mentioned.

Agreed. See page 10 line 4.

p8.114. Maybe remind us what R(w,t) is here, as it is only introduced inline and back at p3.

We have now introduced the relative sea level (RSL) here in Section 4 (page 12 lines 8-9), not earlier in the manuscript.

p8.115 To make this equation more digestible, maybe start with combined symbols. E.g. by combining all the barystatic components like you do in the text.

Other reviewers also have similar concerns. To address these, we have introduced a new section (Section 4) and restructured (and expanded) the original materials as follows.

- 1. We contextualize why we need to consider all contributors of relative sea-level (RSL) change in a mass conserving Earth System framework in order to compute  $\Delta H_S$  and  $\mathcal{T}$  (page 12 lines 2-14);
- 2. We introduce non-steric components of RSL including the one induced by  $\Delta H_S$  itself over the period  $\Delta t$  (page 12 lines 15-27), and interpret them in light of space observations and existing GRD models (page 12 lines 27-31);
- 3. We dissect  $\Delta H_S$  into two parts: the component that modulate both the mass and volume of the oceanic water and drives the GRD response of solid Earth, and the (smaller) component that only modulates the ocean volume (page 12 line 32 page 13 line 6);
- 4. We summarize how the spatial patterns of the "mass" component is computed, which necessitates conservation of mass in the Earth system (page 13 lines 9-18).

We hope that the reader finds it more accessible.

p9.11-24 My experience with the paragraph including Eq. 9 is that a lot of new concepts are suddenly thrown in here without much preparation. Especially the idea to separate the effect of the past from the contemporary would profit from some more introduction. May be a new section with a few introductory sentences could be started p8.16 to prepare the ground for this discussion.

We appreciate the feedback. We have restructured and expanded the original materials (page 12 lines 15-31). See our response to the previous comment.

p9.13 Since  $\Delta R_V(\omega, \Delta t)$  may be the component that takes into account bedrock changes in regime 3 and is complementary to the bedrock changes happening under the ice, I would be interested to see how it is calculated and how the masking works in that case.

One of the strengths of our formalism is that  $\Delta H_S$  as defined by (equation 10) considers the effects of  $\Delta B$  and  $\Delta S$  perturbed by all kind of processes (e.g., glacial/oceanic loads or tectonics). This is clearly stated in the manuscript (page 7 line 5). Our partitioning of the ice domain into 3 distinct regimes is for a convenient interpretation of  $\Delta H_S$ . Regime 3 & other 2 regimes for that matter are defined by considering the effects of  $\Delta S$  and  $\Delta B$ , including  $\Delta B_V$  – the component of bedrock elevation change that corresponds to  $\Delta R_V$ (now denoted by  $R_O$ ). Equation (13) summarizes the processes that contribute to  $\Delta R$ (page 12 lines 15-27). We have interpreted these components in light of space observations and existing GRD models (page 12 lines 27-31). We have also provided summary of how to compute RSL induced by  $\Delta H_S$  (page 13 lines 9-18)). Providing a detailed theoretical/numerical description of each term, including  $\Delta R_O$ , is beyond the scope of this study. In terms of masking, both  $\Delta B$  and  $\Delta R$  are globally-defined fields and there is no need for masking.

p9.128 By your definition, the grounding lines are neither part of the grounded ice nor part of the floating ice domain. What is this category called and how are you accounting for it?

Just like how the ocean O and the land L are complementary fields such that their surface areas make up the total area of the Earth's surface (**page 5 lines 21-22**), the grounded and floating masks are also complementary fields they together make up the ice domains. The grounding lines, like coastlines, form their own level set, with no surface areas (**page 6 line 3**), which can be tracked straightforwardly as discussed on **page 5 lines 25-26**. We agree that we did not provide a mathematical description of the coastlines and grounding lines in the original manuscript, which is now included in terms of a level-set function in the revised version (**page 4 line 16 – page 5 line 14**).

p10.14-5 How does your approach compare to/ differ from that proposed by Goelzer et al. (2020).

In a nutshell, they provide correction terms associated with an evolving bedrock and an externally-forced (spatially uniform) sea level for a common method of estimating ice sheets' contribution to the barystatic sea level change (BSLC). Their focus is the marine portions of stand-alone ice sheet models. In contrast, we provide a new field altogether,  $\Delta H_S$ , that yields BSLC by accounting for spatially variable  $\Delta B(\omega, t)$  and  $\Delta S(\Delta t)$  caused by any geophysical processes. Our method is applied to all plausible settings of ice sheets (e.g., terrestrial- vs. marine-based and grounded vs. floating). In addition, our goal is also to track grounding lines and coastlines in a seamless manner in order to conserve mass in the Earth System and facilitate complex interactions between ice sheets, solid Earth and sea level. While we do not wish to be critical of the Goelzer paper, the new materials that summarizes their work and contrast it against our goals in the Introduction (see our response on pages 2-3 of this document) should provide enough information in this regard.

Figures

Figure 1. It seems confusing to introduce lakes, subglacial lakes and proglacial lakes and then not consider them at all. It would make the figure much clearer to remove them.

Our goal is to make our formalism as generic as possible. Given the complexities of Earth's surface features, it is an extremely challenging task. Figure 1 depicts some of the relevant features (and associated modeling challenges), and in the caption we have clearly stated what is considered and what is not considered in our analysis. We think this is important. We prefer to keep the figure as is.

Figure 2. The geoid (and S) is also defined over land. Please add in both panels.

Agreed. In the revised **Figure 2**, the MSL is drawn over land as well.

Figure 3. Please include a panel with  $\Delta H_F$ . This is important as it is discussed as the conventional method and appears in the difference in panel c. Please also add contour lines to delineate the regions 1-3. Mention that regime 4 does not exist in this region if that is true. Otherwise, delineate regime 4. If different from results in Goelzer et al (2020) it would also be interesting to add a comparison as figure here.

Since the difference between  $\Delta H_S$  and  $\Delta H_F$  is smaller than the either field by a factor of 10 or so (compare **Figure 3b** vs **Figure 3c**), we are hesitant to include a new panel for the latter field (which looks almost same as Figure 3b). We have added contours to show 3 Regimes, as suggested. We find that 4 regimes defined in Fig. 2a to interpret  $H_F$  may be confused with 3 regimes defined in Fig. 2b to interpret  $\Delta H_S$ . We have therefore labelled the formers as Sectors A, B, C, and D. These sectors are not relevant for Figure 3. In principle, the Goelzer method should yield similar, if not the same, result as ours. Because it was not clear to us how the effect of MSL is handled, we decided not to compare against their method. However, we have added two new panels (Figure 3d-e) to show the time series of the Antarctic ice-volume change that is attributable to the sea-level change, computed by ours and the more common HAF method. We find significant difference (on the order of 10-15%) between the two methods (see Figure 3e).

**Referee #2**

**Summary**

This paper presents a framework for describing the geometry of an evolving ice sheet margin in Earth system models, in which the geometry of the bedrock and mean sea level also can be dynamic. The authors define relevant terms and give a mathematical description of the way different quantities are related. In particular, they quantify the portion of ice thickness change that contributes to changes in ocean mass and global mean sea level. As Earth system modelers work to integrate dynamic ice sheet models with solid-Earth and sea-level models, this paper will be a useful reference.

In general, the paper is well written, and the figures are very helpful. Sometimes, however, technical terms related to sea level are used without precision, or are introduced without giving enough background information. Some of the equations contain terms that are not clearly defined or explained physically. In the comments below, I suggest where the exposition could be improved, especially for readers approaching these concepts for the first time.

We thank the reviewer for a positive and constructive review. We have clarified concepts and terminologies that are ambiguous, supplied additional background materials as required, and explained physical meanings of key mathematical terms that were missing. Find our response to individual comments below. The changes made in the revised manuscript are pointed by the **bold** text.

Also, the text refers to "traditional" or "customary" approaches of estimating sea-level contributions from ice sheets based on the change in height above flotation, in contrast to the approach described here. It would be helpful to see some specific examples of customary approaches from published papers, with estimates of the magnitude and sign of the associated errors. This would enable readers to better assess the value of the proposed formalism.

Most of the stand-alone ice-sheet models use  $\Delta H_F$  to estimate ice sheet's contribution to barystatic sea-level change (BSLC), termed the HAF method (page 2 lines 24-26, page 7 lines 15-20). Many of participating models of SeaRISE or ISMIP6 projects are some examples. Unless we compare these BSLC estimates with those derived from better/improved methods (e.g., our methods or Goelzer et al., 2020), which is not obviously in literature, we cannot quantify/report errors associated with these estimates. This is why the results presented in Figure 3, based on a particular model run by Larour et al. (2019), is useful. We have now included two additional panels in the figure (Figures 3d-e) that show the time series of Antarctic ice-volume change that is attributable to sea-level change. We find a significant difference between ours and the HAF method, which is on the order of 10-15% (see Figure 3e). This is now reported in the main text (page 10 lines 24-31) and highlighted in the Conclusion (page 14 line 8).

**Major comments**

p. 1, Title: The title includes the words "mass conserving", but in the text I did not find

an operational definition of what this means in an ESM with evolving ice sheets. Please provide such a definition, and perhaps an example of how mass conservation would be violated.

Although "mass conservation" is still an important element of the paper (see, for example, **page 1 line 10**), we also think that the original title was a bit too vague. We have now suitably modified the title, **A kinematic formalism for tracking ice-ocean mass exchange on the Earth's surface and estimating sea-level change**, which is better aligned with the key goals of the paper (**page 1 lines 5-8**). As a compensation, we have now introduced **Section 4** wherein we place the presented formalism in a broader context of sea-level change and mass conservation in the Earth System. We have now defined what mass conservation means in the context of our formalism (**page 2 lines 32-35**) and also provided example cases where mass conservation is violated (**page 13 lines 18-21**).

Also, the title implies that there will be a detailed analysis of ice-sheet interactions with solid Earth and sea level, but the actual scope seems narrower: to accurately compute the contribution of dynamic ice sheets to barystatic sea level rise (i.e., the sea-level component associated with a redistribution of mass between land-based ice and the ocean) in ESMs. I suggest a revised title that better reflects the scope.

We fully understand the reviewer's concerns, and we agree that the original title was a bit vague. We have revised it appropriately to better align with the the key goals of the paper (see our response to the previous comment). We appreciate the comment.

p. 1, Abstract: The abstract is very general and does not provide a clear sense of the scope of the paper. If a central goal is to describe how to accurately compute the ice sheet contribution to barystatic sea level rise, then this goal should be clearly stated.

We agree that our original abstract was also a bit too vague. We have rewritten the second half of it, aiming at more focused and impactful abstract (**page 1 lines 5-10**).

p. 1, l. 1: Although I am fully in favor of including dynamic ice sheets in ESMs, I would not go so far as to suggest that "any Earth System model" already includes them. For some ESMs, ice sheets are still on the back burner.

Agreed. We now write "...components of the Earth System..." (page 1 line 1).

p. 2, l. 2: "Defining geometry". I am not sure what this means – something like defining geometric concepts that are relevant to models with dynamic ice sheets?

Referee #1 also had a concern about the quoted phrase. We agree that this sentence is ambiguous. We have decided to delete it altogether. The removal of this sentence does not break the flow of the write-up.

p. 2, l. 4: "future debate and reconciliation." Ideally, the concepts are set forth clearly in a way that leads to greater mutual understanding and less debate.

This phrase and "defining geometry" (see previous comment) both appear in the same sentence, which we have removed from the revised manuscript.

p. 2, l. 5: "basic configuration setup". I am not sure what this means. It seems an odd way to describe what seem to be theoretical or analytical frameworks.

Agreed. Referee #1 also found this phrase unclear. We have rephrased the sentences in a much clearer way (page 2, lines 5-9). The quoted words do not appear in the manuscript. Also see our response to your next comment.

p. 2, l. 8: I am not clear why these previous analyses are referred to as "traditional configurations". The word "traditional" suggests a contrast with something novel and untraditional to be introduced here. However, I don't see this work as heading off in a different direction from the cited papers, but rather as clarifying concepts that are particularly relevant for ESMs. As above, "configuration" doesn't seem to be the right word.

We no longer label the previous analyses as "traditional", and have avoided its usage in the manuscript along with the word "configuration". We have deleted this sentence.

We labeled the previous studies that considered simplified geometries of ice/bedrock/ocean system (e.g., Hutter et al., 1983; Lambeck et al., 2003; Mitrovica and Milne, 2003) as "traditional" because these studies do not capture kilometer-scale geometric features, e.g. ice rises and rumples and rugged fjord geometries (page 2 lines 11-15), that are critical to understand grounding line dynamics. One of our key goals is to present a method that can track the grounding lines and coastlines of arbitrary geometries within a system of ice sheet, solid Earth and sea level models (page 2 lines 16-19).

p. 2, l. 13: Similarly, what is meant by "traditional theory for ice-bedrock-ocean interface changes"?

We believe that our response to the previous comment fully addresses this question. See the revised paragraph (**page 2 lines 5-19**) in the manuscript.

p. 2, l. 25: "sea surface elevation". This is an ambiguous term; it can refer either to the mean (on some appropriate time scale) or to a quantity that varies on short time and spatial scales. There is some explanation below, but it is better to be as clear as possible when introducing the quantity  $S(\omega, t)$ . I think that what is meant here is what Gregory et al. (2019) call "mean sea level", a term they recommend in place of the deprecated term "mean sea surface". If S is actually meant to represent the geoid, which is not quite the same as mean sea level, then this should be stated clearly.

In general, Gregory et al. (2019) is a comprehensive, carefully written reference. I suggest that the authors adopt similar terminology, paraphrasing and referring to that paper as appropriate.

We agree that Gregory et al. (2019) is the key reference here. In the revised manuscript, we have complied with their definitions and made any difference explicit (**page 3 lines 22-29**). The key difference is that the mean sea level (MSL) in the context of our formalism does not account for the steric and dynamic components. Specifically, the global-mean value of MSL is given by the barystatic sea level, and its spatial pattern is dictated by

GRD (gravitational, rotational, deformational) response of the solid Earth to land-ocean mass exchange. This definition of MSL is familiar in GIA modeling (**page 3 lines 30-33**).

p. 2, l. 26: What is the International Terrestrial Reference Frame, and how is it defined? Is it similar to what Gregory et al. (2019) call the reference ellipsoid?

The International Terrestrial Reference Frame (ITRF) is the standard reference frame defined by the International Earth Rotation and Reference Systems Service (www.iers.org). It is being updated every few years. Yes, the ITRF solutions (e.g., GRS80) are reference ellipsoids as also noted by Gregory et al. (2019). We have decided not to specifically mention ITRF in the manuscript (whose description would divert the flow of the paper), because the theoretical formalism presented here is valid as long as both the bedrock and MSL are measured with respect to the same reference frame (page 3 lines 20-21).

p. 2, l. 32: It is stated first that S is highly variable in space and time, and then it is stated that S is quasi-static and does not, in fact, include short-term dynamic processes. Please use S only to refer to quasi-static mean sea level, and use a different term when discussing short-term dynamics.

These sentences do appear in the revised manuscript. We have now provided a more precise definition of S (**page 3 lines 22-33**), which generally complies with the definition given by Gregory et al. (2019).

p. 3, l. 5: "Sea level" is another ambiguous term, as discussed by Gregory et al. I suggest "mean sea level."

Agreed. We now use mean sea level (MSL) (page 3 line 20). See also the revised paragraph (page 3 lines 22-33) in the manuscript.

p. 3, l. 5: "represents an equipotential surface whose spatial pattern mimics the geoid." This is confusing. First, how is the geoid defined? Gregory et al. define it as the geopotential surface chosen so that the volume between the geoid and the sea floor is equal to the time-mean volume of sea water (including the liquid-water equivalent of floating ice) in the ocean. Second, what is meant by "mimics" the geoid? Does "mimics" mean "is equivalent to", or "is similar to"? If the latter, in what way does  $S(\omega, t)$  differ? If S is mean sea level, then it is not an equipotential surface; for instance, mean sea level has a higher geopotential on one side of the Gulf Stream or ACC than the other. (Though it could be convenient to define S as an equipotential surface in areas not covered by ocean.)

The mean sea level (MSL), as defined in the revised manuscript, is familiar in GIA modeling, which requires to solve the so-called "sea level equation". This particular definition, which complies with general definition of MSL (Gregory et al., 2019), represents the equipotential surface and it differs from the geoid only by a spatial invariant for the sake of mass conservation in sea-level equation (Tamisiea, 2011). See **page 3 lines 30-33**.

p. 3, l. 11: Since ocean and ice have variable density, it would be clearer to refer to  $\rho_o$  and  $\rho_i$  as reference densities.

Agreed. See page 4 line 7.

p. 3, l. 12: "sea surface relative to the seafloor". I suggest "local mean sea level relative to the seafloor".

Agreed. This sentence has been moved to Section 4 (page 12 line 8).

p. 4, Eq. (3): Why are the ocean and land functions undefined at coastlines and grounding lines? Is it problematic not to include them in one domain or the other?

The coastlines and grounding lines are neither part of the land nor of the ocean. They are the interfaces between the two. However, we understand the reviewer's concern given the lack of a precise description of these interfaces. We have included a mathematical description of a generic land-ocean boundary, including grounding lines (page 4 line 16 – page 5 line 14). We have also given a specific example of how one can carry information about the domeain interfaces in numerical models (page 5 lines 22-26).

p. 4, l. 7: Please say precisely what is meant by "connected to the open ocean". I would guess that the connected regions include marginal seas (e.g., the Mediterranean) but not inland lakes (e.g., the Great Lakes). Also, one needs an operational definition of the open ocean before defining a connection to the open ocean.

With new materials that present a mathematical description of level set function  $\mathcal{F}(\omega, t)$ and its zero level set  $\mathcal{T}(\mathcal{F})$  (page 4 line 16 – page 5 line 14), we do not need to invoke the (relatively subjective) concepts of "connectivity" and "open ocean". The ocean function as defined by equation (5) is free of any ambiguity.

p. 4, l. 17: "is connected to" is more precise than "is in direct contact with"?

As noted in our response to the previous comment, the concept of "connectivity" has been discarded altogether by providing a mathematical description of the level-set function. Related phrases do not appear in the revised manuscript.

p. 4, l. 23: Are there published examples of frameworks (i.e., "traditional theory") that cannot handle pinning points? It seems natural to define the ice domain as in Eq. (4), and I'm not aware of frameworks with a different or less natural definition.

No doubt that Eq. 4 (now equation 6) is a simple and perhaps the most generic definition of ice domains (page 6 line 5). And, there is nothing new about it here, compared to the previous studies. To avoid a confusion, we have deleted the referred sentence. However, what we meant to imply by referring to the previous (aka traditional) studies is that they consider simplified geometries of land, ocean and ice sheets, perhaps due to the lack of constraining data or computational resources (page 2 lines 5-9). Consequently, these studies do not capture kilometer-scale features such as rugged fjord geometries and pinning points (page 2 lines 11-15) that are critical for accurate modeling of the grounding-line migration and hence ice-sheet dynamics. Also, see our response to your comment #p.2,1.8.

p. 4, l. 24: Which "employed assumption" is being referenced here? Maybe the assumption that only ice that is part of the ocean domain is included within the floating ice mask?

Correct. This has been clarified (page 6 lines 7-8).

p. 4, l. 27: "the first generation of Earth system models". I'm not sure we are still in the first generation, since ESMs have been around for about a decade. Maybe "current Earth system models".

Agreed. See page 6 line 10.

p. 5, Eq. (6): Why is the grounded mask needed in this expression, if it is true that  $H = H_0$  for floating ice shelves?

We define the flotation height for ice  $H_0$  (equation 7) based on the height of the oceanic water rather than the ice thickness, and we interpret  $H_0$  as the fraction of ice thickness that can potentially contribute to sea level by channing the mass of the oceanic water (page 6 lines 24). Ice shelves must have smaller thickness than  $H_0$ , implying that the ice shelves can potentially contribute to the barystatic sea-level change, which obviously is not correct. Therefore we must invoke the grounded ice mask in equation (8).

p. 5, l. 8: What is the referent in "it can be negative"?

It is  $H_F$ . Now clarified on page 6 line 29.

p. 5, l. 21: What is meant by "directly affects"? Is this just the (fairly trivial) statement that some, but not necessarily all, of the net change in ice mass results in a change in ocean mass?

Yes to your second question. We have rephrased the statement (page 7 line 12), and the quoted phrase does not appear anymore.

p. 5, ll. 21-22: "Quantifying the fraction of ice mass change that contributes to sea level" I thought that this was the main point of the section. In what sense is it analytically unapproachable and beyond the scope of the study?

We meant to imply that not all of ice mass loss makes it to the ocean. A fraction of it, for example, may get impounded in proglacial lakes. However, the quoted statement is in direct conflict with the assumption that "the net change in grounded ice mass results in the equivalent change in ocean mass..." (**page 7 line 12**), so we have removed the referred sentence. The reviewer is correct that  $\Delta H_S$  (well, in fact, its one component  $\Delta H_M$ ) provides the accurate estimate of ice mass loss that contributes to the barystatic sea level (see **page 7 equation 10 and the text that follows**).

p. 5, l. 23: "Despite the assumption"? Which assumption? If the reference is to the assumption that "the net change in ice sheet mass directly affects the ocean mass", I'm not sure there is a contradiction, but I'm not clear on the precise meaning of the assumption.

Yes, it was in reference to the quoted assumption. We agree that there is no contradiction either. We have deleted the quoted phrase and revised the sentence (**page 7 line 13**).

p. 5, l. 27: "...yields some error." After reading this statement, I was expecting to see quantitative error estimates later in the text. There is an illustration in Fig. 3, but is it possible to state the typical order of magnitude of the error? For instance, is it closer to

1% or 10%?

This is an excellent point. We have now added two new panels in the figure (Figures **3d-e**) to show the time series of ice-volume change that is attributable to the sea-level change. As shown in Figure 3e, we find that the new method predicts much larger sea-level contribution (on the order of 10-15%) than the usual method (the height above flotation, HAF, method) throughout the model simulation. This has been discussed in Section 3.3 (page 10 lines 24-31) and highlighted in the Conclusion (page 14 line 8).

p. 5, l. 28: Please say precisely what is meant by "contributes to sea level". For example, if an ice sheet loses mass, the geoid will change because of gravitational and rotational effects, but these changes aren't part of  $\Delta H_S$ . What is meant, I think, is the part of the ice loss that adds to the mass of the ocean, i.e. the barystatic sea-level component. If so, then barystatic SLR and related terms should be defined here or earlier.

We agree with the reviewer's assessment. We have now explicitly stated that  $\Delta H_S$  contributes to the sea-level change by modulating both the mass and volume of the oceanic water (**page 7 lines 27-28**). In fact, we have restructured the original equation to isolate "mass" and "volume" component of  $\Delta H_S$  so that it can be easily interpreted in light of GMSL estimates or GRD computations (**page 7 line 30 – page 8 line 7**).

p. 5, l. 30: Since this is a central equation in the paper, I would like to see a clearer description of the physical meaning of each term, and when appropriate a derivation. I convinced myself that the first two terms on the RHS are correct, but I was not able to derive the third term or understand the physical motivation. In the text, the closest thing to an explanation is on p. 7, l. 20: "The last term in the equation accounts for the fact that fresh water density evolves during the accretion and ablation of ice, whereas the average ocean water density in the vicinity of the grounding line acts to determine the ablation height." This is confusing, in part because freshwater density  $\rho_w$  is a physical constant that does not evolve. Please provide a clearer explanation and, if possible, a supporting figure.

The last-term of equation (7) in the original manuscript was incorrect, which is now corrected and is given as a separate equation (Equation 12) because this term only modulates the ocean volume (not mass) and does not participate in GRD calculations. We have now added an Appendix (page 15 line 3 – page 18 line 5) wherein we have derived individual terms appearing in Equations (11-12) and provided their physical interpretation. We have separated three regimes of the ice domain (page 9 lines 4-31) to summarize such interpretation in the main text. The quoted sentence does not appear in the manuscript. See page 9 lines 20-24 and 27-29 for the revised statements.

p. 6, Figure 2: The figure and caption are helpful, especially panels a and b with the four different regimes.

**Thank you.**

p. 7, ll. 3ff: The text refers to three distinct "regimes", whereas Fig. 2 refers to four regimes that are defined differently from the regimes in the text. Please use the term "regimes" consistently. In the text, paragraph 1 corresponds to the first term on the RHS

of Eq. (7), and paragraph 2 to the second and third terms. But as stated above, the explanation of the third term is not clear. Perhaps revise so that paragraph 2 addresses the second term and paragraph 3 the third term. Then the current paragraph 3 would become a short paragraph 4.

Indeed, we also realize that the 4 regimes originally defined to interpret  $H_F$  (Figure 2a) may be confused with the 3 regimes defined to interpret  $\Delta H_S$  (Figure 2b). To avoid this confusion, we have labelled the formers as Sectors A, B, C and D, and the latter as Regimes 1, 2, and 3 (see Figure 2). The description of three regimes has been restructured, as suggested when possible (page 9 lines 4-31). The explanation of third term appearing in equation 7 of the original manuscript (now Equation 12) has been considerably improved (page 9 lines 20-24 and 27-29). Also see our response to your comment #p.5, l.30.

p. 7, l. 19: Could you give an example of when the magnitude of the change in  $H_F$  would be equal to the magnitude of the change in H, and when it would be less?

We have now provided a detailed interpretation of the  $\Delta H$ - $\Delta H_F$ - $\Delta H_S$  relationship in the **Appendix**, in light of evolving ice thickness, bedrock elevation and mean sea level. As described in **Appendix A2 (page 16 line 8 – page 17 line 29)**, the inequality (not equality) must hold in the mentioned relation for a grounded ice colume to float, or the reverse, in the absence of the externally-forced bedrock and mean sea level change (see Figure A1b). This has now been corrected (page 9 line 20). We thank the reviewer for this comment.

p. 7, ll. 29ff: I am confused about the difference between events in regimes 1 and 2. My understanding is that regime 1 consists of regions that are grounded at both the start and end of the simulation, whereas regime 2 consists of regions that transition from grounded to floating during the simulation. If so, this should be stated clearly. For regime 1, it is stated that  $\Delta H_S$  is different from  $\Delta H_F$  because of "evolving bedrock and sea level". Are bedrock and sea level not evolving in regime 2? Or is the point rather that in region 1, the ice remains grounded throughout the simulation, and therefore the entire DeltaH contributes to  $\Delta H_S$ , whereas  $\Delta H$  differs from  $\Delta H_F$  because of bedrock changes? For regime 2, it is stated that the discrepancy is due to the ?missing fraction of newly grounded or newly floating ice.? I am not sure what this means. I understand why  $\Delta H_S$ differs from  $\Delta H$  in this region, but not why  $\Delta H_F$  differs from  $\Delta H_S$ .

The reviewer is correct about how we separate three regimes in Figures 3a-c, which has now been mentioned explicitly in the manuscript (page 10 lines 18-19; also see the figure caption). It is not that we consider the effects of evolving bedrock and mean sea level in one regime, and not in others. We first define  $\Delta H_S$  by accounting for  $\Delta H$ ,  $\Delta B$  and  $\Delta S$  (Equations 10-12). Only then we separate 3 regimes for the ease of interpretation. We agree that our interpretation of the results, particularly Figure 3c, is too brief and perhaps confusing, which has been significantly improved now (page 10 lines 20-23). The reviewer's second interpretation about the results in Regime 1 is accurate (page 10 lines 20-21). Mathematically, the difference between  $\Delta H_S$  and  $\Delta H_F$  in Regimes 2 and 3 is simply  $\Delta H_V$  (equation 12), which accounts for the volumetric contribution of ice-thickness change in excess of the change in HAF (page10, lines 22-23). p. 7, l. 30: Please say more precisely what is meant by the "customary approach of using  $\Delta H_F$ ". Can you cite specific examples in the literature in which the ice-sheet contribution to SLR was derived from  $\Delta H_F$ , yielding a significant error? In the literature (beyond this specific example from Larour et al. (2019)), do the errors have a systematic sign? Are these errors prevalent in ice sheet models that include isostatic adjustment (i.e., where  $\Delta H_S$  could have been computed accurately, but  $\Delta H_F$  was reported instead)? Or is the problem that most ice sheet models ignore isostatic adjustment, so that they are missing a key term needed to compute  $\Delta H_S$ ?

By "customary approach" we meant to imply the methods that use the change in HAF to estimate the sea-level contribution (**page 7 line 17**). The phrase does not appear in the manuscript anymore (see Section 3.3 for the revised text).

Regarding the second question: Most of the stand-alone ice sheet models (e.g., those contributing to SeaRISE and ISMIP6 projects) derive ice sheets' contribution to sea-level change from  $\Delta H_F$  (page 7 lines 16-17). Unless we compare these estimates with those derived from better/improved methods (e.g., our methods or Goelzer et al., 2020), we cannot quantify/cite errors associated with these estimates. This is why the results presented in Figure 3 is useful. We have now included two additional panels in the figure (Figures 3d-e) that show the time series of Antarctic ice-volume change that is attributable to sea-level change. We find a significant difference between ours and the HAF method, which is on the order of 10-15% (see Figure 3e). This is now reported in the main text (page 10 lines 24-31) and highlighted in the Conclusion (page 14 line 8).

Regarding the third question: As noted above there are no error estimates available in the literature, so we cannot comment on whether they have a systematic sign. But, for the particular case consider in our study, we find that the HAF method systematically underpredicts the sea-level contribution (**page 10 line 25, also see the figure caption**).

Regarding the fourth question: It may be possible. Goelzer et al. (2020), for example, write: "In our own ice-sheet modelling experience and from exchange with colleagues in different groups, it is not always clear how the sea-level contribution should exactly be calculated and what corrections need to be applied."

Regarding the fifth question: The reviewer's speculation is absolutely correct. Most of the participating models of SeaRISE and ISMIP6 experiments are stand-alone ice models.

Also, can you state the magnitude of the systematic error? That is, what is the magnitude of the integrated error in Fig. 3 panel c, relative to the integrated value of  $\Delta H_F$ ?

Ok. This error, at least for the example case considered in our study, is on the order of 10-15%, now reported in the main text (**page 10 line 26, see also Figure 3e)** as well as in the Conclusion (**page 14 line 8**).

p. 8, Fig. 3: In addition to the 2D fields, it would be useful to show a graph consisting of time series of the area-integrated values of  $H_S$  and  $H_F$ . This graph could show not only the total values, but also the values computed separately for regions 1, 2, and 3. Also, please cite Larour et al. (2019) in the caption.

We appreciate the comment. We have now added two new panels as suggested by the reviewer (Figures 3d-e). Instead of partitioning the results by the regime, we believe that it is more instructive to partition them by the "mass" and "volume" components as they have direct implications for interpreting results in terms of GMSL or GRD computations. It is interesting that the two methods differ from each other by about 5% consistently throughout the model simulation in terms of their mass contribution to the ocean, the component that drives the GRD response (page 10 lines 27-29). We have now cited Larour et al. (2019) in the caption, as suggested.

p. 8, l. 6: Here, barystatic sea level change is finally defined. I suggest introducing and defining this concept earlier in the paper. Also "ocean mass-related" is a bit vague; I suggest phrasing similar to that of N19 in Gregory et al. (2019): e.g. "the part of global-mean sea-level rise which is due to the addition to the ocean of water mass that formerly resided within the land area as land water storage or land ice." Then  $\Delta R^{I}$ , introduced below, would be the land-ice contribution, and  $\Delta R^{L}$  would be the contribution from other land terms.

We have defined the barystatic sea level much earlier in the manuscript (page 3 lines 27-28). The quoted phrase does not appear in the manuscript (page 9 lines 31-32). See Section 4 for improved definitions of the barystatic components that contribute to RSL change (page 12 lines 15-31).

p. 8, Eq. (8): The term on the LHS includes a subscript, a superscript, and an overbar, without immediately saying what these things mean. I suggest a more gradual and systematic introduction to the notation. Also, could you explain why the denominator contains  $\rho_w$  instead of  $\rho_o$ ? At first, I assumed that the denominator represents the mass of the ocean, but I think the reason for  $\rho_w$  is that we are converting a mass of fresh ice into an equivalent ocean volume, ignoring halosteric effects. Again, a more detailed physical explanation would be helpful.

As we have restructured the materials, largely under Section 4 (page 12 line 1 – page 13 line 21), the cited equation does not appear in the manuscript. We have rather explicitly stated in the text that the spatial integration of  $-\rho_i/\rho_w \Delta H_S$  gives the total freshwater volume being added to the ocean (page 9 line 33). Also see our response to your comment that follows.

p. 8, Eq. (9): This equation introduces several more terms without preamble, and the reader has to study the following paragraph carefully to translate each term. Please rewrite in a way that is gentler for the reader.

We appreciate the comment. To address this and similar comments, we have introduced a new section (Section 4) and restructured (and expanded) the original materials as follows.

- 1. We contextualize why we need to consider all contributors of relative sea-level (RSL) change in a mass conserving Earth System framework in order to compute  $\Delta H_S$  and  $\mathcal{T}$  (page 12 lines 2-14);
- 2. We introduce non-steric components of RSL including the one induced by  $\Delta H_S$  itself

over the period  $\Delta t$  (page 12 lines 15-27), and interpret them in light of space observations and existing GRD models (page 12 lines 27-31);

- 3. We dissect  $\Delta H_S$  into two parts: the component that modulate both the mass and volume of the oceanic water and drives the GRD response of solid Earth, and the (smaller) component that only modulates the ocean volume (page 12 line 32 page 13 line 6);
- 4. We summarize how the spatial patterns of the "mass" component is computed, which necessitates conservation of mass in the Earth system (page 13 lines 9-18).

We hope that the reader finds it more accessible.

p. 9, ll. 7-9. I am not clear on the meaning of the third and fourth ("past") terms, and how these terms change the ocean mass. I understand that past ice-sheet changes affect sea level through ongoing glacial isostatic adjustment, but isn't GIA included in term 6, the vertical-land-motion term?

The third and fourth terms are indeed related to GIA processes (page 12 lines 30-31), that capture the ongoing viscous response of the solid Earth to the ice-ocean mass exchange since the Last Glacial Maximum. They modulate not only the present-day bedrock elevation but also the geoid field, hence their imprints are imbedded both in  $\Delta B$ and  $\Delta S$ . The GIA processes are part of what we call the "barystatic components" (page 12 line 19). The last term  $\Delta R_V$  (now  $\Delta R_O$ ) is reserved for other non-barystatic processes that at least modulate the ocean bathymetry or coastal geometry (e.g., earthquakes, landslides, etc.), now clarified on page 12 lines 25-27.

A more general comment: It could be easier for the reader if the text were organized from general to specific, instead of specific to general. That is, first define the various kinds of sea level rise, introduce notation, and state the various source terms. Then state that this paper is focused on  $R_C^I$ , as computed in equation (8) based on  $\Delta H_S$ . Finally, show how to compute  $\Delta H_S$ . This would be a fairly major rewrite, and I don't want to be too prescriptive, but it is challenging for readers to introduce basic concepts just a page or two before the conclusions.

We really appreciate this comment. We have restructured and expanded the content of the original materials, now under **Section 4 (page 12 line1 – page 13 line 21)**. See our response to your commennt #p.8, Eq.9 for an overview of Section 4.

p. 9, l. 16: When  $\Delta M$  is described as a mass-conserving field, is this equivalent to saying that its global integral is zero? Also, is it strictly true that  $\rho_i \Delta H_S$  is equal to the change in ice mass at each location? Here, I'm wondering about the third term in Eq. (7); is that term associated with a change in the local mass per unit area?

Yes to the first question. See **page 13 line 14**. The change in  $\rho_i \Delta H_M$  (Equation 14) is equal to the change in ice mass per unit area that contributes to the change in ocean mass. The correct version of the third term of Eq. (7) in the original manuscript now appears in (Equation 12), whose units are ice-equivalent height, not mass per unit area. See **page 9** lines 20-23 and lines 27-29 for its physical interpretation. p. 9, Conclusions: As mentioned above, it would be helpful to quantify the benefits of the new methods, e.g. by estimating the errors associated with the older methods.

Agreed. The level of improvements are quite large (about 10-15%, see **Figure 3e**), now highlighted in the Conclusion (**page 14 line 8**). We appreciate the comment.

**Minor corrections**

p. 1, l. 6: "and include the ice shelves and adjacent ocean mass." The phrasing is awkward. Please use a parallel grammatical construction.

We have completely rewritten the second half of the Abstract, highlighting the two main goals of the paper (page 1 lines 5-8) and their utility and implications (page 1 lines 8-10). The referred sentence does not appear in the revised manuscript.

p.1, l. 7: "is"  $\rightarrow$  "can be"? p. 1, l. 15: "grounding line"  $\rightarrow$  "grounding lines" p. 1, l. 17: Delete "involved" p. 2, l. 2: "first order"  $\rightarrow$  "first-order" p. 2, l. 28: Delete "for" p. 3, l. 5: "refer"  $\rightarrow$  "refer to" p. 4, l. 30: "ice sheet driven"  $\rightarrow$  "ice-sheet-driven" p. 4, l. 32: "far field"  $\rightarrow$  "far-field" p. 5, l. 2: Add "the" after "estimate" p. 5, l. 20: "farfield"  $\rightarrow$  "far-field" p. 5, l. 20: "farfield"  $\rightarrow$  "far-field" p. 5, l. 26: Add a comma after "below" p. 6, l. 7: "predict"  $\rightarrow$  "predicts" p. 9, l. 15: Insert "the" before "ice sheet" p. 10, l. 6: "analyses"  $\rightarrow$  "analysis" We have implemented all of these changes.

**Referee #3**

The paper by Adhikari et al. presents a formalism to calculate the contribution of dynamic ice sheets to mean sea level by considering a changing sea-level, bedrock and land-ice-ocean mask. Compared to standard methods the contribution to mean sea level is computed from ice thickness and not from the height above flotation. The formalism is valid on all timescales, but the full benefit is on longer timescales. As there are currently huge efforts to include dynamic ice sheets in Earth system models the presented paper is a good reference.

Generally, I found the paper well written and structured with illustrative and clear figures. The paper is worth publishing in The Cryosphere but as the focus is rather technical it would fit much better to GMD. The current form of the manuscript lacks a bit in the presentation and clarity. Therefore, I have a few suggestions that should be addressed in a revised version.

We appreciate the positive and constructive review by the referee. Find our response to individual comments below. The changes made in the revised manuscript are pointed by the **bold text**.

1) I found the title a bit misleading to the content of the paper. I am missing the definition of "mass conservation" in the text. I also would like to see (e.g. with an example), if the formalism is mass-conserving (or better mass-conserving than traditional or other methods). Also, the only example in the manuscript was about calculating SLE relevant thickness changes rather than mass-conservation.

Although "mass conservation" is still an important element of the paper (see, for example, page 1 line 10), we also think that the original title was a bit too vague. We have now suitably modified the title, A kinematic formalism for tracking ice-ocean mass exchange on the Earth's surface and estimating sea-level change, so that it reflects the key goals of the paper (page 1 lines 5-8). As a compensation, we have now introduced Section 4 wherein we place the presented formalism in a broader context of sea-level change and mass conservation in the Earth System. We have now defined what mass conservation means in the context of our formalism (page 2 lines 32-35) and also provided example cases where mass conservation is violated (page 13 lines 18-21). In terms of results, there is nothing really to show to justify mass conservation except to state that Equation (14) takes  $\Delta H_M$  from equation (11) and  $\Delta R_C^I$  is computed using a global GRD (gravitational, rotational, deformational) solid Earth model, such that the global integral of  $\Delta M$  equals zero (page 13 lines 9-18).

2) The Introduction should refer to Goelzer et al. (2020). You do very quick comparisons to Goelzer et al. (2020) on page7, line6 and 23, but I think the Introduction should clearly say what you are doing differently and why. This could be a motivation to release your new formalism. An appropriate discussion to Goelzer et al. (2020) is also missing. Additionally, from the Introduction it was not really clear to me what is actually wrong with the traditional methods, the order of error on SLE they could introduce and what you are now aiming to improve.

In the Introduction, we have briefly discussed the common method used in estimating sea-level contribution from ice sheets and its key limitations and summarized the effort by Goelzer et al. (2020) to provide appropriate corrections (page 2 lines 24-27). We then contrast the goal of our paper (page 2 lines 28-35). A detailed discussion about the common method, which is based on the concept of ice-height above flotation (HAF), is given on page 7 lines 16-25. Regarding ours versus the Goelzer method, kindly see our response to General Comments by Referee #1 on pages 2-3 of this document. As for the knowledge about errors associated with the existing methods, kindly see our response to Referee #2's Summary on page 15 of this document. For the example case that we have considered in this study (see Figure 3e), however, we find that our method predicts much larger sea-level contribution compared to the common HAF-based approach (by about 10-15%). This has now been discussed in the main text (page 10 lines 24-31), and highlighted in the Conclusion as well (page 14 line 8).

3) Not sure if this could be really addressed, but it would be interesting to estimate the errors (i.e. traditional versus your formalism) of current projections of SLE from the two big ice sheets e.g. within the ISMIP6 framework (Antarctica: Seroussi et al., 2020; Greenland: Goelzer et al., 2020b). Or from current remote sensing products like IMBIE (Shepherd et al., 2019). My point here is, that I would get a better feeling for the error on e.g. different timescales, regional settings and how it differs for Greenland and Antarctica. The example based on the Larour et al. (2019) simulation is very helpful (see also my comment to P7,129ff) but very specific – and, as I understood – not in line with current projections efforts. I do not strictly insist that you show an error for ISMIP6 or IMBIE, but as also commented below, I would like to have a better error estimate and its impact on current research (compare Fig. 3 in Goelzer et al. (2020)).

We appreciate the comment. This sounds like a great idea, but due to the lack of enough information in these datasets and model results, i.e.  $\Delta B$  and  $\Delta S$ , we really cannot evaluate  $\Delta H_S$  (equation 10). Computing  $\Delta B$  and  $\Delta S$  is beyond the scope of this theoretical paper. That said, based on Larour et al. (2019), we have shown the results to highlight the improvement that our method makes over the more common HAF method (Figure 3). We have now included two additional panels (Figures 3d-e) wherein we compare the two methods in terms of estimated Antarctic ice-volume change that is attributable to the global-mean sea-level (GMSL) change. The figure (panel e) also shows the level of improvements possible by employing the new method over the HAF method, which is quite large: on the order of 10-15%.

**Minor comments:**

P1,18: I have not found in the text, which computational strategies you have simplified. What do you mean with computational strategies?

For improved clarity and focus, we have completely rewritten the latter half of the Abstract (**page 1 lines 5-10**). The cited sentence does not appear anymore.

P2,111: Why are "floating ice shelves, ice rises and rumples, and retrograde bedrock slopes" complex features?

We have slightly modified the phrasing "...kilometer-scale geometric features, such as ice rises and rumples, rugged fjord geometries..." (page 2 line 12). We do not call these features complex in the revised text. What we really meant to imply in this sentence, however, is that these kilometer-scale geometric features are not generally captured in the previous studies of evolving ice-bed-ocean interfaces (page 2 lines 13-15), although they are absolutely essential to understand the dynamics of marine-based ice sheet.

P2,113: What is "traditional theory for ice-bedrock-ocean interface changes"?

This question is fully addressed in our response to comment #p.2,l.8 on page 17 of this document.

P2,118: "... that can be straightforwardly employed in any Earth System model"?. I think it would be worth to mention (somewhere), if this new formalism could be adopted to other disciplines (e.q. remote sensing, standalone ice sheet modelling). In the current form, it sounds the formalism in only valid/applicable in ESMs.

By design, the proposed formulation is very generic and we do not see why it cannot be applied for both (stand-alone) modeling and observational data (**page 2 lines 18-19**). For example, definitions of land, ocean, and ice domains, as well as their interfaces are valid in both realms. However, due to lack of modeled/observed data for some fields (e.g.,  $\Delta B$  and  $\Delta S$ ) some aspects of the formalism (e.g., equation 10) are best suited for ESMs (**page 1 lines 9-10, page 2 lines 29-35**).

P4,123: I cannot see from Eq. 4 that your new setup diverges from traditional approaches. Eq. 4 is a very common equation to define an ice-mask. On page5,line 25 you give another example of how the traditional setup differs from your setup. Maybe outlining the differences could be gathered together.

Agreed that Eq. 4 is a simple and perhaps the most generic mathematical description of ice domains (page 6 line 5), and there is nothing special about it here. To avoid any confusion, we have simply deleted the referred sentence. Regarding "traditional" versus our setup: kindly see our response to comments #p.2, l.8 on page 17 and #p.4, l.23 on page 19 of this document.

P5,18: I don't understand this sentence. What is negative?

 $H_F$  is negative. It is clarified now (page 6 line 29).

P5,l9: "...hence contribute to sea level inversely." Maybe say sea-level drop/fall to avoid confusion.

Agreed. See page 6 line 30

P5,110: I am not a native speaker, but are both "evolving" really needed?

We have rephrased the sentence (page 7 line 1).

P5,125: Can you add a reference to a Figure after "show"?

We believe that line 26 is being refereed here, not 25. Considering this and Referee #1's comment (#p5.26 on page 10 of this document), we have decide to remove the phrase containing the quoted word. See **page 7 lines 24-25** for the revised sentence.

P7,l4: "...and the elevations"?

Should be "at the elevations" as is in the manuscript. But, we understand the reviewer's confusion. We have suitably revised the sentence (page 9 lines 5-6).

P7,129ff: I would first describe the Larour et al. (2019) setup and then present the results. Can you give an integrated value (e.g. SLE) for both approaches to get a better feeling for the error? The simulations were run over 500 years. Why do you choose to present the results after 350 years?

We appreciate the comment. We have expanded and restructured the paragraph as advised, under the new Subsection 3.3 (page 10 lines 5-31). There is no particular reason as to why we chose 350 years, but we keep it as is. However, we now include time series of the total Antarctic ice-volume change over 500 years that is attributable to the sea-level change (Figures 3d-e). The panel e shows that the difference between ours and the more common HAF method is significant: on the order of 10-15%.

P9,16ff: The following paragraphs and Equations appeared very suddenly and without introducing their purpose. According to the title, I would expect Eq. 10 (the mass conserving field M) is the main point in your paper. But this is not illustrated and somehow contradicts with your statement in the conclusion (p9,131-32); here you say  $\Delta H_s$ is the main point. This is perhaps personal matter, but I found it a bit brutal to stop the results of the paper with these equations. An illustrative example on "implication of this new geometrical setup for sea level and solid Earth loading studies" (your comment on p2,120) would make more sense to me.

We have restructured the materials under the new Section 4, where we

- contextualize why we need to consider all contributors of relative sea-level (RSL) change in a mass conserving Earth System framework in order to compute  $\Delta H_S$  and  $\mathcal{T}$  (page 12 lines 2-14);
- introduce non-steric components of RSL including the one induced by  $\Delta H_S$  itself over the period  $\Delta t$  (page 12 lines 15-27), and interpret them in light of space observations and existing GRD models (page 12 lines 27-31);
- dissect ΔHS into two parts: the component that modulate both the mass and volume of the oceanic water and drives the GRD response of solid Earth, and the (smaller) component that only modulates the ocean volume (page 12 line 32 page 13 line 6);
- summarize how the spatial patterns of the "mass" component is computed, which necessitates conservation of mass in the Earth system (page 13 lines 9-18).

As noted earlier, we have also changed the title suitably. And, we believe that the newly added two panels in Figure 3 (Figures 3d-e) further strengthen the utility of the proposed

method. We have also revised the Conclusion to even out the importance of the generic level-set for coastlines and grounding lines and the new method for estimating ice sheets' contribution to the sea-level change.

Fig.3: What is the grey line? And in the caption: Is "conventional"=="traditional"? I guess yes. Please use the same wording in the whole text. Eq. 3 and 4: consider rewriting with "latex-cases".

The gray (now black) line shows the present-day ice-ocean interface, now mentioned in **the figure caption**. Yes, the quoted words are equivalent. We have accommodated suggested changes when possible.

**A [..\* ]kinematic formalism for [..† ]tracking ice-ocean mass exchange on the Earth's surface and [..‡ ]estimating sea-level change**

Surendra Adhikari1, Erik R. Ivins1, Eric Larour1, Lambert Caron1, and Helene Seroussi1 1Jet Propulsion Laboratory, California Institute of Technology, Pasadena, CA 91109, USA. **Correspondence:** Surendra Adhikari (surendra.adhikari@jpl.nasa.gov)

[revised manuscript text omitted]

<sup>27removed: observation and numerical simulation

<sup>28removed: complex geometric featuresincluding floating ice shelves,

<sup>29removed: and retrograde bedrock slopes.

<sup>30removed: traditional theory for ice-bedrock-ocean

<sup>32removed: definition

<sup>33removed: sheet domains and that of coastline and grounding line positions

and distributed system of ice domains  $[...^{34}]$ , comprising glaciers, ice sheets and ice shelves, that can be straightforwardly

- 5 employed in any Earth System model in order to track the global mass transport and assess the evolution of a dynamic system of ice sheets, solid Earth and sea level. [..35] In Section 3[..36], we briefly review the common method of estimating sea-level contribution of ice sheets and present a new method, wherein we isolate mass and volume contributions to the ocean, which is critical to accurately drive the GRD response of the solid Earth. In Section 4, we assess our formalism in a broader context of sea-level change and mass conservation in the Earth System. Finally, in Section [..37]5, we summarize
- 10 the key conclusions[ $..^{38}$ ].

**2 Land, ocean and ice [..39] domains and their [..40] interfaces**

To begin our discussion[..41], we consider a spherical planet whose surface is divided into complementary domains of land and ocean. The ocean may be thought of as an interconnected system of oceanic basins – just like Earth's ocean that also includes fjords and marginal seas such as Mediterranean – that are able to freely exchange and redistribute mass between them. This assumption simplifies what would otherwise be an arduous task for mass attribution and conservation in the Earth System. Distributed ice domains including glaciers[..42], ice sheets and ice shelves exist on the land or the ocean (Figure 1). We [..43]generally consider ice domains as part of the land, except where they float on the oceanic water as ice shelves. In order to present mathematical descriptions of these domains and their interfaces at time *t*, we denote 2-D spatial coordinates on the planetary surface [..44]by  $\omega$ . Depending upon the spatial scale [..45](e.g., the ocean versus glaciers), we interchangeably use  $\omega$  to represent [..46]geographic coordinates ( $\theta$ , $\phi$ ) or Cartesian [..47](*x*,*y*), assuming that an appropriate coordinate transformation is applied, [..48]

15

44removed: at time t

<sup>34removed: comprised of glaciersand ice sheets of arbitrary geometric configurations

[revised manuscript text omitted]

---

## Author Response (AR2)

07/15/2020

Dear Editor,

Thank you for your positive response. We really appreciate your smooth handling of the review process. We have updated the manuscript by considering all of your editorial suggestions (see below). We hope that you will find the revised version suitable for publication.

On behalf of co-authors,
Surendra Adhikari
Jet Propulsion Lab, Caltech.

**Response to the Editor's comments.**

To the authors,

You have done an excellent job responding to the reviewers' comments, which were also quite thorough. I believe the quality of the manuscript and its clarity have improved substantially. The method is now easy to understand and clearly structured. I only have minor revisions to suggest before publication.

Thank you for your kind words. We also believe that the revised manuscript is substantially improved both in terms of clarity and content. We truly appreciate thorough and constructive comments from all three reviewers.

P3L14: "We generally consider ice domains as part of the land, except where they float on the oceanic water as ice shelves." <= This sentence seems a bit unnecessary. If you define the ice domains as in the previous sentence, then you do not need to "consider" them one way or another. I would suggest deleting this sentence, but in the previous sentence modify "or the ocean" to be "or floating on the ocean".

Agreed. We have now deleted the quoted sentence and revised the previous sentence as suggested.

P4, Fig. 1 caption: blue sheds => blue shading [two times]

Done.

P6L30: This sentence sounds a bit strange to me: "Such a region, when physically connected to the ocean by oceanic water, can take up water and contribute to sea-level fall." <= According to your equations, this would only be true when the region is floating, and then H_F would necessarily be zero. Correct? I think I understand the sentiment, but perhaps rephrasing is needed.

The Sector D in Figure 2a, for example, has negative H_F but is still part of the grounded (not floating) ice. Had it been in "contact" with ocean water, this region would have already been floated and become part of floating ice. But because it is "shielded" by the ice that has positive H_F (Sector B between Sectors D and C), it cannot really float. When ice sheet disintegrates, for instance, Sector D (unlike the floating portions of the ice sheet, aka ice shelf or Sector C) can take up oceanic water (because it has "negative" potential) and contribute to sea-level fall. We have now rewritten lines 23-30 on page 6. We hope that it is clearer. Please let us know if you want to do one more iteration on this.

P9L30: "More importantly, the change in ice thickness in this regime can affect the interior-ice sheet dynamics via modulation of buttressing force (e.g., Gudmundsson et al., 2019) and may amplify the future sea-level change." <= Perhaps add that this is not treated here for clarity, or even remove the sentence as it is outside the scope.

Agreed. To avoid the potential confusion, we have now removed this sentence and the cited reference. This, in fact, improves the flow of the write-up.

P11, Fig. 3 caption: Mislabeled reference to panel (e) as (d).

Done.

P11, Fig. 3e: Consider label change from "Improvement to the HAF method [%]" to something more objective like "Relative difference with HAF method [%]".

Agreed. We have changed the label as suggested. We have also added the x-axis labels "year [AD]" in panels d and e.

P14L2: bookkeeping the => bookkeeping of the

Done.

Best regards,
Alex